# Learning with Statistical Equality Constraints

**Aneesh Barthakur**
University of Stuttgart
aneesh.barthakur@simtech.uni-stuttgart.de

**Luiz F. O. Chamon**
École polytechnique, Institut Polytechnique de Paris
luiz.chamon@polytechnique.edu

## Abstract

As machine learning applications grow increasingly ubiquitous and complex, they face an increasing set of requirements beyond accuracy. The prevalent approach to handle this challenge is to aggregate a weighted combination of requirement violation penalties into the training objective. To be effective, this approach requires careful tuning of these hyperparameters (weights), involving trial-and-error and cross-validation, which becomes ineffective even for a moderate number of requirements. These issues are exacerbated when the requirements involve parities or equalities, as is the case in fairness and boundary value problems. An alternative technique uses constrained optimization to formulate these learning problems. Yet, existing approximation and generalization guarantees do not apply to problems involving equality constraints. In this work, we derive a generalization theory for equality-constrained statistical learning problems, showing that their solutions can be approximated using samples and rich parametrizations. Using these results, we propose a practical algorithm based on solving a sequence of *unconstrained*, *empirical* learning problems. We showcase its effectiveness and the new formulations enabled by equality constraints in fair learning, interpolating classifiers, and boundary value problems.

## 1   Introduction

Across a wide range of domains, machine learning (ML) is becoming the core technology driving entire systems rather than specific components. It therefore increasingly faces multi-faceted problems involving not only accuracy, but also requirements such as fairness [1], robustness [2, 3], privacy [4], and safety [5, 6]. The standard approach to handling multiple criteria is to represent them as alternate loss functions (*penalties*) aggregated into a single training objective [7–9, 3]. Designing effective penalties and aggregation weights, however, is often a time consuming process involving trial-and-error, hyperparameter search, and cross-validation data. Hence this approach becomes unwieldy even for a moderate number of criteria.

Constrained learning offers an alternative by framing each requirement as a constraint rather than a penalty [10–12]. While in convex settings these approaches are equivalent [13], this is not the case for the non-convex optimization problems arising in ML. Nevertheless, recent work has shown that such a duality still holds for rich parametrizations under mild conditions. This result has been used to establish generalization guarantees similar to those of unconstrained learning theory and develop practical algorithms based on (primal-)dual methods [11, 12]. Yet, these results hold only for inequality constraints and do *not* account for equality-constrained learning tasks.

39th Conference on Neural Information Processing Systems (NeurIPS 2025).

This gap is significant. Indeed, many important requirements are expressed as equality constraints, including group fairness [14–16], invariance [17–19], calibration [20, 21], interpolation [22], distribution matching [23], and independence [24]. These are fundamentally different from inequalities since they impose a specific target value rather than a bound, making their feasibility set and sensitivity more intricate to analyze. As such, reformulations of these requirements as inequalities or relaxations based on penalties can substantially change the solution of an ML task. In fact, none of the duality or generalization results obtained for inequality-constrained learning apply to these problems (see Remark 2.1).

In this paper, we address this knowledge gap by developing a generalization theory for equality-constrained learning tasks. To do so, we derive additional regularity conditions under which we obtain duality and sensitivity results for non-convex equality-constrained optimization problems (Section 3). Based on these results, we put forward an empirical, unconstrained (saddle-point) problem and characterize the approximation and generalization error of its solutions, showing a trade-off between model capacity ("bias"), sample size ("variance"), and constraint difficulty (Theorem 3.1). This problem is amenable to a practical dual ascent algorithm (Section 4) whose effectiveness we illustrate in learning problems involving fairness, boundary value problems, and interpolating classifiers (Section 5).

## 2 Problem formulation

### 2.1 Equality-constrained learning

Consider data pairs $(\mathbf{x}, y) \in \mathcal{X} \times \mathcal{Y}$ and a model $f_\theta : \mathcal{X} \to \mathbb{R}^K$ parametrized by a finite dimensional vector $\theta \in \Theta \subset \mathbb{R}^p$ and denote by $\mathcal{H} = \{f_\theta : \mathcal{X} \to \mathcal{Y} \mid \theta \in \Theta\}$ the hypothesis class induced by these models. The goal of classical (unconstrained) learning is to use $f_\theta$ to map a *feature vector* $\mathbf{x} \in \mathcal{X} \subseteq \mathbb{R}^d$ to its *target* $y \in \mathcal{Y}$, which can be continuous ($\mathcal{Y} \subseteq \mathbb{R}^K$, e.g., regression) or discrete ($\mathcal{Y} = \{1, \ldots, K\}$, e.g., classification). This is usually done by minimizing the expected value of *one* loss function $\ell : \mathbb{R}^K \times \mathcal{Y} \to \mathbb{R}$ describing a top-line objective, e.g., accuracy, with respect to *one* distribution $\mathbb{D}$. In contrast, *constrained learning* accounts for additional loss functions $g_i, h_j$ and distributions $\mathbb{P}_i, \mathbb{Q}_j$ by tackling

$$
\begin{aligned}
P^\star = \inf_{\theta \in \Theta} \quad & \mathbb{E}_{\mathbb{D}}\big[\ell(f_\theta(\mathbf{x}), y)\big] \\
\text{subject to} \quad & \mathbb{E}_{\mathbb{P}_i}\big[g_i(f_\theta(\mathbf{x}), y)\big] \leq 0, \quad \text{for } i = 1, \ldots, I, \\
& \mathbb{E}_{\mathbb{Q}_j}\big[h_j(f_\theta(\mathbf{x}), y)\big] = 0, \quad \text{for } j = 1, \ldots, J.
\end{aligned}
\tag{P}
$$

We consider homogeneous constraints (whose right-hand side is zero) without loss of generality since any constant can be absorbed into the losses $g_i, h_j$. We denote the set of optimal solutions for (P) by Opt(P) and the set of feasible solutions (i.e., those that satisfy the constraints of (P)) by Feas(P). These additional expected losses can be used to impose a rich class of constraints and ancillary requirements based on data, such as fairness [10, 11], robustness [25], and invariance [19]. These works and the majority of the constrained learning literature considers problems with inequality constraints [$J = 0$ in (P)]. In contrast, this paper is concerned with problems explicitly involving *equality* constraints. Before proceeding, we illustrate a few instances in which this additional expressiveness is beneficial (see also Section 5).

**Fairness.** Statistical definitions of fairness in ML are naturally formulated as *equality constraints* [26, 27]. Consider, for instance, demographic parity (DP) in binary classification, where the score $f_\theta(\mathbf{x}) \in [0, 1]$ is thresholded to determine a positive ($f_\theta(\mathbf{x}) > 0.5$) or negative ($f_\theta(\mathbf{x}) \leq 0.5$) outcome. DP requires that the prevalence of positive (or negative) outcomes within the protected groups $\{\mathcal{G}_j\}$ be the same as that of the whole population [28, 29, 14]. Protected groups are often based on a sensitive attribute of the feature vector $\mathbf{x}$, although we impose no such restrictions. DP-constrained binary classification can be cast as (P), explicitly,

$$
\begin{aligned}
\underset{\theta \in \Theta}{\text{minimize}} \quad & \mathbb{E}_{\mathbb{P}}\big[\ell(f_\theta(\mathbf{x}), y)\big] \\
\text{subject to} \quad & \mathbb{E}_{\mathbb{P}}\big[\mathbb{I}\left[f_\theta(\mathbf{x}) > 0.5\right] \mid \mathbf{x} \in \mathcal{G}_j\big] = \mathbb{E}_{\mathbb{P}}\big[\mathbb{I}\left[f_\theta(\mathbf{x}) > 0.5\right]\big], \quad \text{for } j = 1, \ldots, J,
\end{aligned}
\tag{P-DP}
$$

where $\ell$ is a classification loss and $\mathbb{I}[\mathcal{E}] = 1$ on the event $\mathcal{E}$ and 0 otherwise. Note that (P-DP) indeed has the form (P), for $h_j(f_\theta(\mathbf{x}), y) = \frac{1}{\mathbb{P}(\mathbf{x} \in \mathcal{G}_j)} \mathbb{I}\left[f_\theta(\mathbf{x}) > 0.5\right] \mathbb{I}\left[\mathbf{x} \in \mathcal{G}_j\right] - \mathbb{I}\left[f_\theta(\mathbf{x}) > 0.5\right]$ and

$\mathbb{Q}_j = \mathbb{P}$. Other statistical definitions of fairness, such as equality of opportunity, can be similarly formulated [15]. While the equality in (P-DP) is sometimes approximated by an inequality with some slack [10, 11], this can change the solution in non-trivial ways (see Figure 1).

Equality constraints also enable *prescriptive* forms of equity that enforce specific rates $r_j > 0$ of positive outcomes for each group $\mathcal{G}_j$, which can be cast as

$$
\begin{aligned}
\underset{\theta \in \Theta}{\text{minimize}} \quad & \mathbb{E}_{\mathbb{P}}\big[\ell(f_\theta(\mathbf{x}), y)\big] \\
\text{subject to} \quad & \mathbb{E}_{\mathbb{P}}\big[\mathbb{I}\left[f_\theta(\mathbf{x}) > 0.5\right] | \mathbf{x} \in \mathcal{G}_j\big] = r_j, \quad \text{for } j = 1, \ldots, J.
\end{aligned}
\tag{P-F}
$$

**Boundary value problems.** Boundary value problems (BVPs) arise in many scientific applications. While they can be solved with classical methods such as the finite element methods [30, 31], they have recently also been tackled using learning methods by directly parametrizing their solution with a neural network (e.g. PINNs [32]). Indeed, let $\Omega \subseteq \mathbb{R}^d$ be a bounded connected region with boundary $\partial\Omega$ and define the domain $\mathcal{D} = \Omega \times (0, T]$ (where $T > 0$) and let $\mathcal{H} = \{f_\theta \mid \theta \in \Theta\}$ be a set of functions defined on $\mathcal{D}$. A BVP is typically posed as

$$
\begin{aligned}
\text{find} \quad & f_\theta \in \mathcal{H} \\
\text{subject to} \quad & D[f_\theta](\mathbf{x}, t) = \tau_f(\mathbf{x}, t), && \forall (\mathbf{x}, t) \in \mathcal{D}, \\
& f_\theta(\mathbf{x}, t) = \tau_b(\mathbf{x}, t), && \forall \mathbf{x} \in \partial\Omega, t \in (0, T], \\
& f_\theta(\mathbf{x}, 0) = \tau_i(\mathbf{x}, 0), && \forall \mathbf{x} \in \Omega,
\end{aligned}
\tag{BVP}
$$

where $D$ is a differential operator defined over a superset of $\mathcal{H}$, $\tau_f$ is called the forcing function, and $\tau_b, \tau_i$ describe the boundary and initial conditions (BC and IC) respectively. It turns out that (BVP) can be solved with an instance of (P) as in

$$
\begin{aligned}
\underset{\theta \in \Theta}{\text{minimize}} \quad & \frac{\alpha}{2}\|\theta\|_2^2 \\
\text{subject to} \quad & \mathbb{E}_{\mathbb{P}_p}[(D[f_\theta](\mathbf{x}, t) - \tau(\mathbf{x}, t))^2] = 0, \\
& \mathbb{E}_{\mathbb{P}_b}[(f_\theta(\mathbf{x}, t) - \tau_b(\mathbf{x}, t))^2] = 0, \\
& \mathbb{E}_{\mathbb{P}_i}[(f_\theta(\mathbf{x}, 0) - \tau_i(\mathbf{x}, 0))^2] = 0,
\end{aligned}
\tag{P-BVP}
$$

where $\mathbb{P}_p, \mathbb{P}_b$, and $\mathbb{P}_i$ are arbitrary distributions (usually uniform) over $\mathcal{D}$, $\partial\Omega \times (0, T]$, and $\Omega$ respectively [33]. Thus, the constraints enforce *mean squared error* versions of the PDE together with the BC, and the IC. This is in contrast to PINNs that aggregate the errors into a single weighted loss, leading to solutions sensitive to the choice of weights [34, 35].

**Interpolating classifiers.** Modern ML models are typically overparametrized and often trained to perfectly fit (interpolate) the training data. Several works [36, 37, 22] have found that interpolating models performs well in practice, contrary to conventional statistical wisdom on overfitting. However, overparametrization leads to problems with multiple optimal solutions that do not all share the same performance. Hence, the quality of the interpolating model is to a large extent determined by the training algorithm. Alternate formulations of the prediction problem can therefore lead to different interpolating algorithms with beneficial properties.

Explicitly, consider a multi-class classification problem ($\mathcal{Y} = \{1, \ldots, K\}$) with a non-negative loss function $\ell$, vanishing when $f_\theta(\mathbf{x}) = y$ (e.g., cross entropy). Instead of directly minimizing the expected loss $J(\theta) = \mathbb{E}_{\mathbb{P}}[\ell(f_\theta(\mathbf{x}), y)]$, we can use (P) to formulate a *classwise interpolation* problem, namely,

$$
\begin{aligned}
\underset{\theta \in \Theta}{\text{minimize}} \quad & \frac{\alpha}{2}\|\theta\|_2^2 \\
\text{subject to} \quad & \mathbb{E}_{\mathbb{P}}\left[\ell(f_\theta(\mathbf{x}), y) | y = k\right] = 0, \quad \text{for } k = 1, \ldots, K.
\end{aligned}
\tag{P-CI}
$$

As shown by our experiments in Section 5, (P-CI) measures and exploits the heterogeneous difficulty of fitting each class— information that is hard to obtain from the training data given that it is interpolated.

## 2.2 Empirical dual formulation

In ML, the objective/constraints of (P) are non-convex functions of $\theta$, either because the parametrization $f_\theta$ is a complex nonlinear function (e.g. a neural network) or because the losses themselves are

non-convex (as in, e.g., (P-DP)). This hinders the use of constrained optimization methods based on projections, conditional gradients, or barrier functions [38–40]. What is more, we only have access to the distribution $\mathbb{D}, \mathbb{P}_i, \mathbb{Q}_j$ through samples, so that the expectations in (P) must be estimated empirically. This leads to errors that affect $P^\star$ and Feas(P) in non-trivial ways. To overcome these issues, we turn to duality-based methods [13]. Explicitly, define the Lagrangian of (P) as

$$L(f_\theta, \lambda, \mu) = \mathbb{E}_\mathbb{D} \left[\ell(f_\theta(\mathbf{x}), y)\right] + \sum_{i=1}^{I} \lambda_i \mathbb{E}_{\mathbb{P}_i} \left[g_i(f_\theta(\mathbf{x}), y)\right] + \sum_{j=1}^{J} \mu_j \mathbb{E}_{\mathbb{Q}_j} \left[h_j(f_\theta(\mathbf{x}), y)\right], \quad (1)$$

for $\lambda \in \mathbb{R}_+^I$ and $\mu \in \mathbb{R}^J$ collecting the $\lambda_i$ and $\mu_j$ respectively. The dual problem of (P) is then

$$D^\star = \sup_{\lambda \in \mathbb{R}_+^I, \mu \in \mathbb{R}^J} \inf_{\theta \in \Theta} L(f_\theta, \lambda, \mu). \tag{D}$$

In this context, the weights $\lambda_i, \mu_j$ are called *dual variables* and the set Opt(D) of all solutions $(\lambda^\star, \mu^\star)$ of (D) is known as the set of *Lagrange multipliers*. The dual (D) is a relaxation of (P), i.e. $D^\star \leq P^\star$ (weak duality) and $P^\star - D^\star$ is called the duality gap. Convex optimization problems are strongly dual, i.e. $P^\star = D^\star$, under certain regularity conditions such as Slater's condition (see [13, Proposition 5.3.1]). Of particular interest in this work is the empirical version of (D),

$$\hat{D}^\star = \sup_{\lambda \in \mathbb{R}_+^I, \mu \in \mathbb{R}^J} \hat{q}(\lambda, \mu) \triangleq \inf_{\theta \in \Theta} \hat{L}(f_\theta, \lambda, \mu), \tag{$\hat{\text{D}}$}$$

where $\hat{q}(\lambda, \mu)$ is the *empirical dual function* defined based on the *empirical Lagrangian*

$$\hat{L}(f_\theta, \lambda, \mu) = \frac{1}{M_0} \sum_{m_0=1}^{M_0} \ell\left(f_\theta(\mathbf{x}_{m_0}), y_{m_0}\right) + \sum_{i=1}^{I} \lambda_i \left[\frac{1}{M_i} \sum_{m_i=1}^{M_i} g_i\left(f_\theta(\mathbf{x}_{m_i}), y_{m_i}\right)\right]$$

$$+ \sum_{j=1}^{J} \mu_j \left[\frac{1}{N_j} \sum_{n_j=1}^{N_j} h_j\left(f_\theta(\mathbf{x}_{n_j}), y_{n_j}\right)\right], \tag{2}$$

that uses independently drawn samples $(\mathbf{x}_{m_0}, y_{m_0}) \sim \mathbb{D}$, $(\mathbf{x}_{m_i}, y_{m_i}) \sim \mathbb{P}_i$, and $(\mathbf{x}_{n_j}, y_{n_j}) \sim \mathbb{Q}_j$.

For inequality-constrained learning problems [$J = 0$ in (P)], prior works [11, 41, 10] have shown that ($\hat{\text{D}}$) provides an effective way of solving (P) by showing that the solutions of ($\hat{\text{D}}$) approximate those of (P) (generalization) and deriving practical algorithms to do so. The goal of this paper is to extend these results to the more general (P). While ($\hat{\text{D}}$) remains an empirical, unconstrained program amenable to be solved using stochastic optimization techniques (see Section 4), it is no longer clear that its solutions generalize to (approximate) those of (P). Indeed, non-convexity hinders the use of classical duality theory to show $P^\star = D^\star$ and the presence of equalities invalidate the results from [11]. What is more, the errors introduced by using empirical, finite sample estimates of the expectations in (P) pose challenges even before considering equality constraints [11, Example 1]. We address these concerns in the sequel after a pertinent remark.

**Remark 2.1.** *While equality constraints can be written as inequalities, these reformulations pose theoretical and numerical challenges. For instance, the feasibility set of* (P) *does not change when replacing its equalities with* $\left(\mathbb{E}_{\mathbb{Q}_j} \left[h_j(f_\theta(\mathbf{x}), y)\right]\right)^2 \leq 0$. *This formulation, however, invalidates current duality and generalization guarantees for constrained learning (see, e.g., [11, 12]), potentially even for convex problems (see, e.g., [42, Theorem 5.5, 5.11]). We may also consider approximating each equality by the pair* $-\epsilon \leq \mathbb{E}_{\mathbb{Q}_j} \left[h_j(f_\theta(\mathbf{x}), y)\right] \leq \epsilon$. *Interestingly, taking* $\epsilon = 0$ *yields the same (empirical) Lagrangians and dual problems as in* (2) *and* ($\hat{\text{D}}$) *(see Appendix D.2), though the feasibility set of the resulting problem has once again no interior. For* $\epsilon > 0$*, it is not straightforward to determine the effect of this relaxation on the solution of* (P) *and the use of very small* $\epsilon$ *can lead to ill-conditioned problems even in the convex case (see Appendix D.1).*

## 3 Generalization error

In this section, we quantify how well the empirical dual problem ($\hat{\text{D}}$) approximates the constrained learning problem (P). We do so by bounding the generalization error $|P^\star - \hat{D}^\star|$ (Theorem 3.1), which can be decomposed as

$$|P^\star - \hat{D}^\star| \leq |P^\star - D^\star| + |D^\star - \hat{D}^\star|. \tag{3}$$

The first term is the *duality gap* of (P) and the second is the *dual estimation error*. The first assumption we make ensures that the dual problems (D) and (D̂) are well-posed. Explicitly, define the constraint value epigraph as

$$\mathcal{C} = \left\{ (\mathbf{u}, \mathbf{v}) \in \mathbb{R}^I \times \mathbb{R}^J \;\middle|\; \begin{array}{ll} \exists \theta \in \Theta \quad \text{s.t.} & \mathbb{E}_{\mathbb{P}_i}\left[g_i(f_\theta(\mathbf{x}), y)\right] \leq u_i, \quad \text{for } i = 1, \ldots, I, \\ \text{and} & \mathbb{E}_{\mathbb{Q}_j}\left[h_j(f_\theta(\mathbf{x}), y)\right] = v_j \quad \text{for } j = 1, \ldots, J \end{array} \right\}. \tag{4}$$

Similarly, we define $\hat{\mathcal{C}}$ by replacing the expectations in (4) with empirical averages as in (2). We will now discuss the assumptions under which Theorem 3.1 is derived, starting with the following.

**Assumption 1.** *There exists $\xi > 0$ such that $B(0^{I+J}, \xi) = \left\{ c \in \mathbb{R}^{I+J} \mid \|c\| \leq \xi \right\} \subseteq int\ (\mathcal{C}) \cap int\ (\hat{\mathcal{C}})$ where int denotes the interior of the set and $0^{I+J}$ is the origin of $\mathbb{R}^{I+J}$.*

Assumption 1 ensures that (D) and (D̂) have non-empty and compact solution sets (see e.g. [13, Proposition 4.4.1, 4.4.2]). It can be seen as a stronger version of Slater's condition [43] used in convex optimization, and in particular, implies that $P^\star$ exists and is finite.

**Duality gap.** The bound on the duality gap is based on analyzing the properties of (P) when $\mathcal{H}$ is large enough to approximate benign function classes that have the property of *decomposability*. We say that a set $\Phi$ is *decomposable* [44], if for any $\phi_1, \phi_2 \in \Phi$ and measurable subset $\mathcal{Z}$,

$$\phi_3(\mathbf{x}) = \begin{cases} \phi_1(\mathbf{x}), & \mathbf{x} \in \mathcal{Z} \\ \phi_2(\mathbf{x}), & \mathbf{x} \in \mathcal{Z}^c \end{cases}$$

is also a member of $\Phi$. The $L^p$ spaces and their analogue for vector valued functions, Bochner spaces [45] (see also Appendix B.1.1), are decomposable function spaces. We now introduce the *functional* version of (P),

$$\begin{aligned} P_\phi^\star = \inf_{\phi \in \Phi} \quad & \mathbb{E}_{\mathbb{D}}\left[\ell(\phi(\mathbf{x}), y)\right] \\ \text{subject to} \quad & \mathbb{E}_{\mathbb{P}_i}\left[g_i(\phi(\mathbf{x}), y)\right] \leq 0, \quad \text{for } i = 1, \ldots, I, \\ & \mathbb{E}_{\mathbb{Q}_j}\left[h_j(\phi(\mathbf{x}), y)\right] = 0, \quad \text{for } j = 1, \ldots, J. \end{aligned} \tag{$P_\phi$}$$

Notice that the Lagrangian for ($P_\phi$) is the same as in (1). Similarly, its dual problem is given by

$$D_\phi^\star = \sup_{\lambda \in \mathbb{R}_+^I, \, \mu \in \mathbb{R}^J} \inf_{\phi \in \Phi} L(\phi, \lambda, \mu). \tag{$D_\phi$}$$

When the distributions $\mathbb{D}, \mathbb{P}_i, \mathbb{Q}_j$ are atomless probability measures and $\Phi$ is decomposable, ($P_\phi$) is strongly dual, i.e., $P_\phi^\star = D_\phi^\star$ (see Appendix B.2 Proposition B.1). We can therefore bound the duality gap in (3) by the decomposition

$$P^\star - D^\star = P^\star - P_\phi^\star + D_\phi^\star - D^\star. \tag{5}$$

The following assumptions allow (P) to inherit similarly favourable duality properties based on (5).

**Assumption 2.** *Let $\mathbb{P}_+ = \mathbb{D} + \sum_{i=1}^I \mathbb{P}_i + \sum_{j=1}^J \mathbb{Q}_j$, and let $L^p(\mathbb{R}^K; \mathbb{P}_+)$ denote the Bochner space corresponding to $(\mathbb{R}^K, \mathbb{P}_+)$. Assume that*

1. *the distributions $\mathbb{D}, \mathbb{P}_i, \mathbb{Q}_j$ are atomless probability measures, and,*

2. *there exists a decomposable set $\Phi \subseteq L^p(\mathbb{R}^K; \mathbb{P}_+)$ such that $\mathcal{H} \subseteq \Phi$ and a solution $\phi^\star \in Opt(P_\phi)$ such that for some $\theta \in \Theta$ and $\nu > 0$, $\|\phi^\star - f_\theta\|_{L^p(\mathbb{R}^K; \mathbb{P}_+)} \leq \nu$.*

**Assumption 3.** *The loss functions $\ell, g_i, h_j : \mathbb{R} \times \mathcal{Y} \to \mathbb{R}$ are $L$-Lipschitz continuous and the model $f_\theta(\mathbf{x})$ is $L_\theta$-Lipschitz continuous with respect to $\theta$ at every $\mathbf{x} \in \mathcal{X}$.*

Atomlessness is satisfied, for instance, if a measure has a density with respect to the Lebesgue measure (Appendix B.1 Lemma B.2). Several works have studied the approximation of decomposable sets like the $L^p$ spaces by parametrized model classes like neural networks [46–48] and support vector machines [49]. A concrete value of $\nu$ can be found in some cases, for example in [48]. The regularity assumptions on the model and loss functions are mild. Loss functions only need to be Lipschitz continuous on the range of $f_\theta(\mathbf{x})$, as is the case for the Huber loss, square loss, and hinge loss when

the range of $f_\theta(\mathbf{x})$ and $\mathcal{Y}$ is bounded. An example of a model that is Lipschitz with respect to its parameters is a feedforward neural network with Lipschitz activations when the data and parameters are bounded in norm.

To bound the terms in (5), we also use the following quantity that measures the sensitivity of the feasibility set Feas(P), namely,

$$R(\nu) = \sup_{\theta \in \Theta_\nu} \inf_{\theta_0 \in \text{Feas}(P)} \|\theta - \theta_0\|_2 \,,$$

$$\text{where } \Theta_\nu = \left\{ \theta \in \Theta \; \middle| \; \begin{array}{ll} \mathbb{E}_{\mathbb{P}_i}\left[g_i(f_\theta(\mathbf{x}), y)\right] \le L\nu, & \text{for } i = 1, \ldots, I \\ \text{and} \quad \left|\mathbb{E}_{\mathbb{Q}_j}\left[h_j(f_\theta(\mathbf{x}), y)\right]\right| \le L\nu & \text{for } j = 1, \ldots, J \end{array} \right\}. \tag{6}$$

$R(\nu)$ measures the maximum distance between $\Theta_\nu$ and Feas(P). Hence, it is a decreasing function of $\nu$ and vanishes when $\nu = 0$. However, if relaxing the constraints causes the feasibility set to change radically, $R(\nu)$ could be unbounded for $\nu > 0$. This can be avoided by ensuring that $\Theta_\nu$ is bounded, by e.g., regularizing $\Theta$ by enforcing $\|\theta\|_2 \le W$ for some $W > 0$.

**Dual estimation error.** The next assumption allows us to bound the dual estimation error $|D^\star - \hat{D}^\star|$.

**Assumption 4.** *Let $\mathbb{P} \in \{\mathbb{D}, \mathbb{P}_1, \ldots, \mathbb{Q}_J\}$ and let $\left(\mathbf{x}^{(k)}, y^{(k)}\right)_{k=1}^N$ be an i.i.d sample of size $N$ drawn from $\mathbb{P}$. Denote $\mathbb{L} = \{\ell, g_1, \ldots, g_I, h_1, \ldots, h_J\}$. Then, assume that there exists a function $\zeta^{UC}(N, \delta) : \mathbb{N} \times [0, 1] \to \mathbb{R}$, such that, for all $\delta \in (0, 1)$, (a) $\zeta^{UC}$ is strictly decreasing in $N$, (b) satisfies $\lim_{N \to \infty} \zeta^{UC}(N, \delta) = 0$, and (c) for any $\mathcal{L} \in \mathbb{L}$, the following is true :*

$$\mathbb{P}\left(\sup_{\theta \in \Theta} \left| \mathbb{E}_\mathbb{P}\left[\mathcal{L}(f_\theta(\mathbf{x}), y)\right] - \frac{1}{N}\sum_{k=1}^N \mathcal{L}(f_\theta(\mathbf{x}_k), y_k) \right| \le \zeta^{UC}(N, \delta)\right) \ge 1 - \delta. \tag{7}$$

Assumption 4 is called uniform convergence, and is satisfied if e.g. $\mathcal{H}$ has a finite VC-dimension or Rademacher complexity (see [50][Corollary 3.19, Theorem 3.3] and also [11][Proposition III.1]). Bounds on the VC dimension and Rademacher complexity, which are measures of *model complexity*, are available for many model classes [51, 52].

**Main Result.** The generalization error in (3) can be bounded under these assumptions by combining (5) and the uniform convergence bound in Assumption 4. We next use $\gamma$ to denote the tuple of dual variables for inequalities and equalities, i.e., $(\lambda, \mu)$.

> **Theorem 3.1.** *Let $N_{min} = \min\{M_0, M_1 \ldots, M_I, N_1, \ldots, N_J\}$ and assume that $R(\nu) < \infty$ and $P_\phi^\star > -\infty$. Under Assumptions 1-3 there exist $\gamma_\phi^\star \in Opt(D_\phi), \gamma^\star \in Opt(D)$, and $\hat{\gamma}^\star \in Opt(\hat{D})$. Moreover, for any $\delta \in (0, 1)$, it holds with probability at least $1 - (1 + I + J)\delta$, that*
>
> $$|P^\star - \hat{D}^\star| \le \left(1 + \|\gamma_\phi^\star\|_1\right) L\nu + LL_\theta R(\nu) + \left(1 + \max\left\{\|\gamma^\star\|_1, \|\hat{\gamma}^\star\|_1\right\}\right) \zeta^{UC}(N_{min}, \delta). \tag{8}$$

The proof of Theorem 3.1 is deferred to Appendix B. We next discuss the main factors driving the generalization error bound.

The main difference between constrained and unconstrained generalization bounds is that (8) is driven by the *constraint sensitivity*. This is represented by the Lagrange multipliers of (P) and its empirical and functional versions, which appear in the bound through their norms. More specifically, it is well known that for strongly dual problems, Lagrange multipliers are sensitivity measures (subgradients) of the optimal value with respect to the constraint constants. Explicitly, suppose that we were to perturb the constraints of (P) to obtain

$$\begin{aligned} P^\star(\mathbf{c}, \mathbf{d}) = \inf_{\theta \in \Theta} \quad & \mathbb{E}_\mathbb{D}\left[\ell(f_\theta(\mathbf{x}), y)\right] \\ \text{subject to} \quad & \mathbb{E}_{\mathbb{P}_i}\left[g_i(f_\theta(\mathbf{x}), y)\right] \le c_i, \quad \text{for } i = 1, \ldots, I, \\ & \mathbb{E}_{\mathbb{Q}_j}\left[h_j(f_\theta(\mathbf{x}), y)\right] = d_j, \quad \text{for } j = 1, \ldots, J. \end{aligned} \tag{P-pert}$$

The problem (P-pert) defines the (perturbation) function $P^\star(\mathbf{c}, \mathbf{d})$. If (P) were strongly dual (e.g. a convex program) and $P^\star(\mathbf{c}, \mathbf{d})$ is differentiable, then

$$\lambda_i^\star = -\frac{\partial P^\star(0^I, 0^J)}{\partial c_i} \quad \text{and} \quad \mu_j^\star = -\frac{\partial P^\star(0^I, 0^J)}{\partial d_j}, \tag{9}$$

for $i = 1, \dots, I$ and $j = 1, \dots, J$ (see [40, Equation 5.58]). Though differentiability and strong duality of (P-pert) does not generally hold in our setting, *approximate* relations can be obtained in terms of the bound on the duality gap $P^\star - D^\star$ (see Remark B.6). Thus, $\|\gamma^\star\|_1 = \|\lambda^\star\|_1 + \|\mu^\star\|_1$, that appears in (8), measures how much the objective changes when the constraints are perturbed. Constraint sensitivity also appears in terms of $R(\nu)$, which marks a clear difference from the generalization bounds for inequality constraints ($J = 0$) that depend solely on $\|\lambda^\star\|_1$ (see Remark 3.1). The bound in (8) also depends on the sensitivity of the loss functions and the model (with respect to its parameters) through their Lipschitz constants $L$ and $L_\theta$ respectively.

While these factors are dictated by the problem formulation, Theorem 3.1 shows us that they can be mitigated by using richer parametrizations (i.e., reducing $\nu$) and larger datasets (i.e., increasing $N_{\min}$). Indeed, as $\nu \to 0$, the first set of terms (relating to the duality gap) vanish. However, the sample complexity $\zeta^{\mathrm{UC}}(N_{\min}, \delta)$ typically increases with larger model classes, i.e., as $\nu$ decreases. Hence, we find a trade-off between approximation error and estimation error that mirrors the trade-off in unconstrained learning [53, Section 5.2].

To summarise, Theorem 3.1 reveals *four* key drivers of the generalization error : (a) constraint sensitivity, (b) sensitivity of the losses and parametrization, (c) model capacity, and (d) sample size. The bound also mirrors the classical decomposition of the unconstrained learning error into an approximation error (here, the duality gap $P^\star - D^\star$) and the estimation error (here, the dual estimation error $|D^\star - \hat{D}^\star|$). We conclude this section with a few remarks.

**Remark 3.1** (Comparison with results for inequality constraints). *Notably, the bounds for problems involving only inequality constraints from [12, 11] do not depend on the smoothness of the parametrization $L_\theta$ (Assumption 3). This fundamental distinction is rooted in the fact that in the absence of equality constraints ($J = 0$) going from $\Phi$ to $\mathcal{H}$ is akin to contracting the functional feasibility set $\mathrm{Feas}(P_\phi)$ by tightening the functional inequality constraints. Consequently, ($P_\phi$) and its parametrized formulation (P) remain closely aligned, and the latter approximately inherits the duality properties of the former. The equality constraints in (P) however make it so that the feasibility sets are no longer nested. Changing the targets lead to more intricate changes in the set of feasible parameters $\theta$. The sensitivity of the parametrization $L_\theta$ therefore affects generalization. This is also the reason why Assumption 1 reduces to the existence of a strictly feasible solution when $J = 0$. Indeed, the presence of equalities requires the stronger regularity conditions in Assumption 1 to ensure that (P) is feasible for any small perturbation. The reader is referred to Appendix B.5 for additional technical distinctions.*

**Remark 3.2** (Functional strong duality). *The core result underpinning Theorem 3.1 is the strong duality of the functional problem ($P_\phi$) (Proposition B.1). The crux of the proof lies in proving that the cost-constraint epigraph,*

$$\mathcal{M}_\phi = \left\{ (f, \mathbf{u}, \mathbf{v}) \in \mathbb{R}^{1+I+J} \left| \begin{array}{ll} \exists \phi \in \Phi \quad s.t. \quad \mathbb{E}_{\mathbb{D}}\left[\ell(\phi(\mathbf{x}), y)\right] = f, \\ \qquad\qquad\qquad \mathbb{E}_{\mathbb{P}_i}\left[g_i(\phi(\mathbf{x}), y)\right] \le u_i, \quad for\ i = 1, \dots, I, \\ \quad and \quad \mathbb{E}_{\mathbb{Q}_j}\left[h_j(\phi(\mathbf{x}), y)\right] = v_j \quad for\ j = 1, \dots, J, \end{array} \right. \right\}$$

*is convex under Assumption 2. Thus, while ($P_\phi$) is not a convex optimization problem, a classical result from convex optimization [13, Proposition 4.4.1] can be applied to the previous fact to show that ($P_\phi$) is strongly dual. The convexity of $\mathcal{M}_\phi$ is established using Lyapunov's theorem on the range of atomless measures [54, Chaper IX Corollary 5], closely related to the bang-bang principle in control theory [55]. Proposition B.1 has appeared with minor variations in [56–58, 11], though our proof improves on [11] for the regression case (continuous $\mathcal{Y}$), which we prove without additional assumptions.*

## 4  Algorithm

Theorem 3.1 establishes that solving ($\hat{D}$) provides (approximate) solutions to (P). Next, we propose an algorithm to tackle ($\hat{D}$) based on the traditional dual ascent method. This algorithm assumes access to an oracle for unconstrained problems as in [10, 11], a fact formalised in the following assumption.

**Algorithm 1** Primal-dual constrained learning algorithm

---

1: *Inputs :* Loss functions $\ell, g_i, h_j$, samples $(\mathbf{x}_{m_0}, y_{m_0}) \sim \mathbb{D}, (\mathbf{x}_{m_i}, y_{m_i}) \sim \mathbb{P}_i, (\mathbf{x}_{n_j}, y_{n_j}) \sim \mathbb{Q}_j$
    for $i = 1, \ldots I, j = 1, \ldots, J$, iterations $T \in \mathbb{N}$, dual learning rate $\eta > 0$.
2: *Initialize :* $\lambda^{(0)} \leftarrow \mathbf{0}^I, \mu^{(0)} \leftarrow \mathbf{0}^J$
3: **for** $t = 1, \ldots, T$ **do**
4:     $\theta^{(t)} \leftarrow \mathcal{O}(\hat{L}(f., \lambda^{(t-1)}, \mu^{(t-1)}))$                       $\triangleright$ Assumption 5
5:     $\lambda_i^{(t)} \leftarrow \max\left\{0, \lambda_i^{(t-1)} + \eta\left(\frac{1}{M_i}\sum_{m_i=1}^{M_i} g_i\left(f_{\theta^{(t)}}(\mathbf{x}_{m_i}), y_{m_i}\right)\right)\right\}$     $\triangleright$ for $i = 1, \ldots, I$
6:     $\mu_j^{(t)} \leftarrow \mu_j^{(t-1)} + \eta\left(\frac{1}{N_j}\sum_{n_j=1}^{N_j} h_j\left(f_{\theta^{(t)}}(\mathbf{x}_{n_j}), y_{n_j}\right)\right)$         $\triangleright$ for $j = 1, \ldots, J$
7: **end for**

---

**Assumption 5.** *The solution $\theta^\star(\lambda, \mu) = \mathcal{O}(\hat{L}(f., \lambda, \mu))$ returned by the oracle in Algorithm 1 approximately minimizes the empirical Lagrangian in* (2)*, i.e., for $\rho \geq 0$ it holds that $\hat{L}(f_{\theta^\star(\lambda,\mu)}, \lambda, \mu) \leq \hat{q}(\lambda, \mu) + \rho$ for all $\lambda, \mu$.*

In Algorithm 1, this oracle is used to update the model $\theta$ (primal variable) in Step 4, while the empirical constraint violations (or slacks) are used to update the dual variables $\lambda, \mu$ (Steps 5-6). If the oracle is optimal (i.e., $\rho = 0$), Line 6-7 constitute a projected *subgradient* ascent with respect to the dual objective $\hat{q}(\lambda, \mu)$ [59]. Since the dual objective is always concave, subgradient ascent converges to a global optimum for certain reducing step size rules [59, Chapter 2]. However, the descent is generally non-monotonic, i.e., the last iterate is not necessarily the best iterate. Hence, guarantees are often of the following form. Note that Theorem 4.1 relates the empirical dual value $\hat{D}^\star$ to the *average* of the Lagrangian iterates.

---

**Theorem 4.1.** *Suppose Assumptions 1 and 5 hold and let the loss functions $\ell, g_i, h_j$ be B-bounded. Let $U_0 = \inf_{\gamma^\star \in Opt(\hat{D})} \left\|\gamma^{(0)} - \gamma^\star\right\|$. Then, for any $T \in \mathbb{N}$, it holds that,*

$$\left|\hat{D}^\star - \frac{1}{T}\sum_{t=0}^{T-1}\left(\hat{L}(f_{\theta^{(t)}}, \gamma^{(t)})\right)\right| \leq \rho + \frac{U_0}{2\eta T} + \frac{1}{2}(I+J)\eta B^2. \tag{10}$$

*If $\eta \leq \frac{\rho}{(I+J)B^2}$ and $T \geq \frac{U_0}{\eta\rho}$, then the bound is equal to $2\rho$.*

---

Theorem 4.1 establishes results for the dual iterates, modulo the averaging, but does not characterise the feasibility or optimality of $\theta^{(t)}$. Since the set of primal variables defined by the Lagrange multipliers $(\hat{\lambda}^\star, \hat{\mu}^\star)$ need not be unique, or all feasible, recovering a feasible near-optimal model can be difficult. This is not a substantial issue in convex optimization, where averaging solves this problem (e.g. [59–62]). Non-convex settings often rely on randomization to overcome this challenge (see, e.g. [11, 10, 63]), although there is empirical and theoretical evidence that this is not a substantial issue in ML [10, 11, 64, 65].

In practice, we do not have access to the oracle from Assumption 5. Our experiments in Section 5 show that replacing line 5 with a single (stochastic) gradient descent step can produce feasible solutions (without additional primal recovery techniques) that also perform well with respect to the objective loss.

## 5 Experiments

In this section, we demonstrate the empirical performance of Algorithm 1 on instances of (P) presented in Section 5. Detailed descriptions of the experiments can be found in Appendix E.

**Exact vs. approximate fairness.** Figure 1 compares models trained on the COMPAS dataset [66] using the fairness formulation in (P-DP) against models trained with a double-sided inequality approximation with tolerance parameter $\epsilon$ (see Remark 2.1) and an unconstrained baseline. The

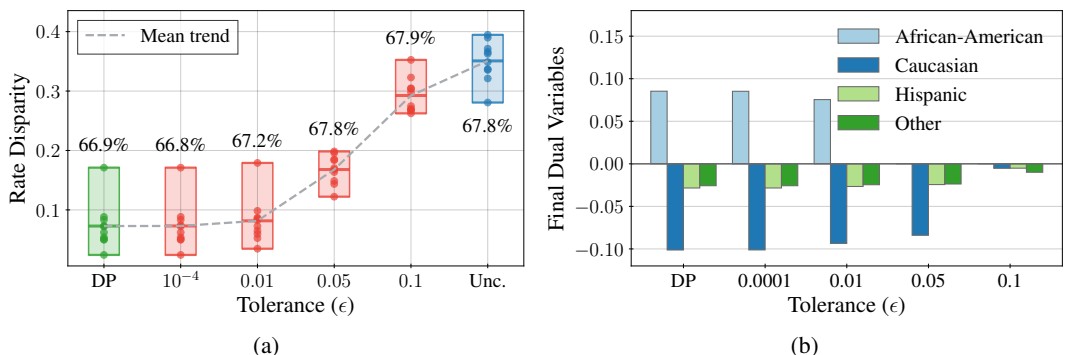

(a)                                                                      (b)

Figure 1: *Exact vs approximate fairness.* (a) Comparison betweeen (P-DP) and the inequality relaxation described in Remark 2.1 with parameter $\epsilon > 0$ (10 random splits). Mean accuracy (across splits) is reported for each method/tolerance. (b) Final (effective) dual variables for Algorithm 1. Indeed, since the inequality relaxation uses two constraints for each group (upper *and* lower bound), we show only the difference between upper and lower dual variables.

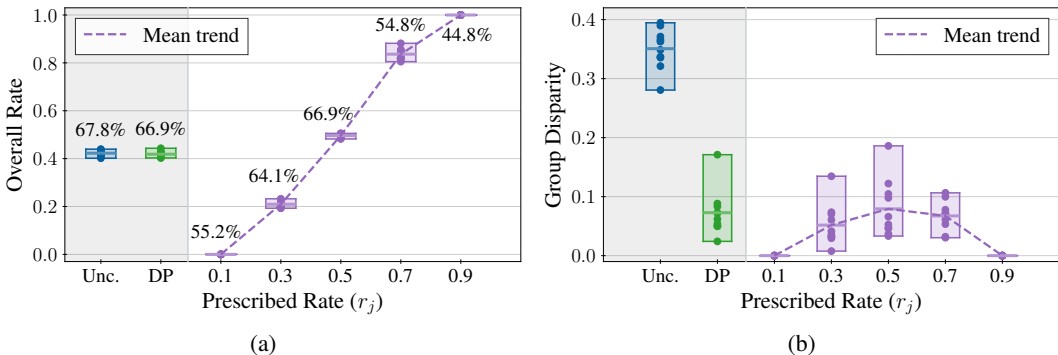

(a)                                                                      (b)

Figure 2: *Prescribed rates.* Solutions of (P-F) for different $r_j$ (10 random splits). (a) Average rate of positive outcomes across population, annotated with the mean accuracy (across splits); (b) Rate disparity across different groups.

indicator functions that appear in the constrained formulations are replaced by sigmoid functions to enable the use of gradient descent to replace the oracle in Step 4 of Algorithm 1. Explicitly, we use

$$\mathbb{E}_{\mathbb{P}}\left[\mathbb{I}\left[f_\theta(\mathbf{x}) > 0.5\right] | \mathbf{x} \in \mathcal{G}_j\right] \approx \mathbb{E}_{\mathbb{P}}\left[\sigma\left(\alpha\left(f_\theta(\mathbf{x}) - 0.5\right)\right) | \mathbf{x} \in \mathcal{G}_j\right]$$

and similarly for the overall rate $\mathbb{E}_{\mathbb{P}}\left[\mathbb{I}\left[f_\theta(\mathbf{x}) > 0.5\right]\right]$, where $\sigma$ denotes the sigmoid function. We split the dataset into a training (70%) and test (30%) set 10 times and report the results.

Figure 1(a) shows that the equality formulation achieves the lowest *disparity* in group rates (measured as the difference between the maximum and minimum group rates) comparable to the inequality-based model with small tolerances (e.g., $\epsilon = 10^{-4}$). Figure 1(b) shows that the final (effective) dual variables for $\epsilon = 10^{-4}$ are also indistinguishable from those of the equality formulation. However, we see that a looser tolerance ($\epsilon = 10^{-2}$) introduces noticeable differences, both in the dual solution and the *group disparity*. This highlights the challenge of selecting an appropriate tolerance, a difficulty circumvented by directly enforcing the equality constraints.

**Prescribed rates.** In Figure 2, we showcase the results of imposing specific group rates using (P-F) and compare them to unconstrained and DP-constrained [i.e., (P-DP)] problems. With the exception of extreme rates (0.1 and 0.9), which yield nearly constant classifiers, models with intermediate targets $r_j$ achieve low group disparities, comparable to those of the DP-constrained model. However, while (P-DP) maintains an overall rate of positive outcomes close to the unconstrained model, (P-F) allows this rate to be adjusted more granularly. This flexibility comes at a negligible difference in accuracy for prescribed rates close to the DP rate. For example, $r_j = 0.5$ achieves the same test accuracy as (P-DP) (Figure 2(a)) and similar (test) group disparities (Figure 2(b)).

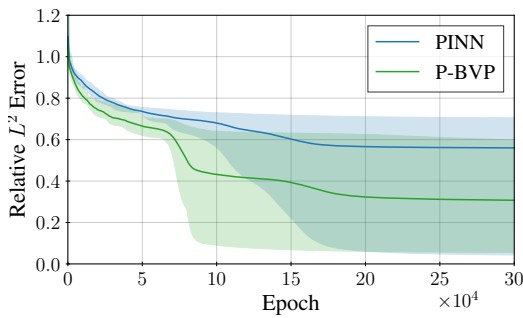

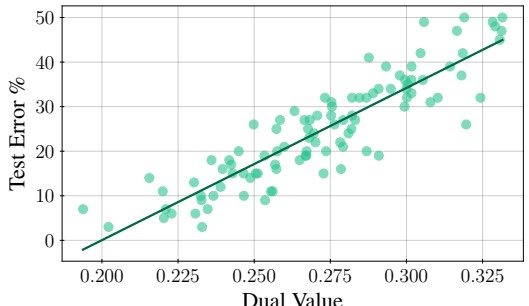

Figure 3: Relative $L^2$ error for convection BVP ($\beta = 50$). The lines show the mean curve across 5 runs, and the shaded region indicates the max and min curves.

Figure 4: Comparison of classwise test errors and dual values for CIFAR-100 trained with (P-CI) (along with the best fit line). The correlation is 0.89 indicating a strong linear relationship.

Table 1: Relative $L^2$ error for convection

| $\beta$ | PINN | (P-BVP) |
|---|---|---|
| 30 | 2.46 ± 0.99 % | 0.62 ± 0.17 % |
| 50 | 56.0 ± 25.8 % | 30.7 ± 24.2 % |

Table 2: Test Accuracy on CIFAR-10/100

| Dataset | ERM | (P-CI) |
|---|---|---|
| CIFAR-10 | 95.03 ± 0.21 % | 95.01 ± 0.10 % |
| CIFAR-100 | 76.11 ± 0.28 % | 75.17 ± 0.21 % |

**Boundary value problems.** Table 1 shows that taking a constrained approach to solving a convection BVP with sinusoidal initial condition (see Appendix E.2 for details) outperforms the unconstrained approach with fixed multipliers for the boundary and initial conditions (PINN). The mean and standard deviation of 5 seeds have been reported. The primary challenge of solving this BVP is propagating the initial condition through time since the solution itself is very regular. Thus, the improvement may be explained by larger dual variables for the boundary conditions (as seen in Appendix F Figure 9).

**Interpolating classifiers.** Table 2 (also computed over 5 seeds) shows that (P-CI) has worse test accuracy compared to the unconstrained problem (ERM) on CIFAR-100—though the gap is not large. On the other hand, solving (P-CI) using Algorithm 1 yields dual variables that exhibit an approximately linear relationship with the mean test error of the class. Since models are trained to interpolation on the train set, this information is not generally available without cross-validation. It can be used as a confidence measure for the performance of the model or to detect biases in the model. Meanwhile, (P-CI) performs virtually the same as ERM on CIFAR-10. This relationship between test error and dual variables may be partially explained by 9, which suggests that a large dual variable indicates a constraint on the class-wise error that significantly contributes to the model complexity ($\|\theta\|_2^2$)—since relaxing the constraint would lead to a large decrease in the parameter norm. Though this norm has been tied to generalization error in [67, 68], a definitive answer would require a more detailed analysis that is beyond the scope of this work.

## 6 Conclusion

In this paper, we studied equality-constrained learning problems, i.e., statistical optimization problems with equality constraints. We extended the existing generalization theory for inequality-constrained problems, showing that equalities are also tractable through Lagrangian methods as long as the parametrization is rich enough. Nevertheless, they demand stronger assumptions than inequalities. We also introduced a practical algorithm based on dual ascent to solve problems with both equality and inequality constraints. We illustrated the behavior of this algorithm in a fair learning problem, showing results for both classical problems, involving demographic parity, as well as new formulations enabled by these equality-constrained problems, namely, learning tasks that enforce specific prediction rates for each group. We also showcase results for solving BVPs and fitting interpolating classifiers.

## Acknowledgements

The work of Aneesh Barthakur is funded by Deutsche Forschungsgemeinschaft (DFG, German Research Foundation) under Germany's Excellence Strategy - EXC 2075 - 390740016. The work of L.F.O. Chamon is supported by the Agence Nationale de la Recherche (ANR) project ANR-25-CE23-3477-01 as well as a chair from Hi!PARIS, funded in part by the ANR AI Cluster 2030 and ANR-22-CMAS-0002. The authors thank the Stuttgart Center for Simulation Science (SimTech) and the International Max Planck Research School for Intelligent Systems (IMPRS-IS) for supporting Aneesh Barthakur and acknowledge the computing time provided on the high-performance computer HoreKa by the National High-Performance Computing Center at KIT (NHR@KIT). This center is supported by the Federal Ministry of Education and Research and the Ministry of Science, Research and the Arts of Baden-Württemberg as well as the DFG.

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

# LEARNING WITH EQUALITY CONSTRAINTS

## ADDITIONAL MATERIAL

# A  Related work

**Unconstrained learning.**  Empirical risk minimization forms the cornerstone of modern machine learning applications, supported by a rich theory of generalization [1–3]. However, as machine learning becomes increasingly embedded in real world applications, there is a growing need to deal with multi-faceted problems involving multiple losses and/or requirements beyond accuracy, such as fairness [4, 5], robustness [6, 7], privacy [8], safety [9, 10], and scientific knowledge [11]. The traditional approach to handling multiple requirements is to use a weighted sum of the loss functions as the objective (e.g., [12–14, 7, 11]), choosing the weights by trial and error, cross-validation, or a problem-specific heuristic. This approach is often brittle and time consuming.

**Inequality-constrained learning.**  Inequality-constrained learning uses constrained optimization to incorporate requirements into traditional learning problems. As in unconstrained learning, these tasks are formulated as statistical risk minimization problem, albeit with inequality constraints. Yet, this leads to non-convex programs for virtually every modern ML model, which make them challenging to solve [15, 16]. In convex settings, classical results for Sample Average Approximation (SAA) methods can be found in [17, 18]. Generalization bounds have also been derived in [19] (for linear classifiers and fairness constraints), [20] (for convex losses, convex-fractional losses, and fairness constraints), and [21] (for fairness constraints). In the general non-convex settings, certain duality properties have been shown to hold when using sufficiently expressive parametrizations, leading to a practical learning rule with generalization guarantees [22, 23]. The resulting primal-dual algorithms can be interpreted as incorporating the combination weights from unconstrained learning into the optimization process and have been used in various ML applications, such as fairness [19, 24, 22, 23, 25], invariance [26], classification [27], and robustness [28]. Our work is an extension of these works on non-convex, inequality-constrained learning problems.

**Equality constrained learning.**  Equality constraints have been used in ML to express important problems, such as group fairness [29–31], invariance [32, 33], calibration [34, 35], distribution matching [36], and independence [37]. While stochastic optimization with *deterministic* equality constraints has been extensively studied in the literature [38–41], less is known about stochastic problems with *statistical* equality constraints—e.g., constraints defined via expectations over data, such as those considered in this work (Section 2).

Equality constraints are often incorporated directly into the model (see, e.g., [42, 43]) or enforced by post-processing schemes (e.g., group fairness in [30]). When the constraint function and its derivatives are only accessible by noisy oracles, solutions based on upper/lower bounds as well as Sequential Quadratic Programming (SQP) or trust region methods have been investigated [44–47].

More similar to our setup is [48] that uses a primal-dual approach similar to ours, but rely on transforming equalities into inequalities using a non-negative wrapping function (e.g., quadratic). While their empirical results are promising, this approach has severe numerical and theoretical issues as discussed in Remark 2.1. From a theoretical perspective, [49] considers the same problem we do, extending the SAA framework to equality constraints using inequality *relaxations*. They show that if the relaxation is carefully tightened as the sample size increases, it is possible to asymptotically obtain solutions of the population problem. These results are, nevertheless, (a) asymptotic, assuming access to infinitely many i.i.d. samples; (b) focused on a relaxation of the original problem; and (c) reliant on interior-point methods that are not well-suited to the large-scale, non-convex settings of ML.

# B Proof of Theorem 3.1

## B.1 Preliminaries

### B.1.1 Bochner spaces

Let $(\mathcal{X} \times \mathcal{Y}, \Sigma_{\mathcal{X} \times \mathcal{Y}}, \mathbb{P})$ be the probability space [50] corresponding to the random variables $(\mathbf{x}, y)$. Let $(\mathbb{R}^K, \mathcal{B}(\mathbb{R}^K))$ be the Borel sigma algebra associated with $\mathbb{R}^K$, i.e. the smallest sigma algebra containing all the open sets. The Bochner spaces [51][Section 1.2.b] are a direct generalization of the $L^p$ spaces, for measurable *vector valued* functions. Consider mappings from $\mathcal{X}$ to $\mathbb{R}^K$. The Bochner space $L^p(\mathbb{R}^K; \mathbb{P})$, for $p \in [1, \infty)$, is the space of all vector valued functions $\phi : \mathcal{X} \to \mathbb{R}^K$, such that

$$\|\phi\|_{L^p(\mathbb{R}^K; \mathbb{P})} = \left( \int \|\phi(\mathbf{x})\|_2^p \ d\mathbb{P} \right)^{1/p}$$

is finite. $L^\infty(\mathbb{R}^K; \mathbb{P})$ is similarly defined by the norm,

$$\|\phi\|_{L^\infty(\mathbb{P})} = \mathrm{esssup}_{\mathbf{x} \in \mathcal{X}; \mathbb{P}} \|\phi(\mathbf{x})\|_2 = \sup \ \{c > 0 \mid \mathbb{P}(\{\|\phi(\mathbf{x})\|_2 \geq c\}) = 0\}.$$

For $p \in [1, \infty]$ the spaces $L^p(\mathbb{R}^K; \mathbb{P})$ are Banach spaces. The space $L^1(\mathbb{R}^K, \mathbb{P})$ is precisely the space of functions for which the Bochner integral, a form of vector integral, exists.

Our primary space of interest, $L^p(\mathbb{R}^K; \mathbb{P}_+)$, where $\mathbb{P}_+ = \mathbb{D} + \sum_{i=1}^I \mathbb{P}_i + \sum_{j=1}^J \mathbb{Q}_j$, is actually the intersection of the Bochner spaces corresponding to the summands. This follows from the linearity of the Lebesgue integral (for scalar functions) with respect to the measure ([50] Exercise 4.27).

---

**Lemma B.1.** *Let $\mathbb{B}, \mathbb{P}$ be positive measures on a measurable space, and $f : \mathcal{X} \to \mathbb{R}$ be a non-negative measurable function. Then,*

$$\int f(\mathbf{x}) \ d(\mathbb{B} + \mathbb{P}) = \int f(\mathbf{x}) \ d\mathbb{B} + \int f(\mathbf{x}) \ d\mathbb{P}.$$

*and moreover, (a) $L^1(\mathbb{B} + \mathbb{P}) = L^1(\mathbb{B}) \cap L^1(\mathbb{P})$, and more generally (b) $L^p(\mathbb{R}^K; \mathbb{B} + \mathbb{P}) = L^p(\mathbb{R}^K; \mathbb{B}) \cap L^p(\mathbb{R}^K; \mathbb{P})$ for $p \in [1, \infty]$.*

---

*Proof. Part 1 :* Suppose $f \geq 0$. Let $\mathcal{X}_A$ be the characteristic function of a measurable set $A$. Recall that there exists a sequence of positive simple functions $f_n(\mathbf{x}) = \sum_{i=1}^n \alpha_i \mathcal{X}_{A_i}(\mathbf{x})$ that are upper bounded by $f$ and converge to $f$. The integral of $f_n$ (with respect to any measure on the same measurable space) is an increasing convergent sequence whose limit is defined as the integral of $f$ (see [50] Definition I.4.3(b) and Theorem I.2.17 for more details).

By applying the fact that $(\mathbb{P} + \mathbb{B})(A) = \mathbb{P}(A) + \mathbb{B}(A)$ (by definition) for any measurable set $A$, we obtain that,

$$\int f_n(\mathbf{x}) \ d(\mathbb{B} + \mathbb{P}) = \int f_n(\mathbf{x}) \ d\mathbb{B} + \int f_n(\mathbf{x}) \ d\mathbb{P}.$$

If the limits (with respect to $n$) of both terms on the RHS exist, then the limit on the LHS exists as well ([52] Theorem 3.4). This proves that $f \in L^1(\mathbb{B}) \cap L^1(\mathbb{P}) \Rightarrow f \in L^1(\mathbb{B} + \mathbb{P})$.

On the other hand, note that $\int f_n(\mathbf{x}) \ d\mathbb{B} \leq \int f_n(\mathbf{x}) \ d(\mathbb{B} + \mathbb{P})$ since $\int f_n(\mathbf{x}) \ d\mathbb{P} \geq 0$. Therefore if $\int f_n(\mathbf{x}) \ d(\mathbb{B} + \mathbb{P})$ is convergent, since $\int f_n(\mathbf{x}) \ d(\mathbb{B} + \mathbb{P}) \leq \int f(\mathbf{x}) \ d(\mathbb{B} + \mathbb{P})$, therefore,

$$\int f_n(\mathbf{x}) \ d\mathbb{B} \leq \int f(\mathbf{x}) \ d(\mathbb{B} + \mathbb{P}).$$

Therefore $\int f_n(\mathbf{x}) \ d\mathbb{B}$ is a convergent sequence since it is increasing and bounded (and by symmetry so is $\int f_n(\mathbf{x}) \ d\mathbb{P}$). This proves that $f \in L^1(\mathbb{B} + \mathbb{P}) \Rightarrow f \in L^1(\mathbb{B}) \cap L^1(\mathbb{P})$. Therefore for a non-negative function $f \in L^1(\mathbb{B} + \mathbb{P}) \iff f \in L^1(\mathbb{B}) \cap L^1(\mathbb{P})$.

*Part 2 :* Now, consider a general $f$. It is known [53][Theorem 1.33] that $f \in L^1(\mathbb{P} + \mathbb{B})$ iff,

$$\|f\|_{L^1(\mathbb{P})} = \int |f(\mathbf{x})| \ (d\mathbb{P} + d\mathbb{B}) < \infty.$$

Therefore,

$$f \in L^1(\mathbb{B} + \mathbb{P}) \iff |f| \in L^1(\mathbb{B} + \mathbb{P}) \tag{11}$$
$$\iff |f| \in L^1(\mathbb{B}) \cap L^1(\mathbb{P}) \tag{12}$$
$$\iff f \in L^1(\mathbb{B}) \cap L^1(\mathbb{P}). \tag{13}$$

Equation (12) follows from the first part, and (11) and (13) are applications of [53][Theorem 1.33]. This proves (a).

*Part 3 :* The proof for (b) follows similarly. For $p \in [1, \infty)$,

$$\phi \in L^p(\mathbb{R}^k; \mathbb{B} + \mathbb{P}) \iff \|\phi\|_2^p \in L^1(\mathbb{B} + \mathbb{P}) \tag{14}$$
$$\iff \|\phi\|_2^p \in L^1(\mathbb{B}) \cap L^1(\mathbb{P}) \tag{15}$$
$$\iff \phi \in L^p(\mathbb{R}^k; \mathbb{B}) \cap L^p(\mathbb{R}^k; \mathbb{P}). \tag{16}$$

Equation (15) follows from the first part, and (14) and (16) are applications of the definition of the Bochner norm.

*Part 4:* Finally we consider the case where $p = \infty$. Let $O(c) = \{\mathbf{x} \in \mathcal{X} \mid \|\phi(\mathbf{x})\|_2 \geq c\}$. Then, consider,

$$\|\phi\|_{L^\infty(\mathbb{B}+\mathbb{P})} = \inf \ \{c > 0 \mid (\mathbb{B} + \mathbb{P})(O(c)) = 0\}$$
$$= \inf \ \{c > 0 \mid \mathbb{B}(O(c)) = 0, \mathbb{P}(O(c)) = 0\}$$
$$= \max \ \left\{\inf \ \{c > 0 \mid \mathbb{B}(O(c)) = 0\}, \inf \ \{c > 0 \mid \mathbb{P}(O(c)) = 0\}\right\}$$
$$= \max \ \left\{\|\phi\|_{L^\infty(\mathbb{B})}, \|\phi\|_{L^\infty(\mathbb{P})}\right\}. \tag{17}$$

The steps are trivial. It is obvious from Equation (17) that $L^\infty(\mathbb{R}^k; \mathbb{B} + \mathbb{P}) = L^\infty(\mathbb{R}^k; \mathbb{B}) \cap L^\infty(\mathbb{R}^k; \mathbb{P})$. $\qquad\square$

### B.1.2 Atomless vector measures

A vector measure [54] over $(\mathcal{X} \times \mathcal{Y}, \Sigma_{\mathcal{X} \times \mathcal{Y}})$ is a set function that takes values in a Banach space (instead of $\mathbb{R}$).

**Definition B.1.** *[54, pp. 1] Let $(\mathcal{X} \times \mathcal{Y}, \Sigma_{\mathcal{X} \times \mathcal{Y}})$ be a measurable space and let $(\mathbb{V}, \|\cdot\|)$ be a Banach space. Then $G : \Sigma_{\mathcal{X} \times \mathcal{Y}} \to \mathbb{V}$ is a countably additive vector measure, if for all sequences of disjoint sets $\{E_i\}_{i=1}^\infty$, $G(\cup_{i=1}^\infty E_i) = \lim_{n \to \infty} \sum_{i=1}^n G(E_i)$.*

We are interested in atomless vector measures, which appear as an intermediate object in the proof of Proposition B.1.

**Definition B.2.** *Let $(\mathcal{X}, \Sigma_{\mathcal{X}})$ be a measurable space and $(\mathbb{V}, \|\cdot\|)$ be a Banach space. A vector measure $G : \Sigma_{\mathcal{X}} \to \mathbb{V}$ is called non-atomic or atomless iff for any $A \in \Sigma_{\mathcal{X}}$, such that $G(A) \neq 0$, there exists $B \in \Sigma_{\mathcal{X}}$ such that $B \subseteq A$ and $G(B) \notin \{0, G(A)\}$.*

We are interested in the following "closure" properties of atomless measures.

**Lemma B.2.** *If $\mathcal{L} : \mathcal{X} \to \mathbb{R}$ is integrable with respect to a measure space $(\mathcal{X}, \Sigma_{\mathcal{X}}, \mathbb{P})$, where $\mathbb{P}$ is an atomless measure, then the measure $\nu : \Sigma_{\mathcal{X}} \to \mathbb{R}$,*

$$\forall A \in \Sigma_{\mathcal{X}}, \quad \nu(A) = \int_A \mathcal{L}(\mathbf{x}) \, d\mathbb{P},$$

*is an atomless scalar measure.*

> **Lemma B.3.** *Let $N \in \mathbb{N}$. If $\nu_1, \ldots, \nu_N : \Sigma_{\mathcal{X}} \to \mathbb{R}$ are atomless scalar measures on a measurable space $(\mathcal{X}, \Sigma_{\mathcal{X}})$, then the vector measure $G : \Sigma_{\mathcal{X}} \to \mathbb{R}^N$ defined by,*
>
> $$A \in \Sigma_{\mathcal{X}}, \quad G(A) = [\nu_1(A), \ldots, \nu_N(A)]^{\top}$$
>
> *is a countably additive atomless vector measure.*

Before we prove Lemma B.2 we note a few properties of the indefinite integral $\nu(A) = \int_A \mathcal{L}(\mathbf{x}) \, d\mathbb{P}$.

**Remark B.1.** *Note that $\nu$ is indeed a measure over the measure space $(\mathcal{X}, \Sigma_{\mathcal{X}})$ [50, Section I.5, Indefinite integrals]. Moreover, $\nu$ is absolutely continuous with respect to $\mathbb{P}$, i.e. for any $A \in \Sigma_{\mathcal{X}}$,*

$$\mathbb{P}(A) = 0 \Rightarrow \nu(A) = 0. \tag{18}$$

*This can be inferred from the fact that if $A$ is a $\mathbb{P}$-negligible set (i.e. $\mathbb{P}(A) = 0$) then the indefinite integral $\int_A \mathcal{L}(\mathbf{x}) \, d\mathbb{P}$ is 0 for every function $\mathcal{L}$ [50, Proposition I.4.13].*

**Remark B.2.** *If $\nu$ is a finite signed measure on $\Sigma_{\mathcal{X}}$, then it admits a so-called Jordan Decomposition [53, Section 6.6],*

$$\forall A \in \Sigma_{\mathcal{X}}, \nu(A) = \nu_+(A) - \nu_-(A), \tag{19}$$

*where $\nu_+, \nu_-$ are non-negative measures. The Hahn decomposition theorem [53, Theorem 6.14] further states that there exists disjoint measurable sets $\mathcal{X}_+$ and $\mathcal{X}_-$ such that $\mathcal{X}_+ \cup \mathcal{X}_- = \mathcal{X}$, and for all $A \in \Sigma_{\mathcal{X}}$,*

$$\nu_+(A) = \nu(A \cap \mathcal{X}_+) \quad \text{and } \nu_-(A) = -\nu(A \cap \mathcal{X}_-). \tag{20}$$

*In particular, Equations (19) and (20) imply that the measure of any subset of $\mathcal{X}_+$ (resp. $\mathcal{X}_-$) with respect to $\nu$ is non-negative (resp. non-positive). We will also need the following set,*

$$\mathcal{X}_0 = \{ \mathbf{x} \in \mathcal{X} \mid \mathcal{L}(\mathbf{x}) = 0 \}.$$

*$\mathcal{X}_0$ is a measurable set since $\mathcal{X}_0 = \mathcal{L}^{-1}(0)$, and $0 \in \mathcal{B}(\mathbb{R})$, the Borel sigma algebra on $\mathbb{R}$, as it can be represented as a countable intersection of open sets (e.g. as $\cap_{i=1}^{\infty} \left( -\frac{1}{n}, \frac{1}{n} \right)$). It is obvious that $\nu(E) = 0$ for any $E \subseteq \mathcal{X}_0$.*

We will now prove Lemma B.2.

**Proof of Lemma B.2 .**

*Proof.* As noted in Remark B.1, $\nu$ is indeed a (signed) measure. Suppose that for an arbitrary $A \in \Sigma_{\mathcal{X}}, \nu(A) \neq 0$. Without loss of generality, assume that $\nu(A) > 0$, the negative case follows by symmetry. To prove that $\nu$ is atomless we need to prove the existence of a measurable set $B \subseteq A$ such that $\nu(B) \notin \{\nu(A), 0\}$.

Consider the Jordan decomposition of $\nu$,

$$\nu(A) = \nu_+(A) - \nu_-(A).$$

Since $\nu_-$ is a non-negative measure, and $\nu(A) > 0$, therefore $\nu_+(A) = \nu(A) + \nu_-(A) > 0$. Remark B.2 states that $\nu_+(A) = \nu(A \cap \mathcal{X}_+)$. Therefore $\nu(A \cap \mathcal{X}_+) > 0$. Clearly,

$$\nu(A \cap \mathcal{X}_+) = \nu(A \cap \mathcal{X}_+ \cap \mathcal{X}_0) + \nu(A \cap \mathcal{X}_+ \cap \mathcal{X}_0^c).$$

Clearly $\nu(A \cap \mathcal{X}_+ \cap \mathcal{X}_0) = 0$, therefore $\nu(A \cap \mathcal{X}_+ \cap \mathcal{X}_0^c) = \nu(A \cap \mathcal{X}_+) > 0$. Let $A_+ = A \cap \mathcal{X}_+ \cap \mathcal{X}_0^c$.

The absolute continuity of $\nu$ with respect to $\mathbb{P}$ implies that $\mathbb{P}(A_+) > 0$ since $\nu(A_+) > 0$. Since $\mathbb{P}$ is atomless, therefore there exists a set $B \subseteq A_+$ such that $\mathbb{P}(B) \notin \{\mathbb{P}(A_+), 0\}$. Clearly,

$$\mathbb{P}(A_+) = \mathbb{P}(B) + \mathbb{P}(B^c \cap A_+). \tag{21}$$

Since $\mathbb{P}$ is a non-negative measure therefore $\mathbb{P}(B) \geq 0$ and $\mathbb{P}(B^c \cap A_+) \geq 0$. Now since $\mathbb{P}(B) \notin \{\mathbb{P}(A_+), 0\}$, (21) implies that

$$0 < \mathbb{P}(B) < \mathbb{P}(A_+) \quad \text{and } 0 < \mathbb{P}(B^c \cap A_+) < \mathbb{P}(A_+). \tag{22}$$

Now, it is also true that,

$$\nu(A_+) = \nu(B) + \nu(B^c \cap A_+). \tag{23}$$

Let $\mathbb{1}_B(\mathbf{x})$ be the indicator function of the set $B$, taking the value 1 when $\mathbf{x} \in B$ and 0 otherwise. Since $B \subseteq \mathcal{X}_+$, therefore $\nu(B) \geq 0$. Now suppose, if possible, that $\nu(B) = \int_B \mathcal{L}(\mathbf{x}) \, d\mathbb{P} = 0$. Then according to [50, Proposition I.4.13], $\mathbb{1}_B(\mathbf{x})\mathcal{L}(\mathbf{x})$ is 0 almost everywhere with respect to $\mathbb{P}$. Since $\mathbb{1}_B(\mathbf{x}) = 1$ everywhere on $B$, this implies that $\mathcal{L}(\mathbf{x})$ must be 0 almost everywhere on $B$, i.e. except for a null subset, say $N$. Explicitly, $\mathcal{L}(\mathbf{x})$ is 0 on $B \setminus N$ and since $\mathbb{P}(B) > 0$, therefore $N \neq B$ and $B \setminus N \neq \emptyset$. However, since $\mathcal{X}_0$ is a superset of $B$, therefore $\mathcal{L}(\mathbf{x}) \neq 0$ everywhere on $B$, including $B \setminus N$, which is a contradiction. Therefore $\nu(B) > 0$.

Similarly, we can argue that $\nu(B^c \cap A_+) > 0$ since $\mathbb{P}(B^c \cap A_+) > 0$ and $B^c \cap A_+ \subseteq \mathcal{X}_+ \cap \mathcal{X}_0$. Therefore,

$$0 < \nu(B) < \nu(A_+).$$

Recall that we showed that $\nu(A_+) = \nu(A \cap \mathcal{X}_+) = \nu_+(A)$. Since $\nu_+(A) = \nu(A) + \nu_-(A)$ and $\nu_-$ is non-negative, therefore $\nu(A_+) = \nu_+(A) \geq \nu(A)$. If $\nu(B) \neq \nu(A)$, we are done since $B \subseteq A$ and $\nu(B) > 0$. Now suppose that $\nu(B) = \nu(A)$. Then since $\mathbb{P}(B) > 0$, there exists $B' \subseteq B$ such that $0 < \mathbb{P}(B') < \mathbb{P}(B)$. As before we can prove that $0 < \nu(B') < \nu(B)$ by utilising the fact that $\mathbb{P}(B') > 0$ and $B' \subseteq \mathcal{X}_+ \cap \mathcal{X}_0$. But this time, $\nu(B') < \nu(B) = \nu(A)$ by construction. Since $B' \subseteq A$ and $\nu(B') \notin \{0, \nu(A)\}$ therefore $\nu$ is atomless. $\qquad\square$

The proof for Lemma B.3 is almost trivial.

**Proof of Lemma B.3 .**

*Proof.* It is easy to verify that countable additivity is preserved by concatenating scalar measures, so we will verify that atomlessness is also preserved. Consider $A \in \Sigma_{\mathcal{X}}$ such that $G(A) \neq 0$. Therefore there exists an index $i$ such that $\nu_i(A) \neq 0$. Since $\nu_i$ is atomless, therefore there exists $B \subset A$ such that $\nu_i(B) \notin \{\nu_i(A), 0\}$. It follows that $G(B) \notin \{G(A), 0\}$ which proves the atomlessness of $G$. $\qquad\square$

### B.1.3 Constrained optimization problems

In this section we will define and discuss the main objects involved in Lagrangian duality in a unified manner with respect to a generic optimization problem $(P_0)$. For *this section*, let $l, g_i, h_j$ be functions from $\mathcal{X}$ to $\mathbb{R}$, then consider the following optimization problem,

$$
\begin{aligned}
P_0^\star = \inf_{x \in \mathcal{X}} \quad & \ell(x) \\
\text{subject to} \quad & g_i(x) \leq 0, \quad \text{for } i = 1, \dots, I \\
& h_j(x) = 0, \quad \text{for } j = 1, \dots, J.
\end{aligned}
\tag{$P_0$}
$$

Let $\mathrm{Feas}(P_0)$ refer to the subset of $\mathcal{X}$ that satisfies all the constraints, i.e. the set of feasible solutions. If $(P_0)$ is infeasible, we set $P_0^\star$ to $+\infty$. If $P_0^\star = -\infty$ then we say that $(P_0)$ is unbounded. If $\ell$ is bounded, then $(P_0)$ is also bounded (from below). Let $\mathrm{Opt}(P_0)$ refer to the subset of $\mathrm{Feas}(P_0)$ that achieves $P_0^\star$, i.e. the set of optimal solutions. Closely related to $(P_0)$ is its Lagrangian,

$$L_0(x, \lambda, \mu) = \ell(x) + \sum_{i=1}^{I} \lambda_i g_i(x) + \sum_{j=1}^{J} \mu_j h_j(x),$$

where $\lambda \in \mathbb{R}_+^I$ (non-negative) and $\mu \in \mathbb{R}^J$ are the dual variables, collecting $\lambda_i$ and $\mu_j$ respectively. The dual function of $(P_0)$ is $q_0(\lambda, \mu) = \inf_{x \in \mathcal{X}} L_0(x, \lambda, \mu)$. The dual function defines the dual problem,

$$D_0^\star = \sup_{\lambda_i \geq 0, \, \mu_j \in \mathbb{R}} q_0(\lambda, \mu) = \sup_{\lambda_i \geq 0, \, \mu_j \in \mathbb{R}} \min_{x \in \mathcal{X}} L_0(x, \lambda, \mu). \tag{$D_0$}$$

The optimal dual variables, when they exist, are denoted with a $\star$, such as $(\lambda^\star, \mu^\star)$. We denote the set of all optimizers of $(D_0)$ as $\mathrm{Opt}(D_0)$, which are also often called Lagrange multipliers. Let

$F_0 : \mathcal{X} \to \mathbb{R}^{1+I+J}$ be the vector function obtained by stacking the objective and the constraint functions, i.e.,

$$\forall x \in \mathcal{X} \qquad F_0(x) = [\ell(x), g_1(x), \dots g_I(x), h_1(x), \dots h_J(x)]^\top .$$

We call $F_0$ the cost constraint vector of $(\mathrm{P}_0)$, and the following set will be called the cost constraint epigraph of $(\mathrm{P}_0)$,

$$\mathcal{M}_0 = F_0(\mathcal{X}) + \mathbb{R}_+^{1+I} \times \{0\}^J .$$

Another important set is the projection of $\mathcal{M}_0$ on the constraint axes,

$$\begin{aligned}
\mathcal{C}_0 &= \left\{ (u, v) \in \mathbb{R}^{I+J} \mid (f, u, v) \in \mathcal{M}_0 \right\} \\
&= C_0(\mathcal{X}) + \mathbb{R}_+^I \times \{0\}^J .
\end{aligned} \tag{24}$$

where $C_0(x) = [g_1(x), \dots, g_I(x), h_1(x), \dots, h_J(x)]^\top$ is the vector function formed by stacking only the $I + J$ constraint functions. We call $C_0$ the constraint vector and $\mathcal{C}_0$ the constraint value epigraph. The relative interior of $\mathcal{C}_0$ is defined as,

$$\mathrm{relint}\,(\mathcal{C}_0) = \{ y \in \mathcal{C}_0 \mid \exists \epsilon > 0, \text{ s.t. }, B(y, \epsilon) \cap \mathrm{aff}\,(\mathcal{C}_0) \subseteq \mathcal{C}_0 \}$$

where $\mathrm{aff}\,(\mathcal{C}_0)$ is the affine hull of $\mathcal{C}_0$ and $B(y, \epsilon) = \{ z \mid \|y - z\| \leq \epsilon \}$ is an $\epsilon$ ball centered at $y \in \mathbb{R}^{I+J}$.

### B.1.4 Geometric conditions for strong duality

$(\mathrm{D}_0)$ is a relaxation of $(\mathrm{P}_0)$, i.e. $D_0^\star \leq P_0^\star$. This fact is called *weak duality* in convex optimization. When the equality holds, this fact is called strong duality and $(\mathrm{P}_0)$ is said to be strongly dual. Classical convex optimization provides us the following result tying the convexity of the cost constraint epigraph $\mathcal{M}_0$ and the strong duality of $(\mathrm{P}_0)$.

> **Theorem B.1.** *[15, Proposition 4.4.1 (variation)] Let $P_0^\star > -\infty$. If $0^{I+J} \in \text{relint}\,(\mathcal{C}_0)$ then $\mathrm{Opt}(\mathrm{D}_0) \neq \emptyset$. Moreover, if $\mathcal{M}_0$ is convex then $P_0^\star = D_0^\star$.*

Theorem B.1 underlies our functional strong duality result (Proposition B.1). It can be seen as a consequence of the fact that $\mathcal{M}_0$ provides a geometric embedding of $(\mathrm{P}_0)$, namely,

$$P_0^\star = \inf_{(f, 0^I, 0^J) \in \mathcal{M}_0} f. \tag{25}$$

Similarly, the convex closure of $\mathcal{M}_0$, denoted $\overline{\mathrm{conv}}\,(\mathcal{M}_0)$, provides a geometric embedding of the *dual* problem $(\mathrm{D}_0)$ when $(\mathrm{P}_0)$ is feasible ([15, Proposition 4.3.2]). Explicitly,

$$D_0^\star = \inf_{(f, 0^I, 0^J) \in \overline{\mathrm{conv}}(\mathcal{M}_0)} f. \tag{26}$$

Equations (25) and (26) suggest that the convexity and closure of $\mathcal{M}_0$ is closely related to strong duality. In fact, convexity and closure are together sufficient for strong duality [15, Proposition 4.3.2]. However, the requirement for closure can also be dropped when the target value of the constraints $(0^{I+J})$ is in the *relative interior* of the constraint set $\mathcal{C}_0$, as seen in Theorem B.1. Theorem B.1 also provides sufficient conditions (i.e. the non-extremality of the constraint levels) for the existence of Lagrange multipliers. This is non-trivial, since strong duality is not sufficient for the existence of Lagrange multipliers—[15, Example 5.3.3] provides an example where $\mathcal{M}_0$ is both convex and closed but $\mathrm{Opt}(\mathrm{D}_0) = \emptyset$.

If the dimension of $\mathcal{C}_0$ is $I + J$, then the set of Lagrange multipliers is bounded, and the converse is also true [15, Proposition 4.4.2]. The linear independence of the constraint functions is sufficient for the dimension of $\mathcal{C}_0$ to be $I + J$ (as we implicitly assume in Assumption 1). For problems with only inequalities it is sufficient for $C_0(\mathcal{X})$ to be non-empty for $\mathcal{C}_0$ to be of dimension $I + J$. In the next section we will analyse another perturbation of $(\mathrm{P}_0)$, one where we relax all the constraints by $\epsilon$.

### B.1.5 Constraint perturbations

In this section we investigate the following relaxation of ($P_0$),

$$
\begin{aligned}
P_\epsilon^\star = \inf_{x \in \mathcal{X}} \quad & \ell(x) \\
\text{subject to} \quad & g_i(x) \le \epsilon, && \text{for } i = 1, \dots, I, \\
& -\epsilon \le h_j(x) \le \epsilon, && \text{for } j = 1, \dots, J.
\end{aligned}
\tag{$P_\epsilon$}
$$

Note that ($P_\epsilon$) has $I + 2J$ inequality constraints. Its dual problem is as follows,

$$
D_\epsilon^\star = \sup_{\lambda_i, \mu_{+,j}, \mu_{-,j} \ge 0} \min_{x \in \mathcal{X}} L_\epsilon(x, \lambda, \mu_{+,j}, \mu_{-,j}),
\tag{$D_\epsilon$}
$$

where $L_\epsilon$ is the Lagrangian function of ($P_\epsilon$). While we know that $P_0^\star \ge P_\epsilon^\star$, the relationship is difficult to characterize further without additional assumptions. However, the relationship between the dual problems follows from the relationship between the Lagrangians.

> **Lemma B.4.** *Assume* $Opt(D_0), Opt(D_\epsilon) \ne \emptyset$. *If* $\gamma_0^\star = (\lambda_0^\star, \mu_0^\star) \in Opt(D_0)$ *and* $\gamma_\epsilon^\star = (\lambda_\epsilon^\star, \mu_{\epsilon,+}^\star, \mu_{\epsilon,-}^\star) \in Opt(D_\epsilon)$, *then it holds that,*
>
> $$
> \epsilon \|\gamma_\epsilon^\star\|_1 \le D_0^\star - D_\epsilon^\star \le \epsilon \|\gamma_0^\star\|_1.
> \tag{27}
> $$
>
> *Moreover, (27) implies that* $D_0^\star \ge D_\epsilon^\star$ *and* $\|\gamma_0^\star\|_1 \ge \|\gamma_\epsilon^\star\|_1$.

*Proof.* **Upper bound .** Let $\lambda \in \mathbb{R}_+^I$ and $\mu \in \mathbb{R}^J$ be arbitrarily chosen. Let $\mu_+$ and $\mu_-$ be defined pointwise as $\mu_{+,j} = \max\{0, \mu_j\}$ and $\mu_{-,j} = \max\{0, -\mu_j\}$. Clearly $\mu = \mu_+ - \mu_-$ and $\|\mu_+\|_1 + \|\mu_-\|_1 = \|\mu\|_1$. Applying these facts, we can verify that $L_0$ can be rewritten as follows,

$$
L_0(x, \lambda, \mu) = \left\{ \ell(x) + \sum_{i=1}^I \lambda_i(g_i(x) - \epsilon) + \sum_{j=1}^J \mu_{+,j}(h_j(x) - \epsilon) + \sum_{j=1}^J \mu_{-,j}(-h_j(x) - \epsilon) \right\} \\
+ \epsilon(\|\lambda\|_1 + \|\mu\|_1).
$$

It is clear that the first term is $L_\epsilon(\lambda, \mu_+, \mu_-)$, therefore,

$$
L_0(x, \lambda, \mu) = L_\epsilon(x, \lambda, \mu_+, \mu_-) + \epsilon(\|\lambda\|_1 + \|\mu\|_1).
\tag{28}
$$

Taking the infimum with respect to $x$ on both sides we obtain,

$$
q_0(\lambda, \mu) = q_\epsilon(\lambda, \mu_+, \mu_-) + \epsilon(\|\lambda\|_1 + \|\mu\|_1).
\tag{29}
$$

Let $\mu_{0+}^\star$ and $\mu_{0-}^\star$ be defined pointwise as $\mu_{0+,j}^\star = \max\{0, \mu_{0,j}^\star\}$ and $\mu_{0-,j}^\star = \max\{0, -\mu_{0,j}^\star\}$. Then, we have,

$$
\begin{aligned}
D_0^\star - D_\epsilon^\star &= q_0(\lambda_0^\star, \mu_0^\star) - q_\epsilon(\lambda_\epsilon^\star, \mu_{\epsilon,+}^\star, \mu_{\epsilon,-}^\star) \\
&\le q_0(\lambda_0^\star, \mu_0^\star) - q_\epsilon(\lambda_0^\star, \mu_{0,+}^\star, \mu_{0,-}^\star),
\end{aligned}
$$

since $(\lambda_0^\star, \mu_{0,+}^\star, \mu_{0,-}^\star)$ is not necessarily optimal with respect to ($D_\epsilon$). Applying (29) to the above inequality, we obtain,

$$
D_0^\star - D_\epsilon^\star \le \epsilon(\|\lambda_0^\star\|_1 + \|\mu_0^\star\|_1) \le \epsilon \|\gamma_0^\star\|_1,
$$

which proves the upper bound.

**Lower bound .** Now let us prove the lower bound. Let $\lambda \in \mathbb{R}_+^I$ and $\mu_+, \mu_- \in \mathbb{R}_+^J$ be arbitrarily chosen. Then we can rewrite $L_\epsilon$ as follows,

$$
\begin{aligned}
L_\epsilon(x, \lambda, \mu_+, \mu_-) &= \ell(x) + \sum_{i=1}^I \lambda_i(g_i(x) - \epsilon) + \sum_{j=1}^J \mu_{+,j}(h_j(x) - \epsilon) + \sum_{j=1}^J \mu_{-,j}(-h_j(x) - \epsilon) \\
&= L_0(x, \lambda, \mu_+ - \mu_-) - \epsilon(\|\lambda\|_1 + \|\mu_+\|_1 + \|\mu_-\|_1).
\end{aligned}
\tag{30}
$$

Again, taking the infimum with respect to $x$ gives us

$$q_\epsilon(\lambda, \mu_+, \mu_-) = q_0(\lambda, \mu_+ - \mu_-) - \epsilon(\|\lambda\|_1 + \|\mu_+\|_1 + \|\mu_-\|_1). \tag{31}$$

Next, we have,

$$\begin{aligned}
D_\epsilon^\star - D_0^\star &= q_\epsilon(\lambda_\epsilon^\star, \mu_{\epsilon,+}^\star, \mu_{\epsilon,-}^\star) - q_0(\lambda_0^\star, \mu_0^\star) \\
&\leq q_\epsilon(\lambda_\epsilon^\star, \mu_{\epsilon,+}^\star, \mu_{\epsilon,-}^\star) - q_0(\lambda_\epsilon^\star, \mu_{\epsilon,+}^\star - \mu_{\epsilon,-}^\star),
\end{aligned}$$

since $(\lambda_\epsilon^\star, \mu_{\epsilon,+}^\star - \mu_{\epsilon,-}^\star)$ is suboptimal for $q_0$. Applying (31) to the last inequality, we obtain,

$$D_\epsilon^\star - D_0^\star \leq -\epsilon(\|\lambda_\epsilon^\star\|_1 + \|\mu_{\epsilon,+}^\star\|_1 + \|\mu_{\epsilon,-}^\star\|_1). \tag{32}$$

Since $\|\gamma_\epsilon^\star\|_1 = (\|\lambda_\epsilon^\star\|_1 + \|\mu_{\epsilon,+}^\star\|_1 + \|\mu_{\epsilon,-}^\star\|_1)$, multiplying (32) by $-1$ completes the proof. $\quad\square$

## B.2 Functional strong duality

### B.2.1 Convexity of $\mathcal{M}_\phi$

In this section we are concerned with the range of the functional problem.

$$\begin{aligned}
P_\phi^\star = \inf_{\phi \in \Phi} \quad & \mathbb{E}_\mathbb{D}\big[\ell(\phi(\mathbf{x}), y)\big] \\
\text{subject to} \quad & \mathbb{E}_{\mathbb{P}_i}\big[g_i(\phi(\mathbf{x}), y)\big] \leq 0, \quad \text{for } i = 1, \ldots, I, \\
& \mathbb{E}_{\mathbb{Q}_j}\big[h_j(\phi(\mathbf{x}), y)\big] = 0, \quad \text{for } j = 1, \ldots, J.
\end{aligned} \tag{$P_\phi$}$$

The Lagrangian associated with ($P_\phi$) is the same as for (P) (see (1)) and ($D_\phi$) was defined in Section 3. Of particular interest in this section is the cost constraint function,

$$\begin{aligned}
F_\phi(\phi) = \big[ & \mathbb{E}_\mathbb{D}\left[\ell(\phi(\mathbf{x}), y)\right], \mathbb{E}_{\mathbb{P}_1}\left[g_1(\phi(\mathbf{x}), y)\right], \ldots, \mathbb{E}_{\mathbb{P}_I}\left[g_I(\phi(\mathbf{x}), y)\right], \\
& \mathbb{E}_{\mathbb{Q}_1}\left[h_1(\phi(\mathbf{x}), y)\right], \ldots, \mathbb{E}_{\mathbb{Q}_J}\left[h_J(\phi(\mathbf{x}), y)\right] \big]^\top.
\end{aligned}$$

We will also need the constraint epigraph and the cost-constraint epigraphs of $P_\phi$, namely,

$$\mathcal{C}_\phi = \left\{ (\mathbf{u}, \mathbf{v}) \in \mathbb{R}^I \times \mathbb{R}^J \,\middle|\, \begin{array}{l} \exists \phi \in \Phi \quad \text{s.t.} \quad \mathbb{E}_{\mathbb{P}_i}\left[g_i(\phi(\mathbf{x}), y)\right] \leq u_i, \quad \text{for } i = 1, \ldots, I, \\ \qquad\qquad\quad \text{and} \quad \mathbb{E}_{\mathbb{Q}_j}\left[h_j(\phi(\mathbf{x}), y)\right] = v_j \quad \text{for } j = 1, \ldots, J, \end{array} \right\}$$

$$\text{and } \mathcal{M}_\phi = \left\{ (f, \mathbf{u}, \mathbf{v}) \in \mathbb{R}^I \times \mathbb{R}^J \,\middle|\, \begin{array}{l} \exists \phi \in \Phi \quad \text{s.t.} \quad \mathbb{E}_\mathbb{D}\left[\ell(\phi(\mathbf{x}), y)\right] = f, \\ \qquad\qquad\qquad\quad \mathbb{E}_{\mathbb{P}_i}\left[g_i(\phi(\mathbf{x}), y)\right] \leq u_i, \quad \text{for } i = 1, \ldots, I, \\ \qquad\qquad\quad \text{and} \quad \mathbb{E}_{\mathbb{Q}_j}\left[h_j(\phi(\mathbf{x}), y)\right] = v_j \quad \text{for } j = 1, \ldots, J. \end{array} \right\}$$

**Remark B.3** (Well definedness of $P_\phi$)**.** *The expectations must exist for all $\phi \in \Phi$ for ($P_\phi$) to be well defined, i.e. we require $\Phi \subseteq \text{dom } F_\phi$. This is true, for example, when $\ell, g_i, h_j$ are bounded functions. We derive another condition later in Lemma B.7, but for now we assume $\Phi \subseteq \text{dom } F_\phi$ for our Lemmas.*

As noted in Section B.1.3, the convexity of $\mathcal{M}_\phi$ is sufficient for strong duality when 0 is in the interior of the augmented constraint set. We will prove that $\mathcal{M}_\phi$ is convex under certain conditions when $\Phi$ is a decomposable set. Let us now define decomposability.

**Definition B.3.** *A function class $\Phi = \{\phi : \mathcal{X} \to \mathbb{R}\}$ is said to be decomposable with respect to a measurable space $(\mathcal{X}, \Sigma_\mathcal{X})$ iff for every $\phi_1, \phi_2 \in \Phi$ and $Z \in \Sigma_\mathcal{X}$, there exists $\phi_3 \in \Phi$ such that,*

$$\phi_3(x) = \begin{cases} \phi_1(x) & x \in Z \\ \phi_2(x) & x \in Z^c. \end{cases}$$

We use Lyapunov's convexity theorem as a tool for the proof.

**Theorem B.2.** *([54] Chapter IX Corollary 5) Let $(\mathcal{X}, \Sigma_\mathcal{X})$ be a measurable space, and let $(\mathbb{V}, \|\cdot\|)$ be a finite dimensional Banach space. If $G : \Sigma_\mathcal{X} \to \mathbb{V}$ is a countably additive vector measure, then the range of $G$ is a compact and convex subset of $\mathbb{V}$.*

The range of $G$ is the set $\{G(A) \mid A \in \Sigma_\mathcal{X}\}$. We will now prove that $\mathcal{M}_\phi$ is convex.

> **Lemma B.5.** *Let $\Phi \subseteq dom\ F_\phi$. If the distributions $\mathbb{D}, \mathbb{P}_i, \mathbb{Q}_j$ are atomless and $\Phi$ is a decomposable set of functions, then it follows that $\mathcal{M}_\phi$ is convex.*

*Proof.* Consider the following set function formed by integrating $\ell(\phi(\mathbf{x}), y), g_i(\phi(\mathbf{x}), y), h_j(\phi(\mathbf{x}), y)$ with respect to $\mathbb{D}, \mathbb{P}_i, \mathbb{Q}_j$ respectively, over sets of the form $A \times \mathcal{Y}$, where $A \in \Sigma_\mathcal{X}$ ($\Sigma_\mathcal{X}$ is the marginal sigma algebra over $\mathcal{X}$) :

$$\forall A \in \Sigma_\mathcal{X}, G(A; \phi) = \left[ \int_A \int_\mathcal{Y} \ell(\phi(\mathbf{x}), y)\ d\mathbb{D}, \int_A \int_\mathcal{Y} g_1(\phi(\mathbf{x}), y)\ d\mathbb{P}_1, \dots, \int_A \int_\mathcal{Y} h_J(\phi(\mathbf{x}), y)\ d\mathbb{Q}_J \right]^\top.$$

Lemma B.2 implies that each entry of $G(A; \phi)$ is an atomless scalar measure, since $\mathbb{D}, \mathbb{P}_i, \mathbb{Q}_j$ are atomless. Suppose that $\phi_1, \phi_2 \in \Phi$. Define another set function, $\mathfrak{p}(A) : \Sigma_\mathcal{X} \to \mathbb{R}^{2(1+I+J)}$ given by

$$\forall A \in \Sigma_\mathcal{X}, \quad \mathfrak{p}(A) = \begin{bmatrix} G(A; \phi_1) \\ G(A; \phi_2) \end{bmatrix}.$$

As before, each of the entries of $\mathfrak{p}(A)$ are atomless scalar measures, therefore Lemma B.3 implies that $\mathfrak{p}(A)$ is an atomless *vector* measure. Therefore Lyapunov's convexity theorem (Theorem B.2) implies that the range of $\mathfrak{p}$, denoted $\mathfrak{p}(\Sigma_\mathcal{X})$, is convex. Therefore for any $\lambda \in [0, 1]$, there exists a set $\mathcal{T}_\lambda \subseteq \mathcal{X}$ such that,

$$\mathfrak{p}(\mathcal{T}_\lambda) = \lambda \mathfrak{p}(\emptyset) + (1 - \lambda)\mathfrak{p}(\mathcal{X}) = (1 - \lambda)\mathfrak{p}(\mathcal{X}), \tag{33}$$

where we applied the fact that $\mathfrak{p}(\emptyset) = 0$. Since for any finite measure it holds that $\mathfrak{p}(A) = \mathfrak{p}(\mathcal{X}) - \mathfrak{p}(A^c)$, therefore it follows that,

$$\mathfrak{p}(\mathcal{T}_\lambda^c) = \mathfrak{p}(\mathcal{X}) - \mathfrak{p}(\mathcal{T}_\lambda) = (1 - (1 - \lambda))\mathfrak{p}(\mathcal{X}) = \lambda \mathfrak{p}(\mathcal{X}). \tag{34}$$

We will use these relations to prove that the range of $F_\phi$, denoted $F_\phi(\Phi)$, is also convex. We do so by constructing an input $\phi_3$ such that $F_\phi(\phi_3)$ is a convex combination of $F_\phi(\phi_1)$ and $F_\phi(\phi_2)$. The required function $\phi_3$ is defined as follows,

$$\forall \mathbf{x} \in \mathcal{X}, \quad \phi_3(\mathbf{x}) = \begin{cases} \phi_1(\mathbf{x}) & \mathbf{x} \in \mathcal{T}_\lambda, \\ \phi_2(\mathbf{x}) & \mathbf{x} \in \mathcal{T}_\lambda^c. \end{cases}$$

By the definition of decomposability, $\phi_3$ is also an element of $\Phi$. Next we note that,

$$G(\mathcal{X}; \phi_3) = G(\mathcal{T}_\lambda; \phi_3) + G(\mathcal{T}_\lambda^c; \phi_3),$$

since $\mathcal{T}_\lambda, \mathcal{T}_\lambda^c$ form a mutually exclusive cover of $\mathcal{X}$. Therefore,

$$G(\mathcal{X}; \phi_3) = G(\mathcal{T}_\lambda; \phi_1) + G(\mathcal{T}_\lambda^c; \phi_2)$$

by applying the definition of $\phi_3$. Equations (33) and (34) applied to the above relation implies that,

$$G(\mathcal{X}; \phi_3) = \lambda G(\mathcal{X}; \phi_1) + (1 - \lambda)G(\mathcal{X}; \phi_2). \tag{35}$$

But clearly, $F_\phi(\phi) = G(\mathcal{X}; \phi)$ for any $\phi$. Therefore,

$$F_\phi(\phi_3) = \lambda F_\phi(\phi_1) + (1 - \lambda)F_\phi(\phi_2).$$

This proves that the range $F_\phi(\Phi)$ is convex, because $\phi_1, \phi_2$ were picked arbitrarily.

Recall that $\mathcal{M}_\phi = F_\phi(\Phi) + \mathbb{R}_+^{1+I} \times \mathbf{0}^J$. Clearly $\mathbb{R}_+^{1+I} \times \mathbf{0}^J$ is a convex set, and since the Minkowski sum of two convex sets is convex ([55] Theorem 3.1), therefore $\mathcal{M}_\phi$ is also convex, which concludes the proof. $\qquad \square$

### B.2.2 Strong duality of $(\mathrm{P}_\phi)$

In this section we will utilise Theorem B.1 and Lemma B.5 along with the assumptions we made in Section 3 to prove that $(\mathrm{P}_\phi)$ is strongly dual.

**Proposition B.1.** *Let (a) $\Phi \subseteq$ dom $F_\phi$, (b) $\Phi$ be decomposable and (c) $P_\phi^\star > -\infty$. Under Assumptions 1 and 2.1, $(P_\phi)$ is strongly dual and $Opt(D_\phi)$ is non-empty.*

*Proof.* The premise of the proposition, along with atomlessness from Assumption 2 allows us to invoke Lemma B.5 to conclude that $\mathcal{M}_\phi$ is a convex set.

From Assumption 1, we know that for some $\xi > 0$, $B(0^{I+J}, \xi) \subseteq$ int $(\mathcal{C})$ and $\mathcal{C} \subseteq \mathcal{C}_\phi$, therefore $B(0^{I+J}, \xi) \subseteq$ int $(\mathcal{C}) \subseteq$ int $(\mathcal{C}_\phi)$. In particular, $0 \in$ int $(\mathcal{C}_\phi)$, therefore, we can invoke Theorem B.1 to conclude that $Opt(D_\phi) \neq \emptyset$ and $P_\phi^\star = D_\phi^\star$. $\qquad\qquad\square$

## B.3 Duality gap

In this section we will bound the duality gap of (P). For this, we define the following relaxation,

$$
\begin{aligned}
P_{\theta\nu}^\star = \inf_{\theta \in \Theta} \quad & \mathbb{E}_\mathbb{D}\left[\ell(f_\theta(\mathbf{x}), y)\right] \\
\text{subject to} \quad & \mathbb{E}_{\mathbb{P}_i}\left[g_i(f_\theta(\mathbf{x}), y)\right] \leq L\nu, && \text{for } i = 1, \ldots, I, \qquad\qquad (P_{\theta\nu}) \\
& -L\nu \leq \mathbb{E}_{\mathbb{Q}_j}\left[h_j(f_\theta(\mathbf{x}), y)\right] \leq L\nu, && \text{for } j = 1, \ldots, J.
\end{aligned}
$$

Note that the set $\Theta_\nu$ (defined in Section 3) is nothing but the feasibility set of $(P_{\theta\nu})$, i.e. Feas$(P_{\theta\nu})$. Define the Lagrangian $L_{\theta\nu}(f_\theta, \gamma)$ of $(P_{\theta\nu})$ similarly to (1), where the dual variable $\gamma \in \mathbb{R}_+^{I+2J}$ now corresponds to $I + 2J$ inequality constraints. Let its dual value be $D_{\theta\nu}^\star$, defined similarly to $D^\star$ (Section 3).

We define its functional version,

$$
\begin{aligned}
P_{\phi\nu}^\star = \inf_{\phi \in \Phi} \quad & \mathbb{E}_\mathbb{D}\left[\ell(\phi(\mathbf{x}), y)\right] \\
\text{subject to} \quad & \mathbb{E}_{\mathbb{P}_i}\left[g_i(\phi(\mathbf{x}), y)\right] \leq L\nu, && \text{for } i = 1, \ldots, I, \qquad\qquad (P_{\phi\nu}) \\
& -L\nu \leq \mathbb{E}_{\mathbb{Q}_j}\left[h_j(\phi(\mathbf{x}), y)\right] \leq L\nu, && \text{for } j = 1, \ldots, J.
\end{aligned}
$$

Let its Lagrangian and dual value be denoted as $L_{\phi\nu}(\phi, \gamma)$ and $D_{\phi\nu}^\star$ respectively, where $\gamma \in \mathbb{R}_+^{I+2J}$. We will bound the duality gap of (P) based on (5) which uses the functional strong duality relation $P_\phi^\star = D_\phi^\star$, and which we further decompose as,

$$
\begin{aligned}
P^\star - D^\star &= P^\star - P_\phi^\star + D_\phi^\star - D^\star \\
&= (P^\star - P_{\theta\nu}^\star) + (P_{\theta\nu}^\star - P_\phi^\star) + (D_\phi^\star - D_{\phi\nu}^\star) + (D_{\phi\nu}^\star - D_{\theta\nu}^\star) \\
&\quad + (D_{\theta\nu}^\star - D^\star).
\end{aligned}
$$

Note that this decomposition contains the duality gap of $(P_{\theta\nu})$, i.e. we can alternatively decompose $P^\star - D^\star$ as,

$$
P^\star - D^\star = (P^\star - P_{\theta\nu}^\star) + (P_{\theta\nu}^\star - D_{\theta\nu}^\star) + (D_{\theta\nu}^\star - D^\star).
$$

First, we will prove a few helper lemmas (Sections B.3.1,B.3.2,B.3.3). Then we will bound the duality gap of $(P_{\theta\nu})$ in Section B.3.4, and we will bound the remaining terms, i.e., $P^\star - P_{\theta\nu}^\star$ and $D_{\theta\nu}^\star - D^\star$ in Section B.3.5 and summarise everything with a bound on the duality gap $P^\star - D^\star$ (Proposition B.2).

### B.3.1 Lipschitz continuity of $F_\phi$

First, we will prove that the mapping $F_\phi : L^p(\mathbb{R}^K; \mathbb{P}_+) \to \mathbb{R}^{1+I+J}$ is a Lipschitz continuous mapping.

**Lemma B.6.** *Let the loss functions $\ell, g_i, h_j$ be L-Lipschitz continuous. Then the components of the range function $F_\phi : L^p(\mathbb{R}^K; \mathbb{P}_+) \to \mathbb{R}^{1+I+J}$, e.g. $\phi \mapsto \mathbb{E}_\mathbb{D}\left[\ell(\phi(\mathbf{x}), y)\right]$, are L-Lipschitz continuous and $F_\phi$ is $(\sqrt{1 + I + J})$L-Lipschitz continuous, for $p \in [1, \infty)$.*

*Proof.* We will prove that $\phi \mapsto \mathbb{E}_\mathbb{D}\left[\ell(\phi(\mathbf{x}), y)\right]$ is Lipschitz continuous - the proof is the same for the other components. Let $\phi_1, \phi_2 \in L^p(\mathbb{R}^K; \mathbb{P}_+)$, then,

$$
\begin{aligned}
\left|\mathbb{E}_\mathbb{D}\left[\ell(\phi_1(\mathbf{x}), y)\right] - \mathbb{E}_\mathbb{D}\left[\ell(\phi_2(\mathbf{x}), y)\right]\right|^p &\leq \mathbb{E}_\mathbb{D}\left[\left|\ell(\phi_1(\mathbf{x}), y) - \ell(\phi_2(\mathbf{x}), y)\right|^p\right] \\
&\leq \mathbb{E}_\mathbb{D}\left[L^p \left\|\phi_1(\mathbf{x}) - \phi_2(\mathbf{x})\right\|_2^p\right] \\
&= L^p \mathbb{E}_\mathbb{D}\left[\left\|\phi_1(\mathbf{x}) - \phi_2(\mathbf{x})\right\|_2^p\right]. \quad (36)
\end{aligned}
$$

The first inequality is an application of Jensen's inequality ($|\cdot|^p$ is convex) and the second inequality applies the Lipschitz condition on $\ell(\cdot, \cdot)$. Equation (36) proves the continuity of $\phi \mapsto \mathbb{E}_\mathbb{D}\left[\ell(\phi(\mathbf{x}), y)\right]$ with respect to the norm on $L^p(\mathbb{R}^K; \mathbb{D})$, but we need to prove it for the norm on $L^p(\mathbb{R}^K; \mathbb{P}_+)$. Linearity of the integral with respect to the integral (Lemma B.1) implies that,

$$
\mathbb{E}_{\mathbb{P}_+}\left[\left\|\phi_1(\mathbf{x}) - \phi_2(\mathbf{x})\right\|_2^p\right] = \mathbb{E}_\mathbb{D}\left[\left\|\phi_1(\mathbf{x}) - \phi_2(\mathbf{x})\right\|_2^p\right] + \mathbb{E}_{\sum_{i=1}^I \mathbb{P}_i + \sum_{j=1}^J \mathbb{Q}_j}\left[\left\|\phi_1(\mathbf{x}) - \phi_2(\mathbf{x})\right\|_2^p\right].
$$

Clearly $\mathbb{E}_{\sum_{i=1}^I \mathbb{P}_i + \sum_{j=1}^J \mathbb{Q}_j}\left[\left\|\phi_1(\mathbf{x}) - \phi_2(\mathbf{x})\right\|_2^p\right] \geq 0$ since both measure and integrand are non-negative. This implies that,

$$
\mathbb{E}_\mathbb{D}\left[\left\|\phi_1(\mathbf{x}) - \phi_2(\mathbf{x})\right\|_2^p\right] \leq \mathbb{E}_{\mathbb{P}_+}\left[\left\|\phi_1(\mathbf{x}) - \phi_2(\mathbf{x})\right\|_2^p\right]. \quad (37)
$$

Equation (36) and (37) together imply that,

$$
\left|\mathbb{E}_\mathbb{D}\left[\ell(\phi_1(\mathbf{x}), y)\right] - \mathbb{E}_\mathbb{D}\left[\ell(\phi_2(\mathbf{x}), y)\right]\right|^p \leq L^p \mathbb{E}_{\mathbb{P}_+}\left[\left\|\phi_1(\mathbf{x}) - \phi_2(\mathbf{x})\right\|_2^p\right] = L^p \left\|\phi_1 - \phi_2\right\|_{L^p(\mathbb{R}^K; \mathbb{P}_+)}^p,
$$

which proves the first part (after taking the p$^{\text{th}}$ root on both sides). The second part is proved as follows.

$$
\begin{aligned}
\left\|F_\phi(\phi_1) - F_\phi(\phi_2)\right\|_2^2 &= \sum_{k=1}^{1+I+J} \left(F_{\phi,k}(\phi_1)_k - F_{\phi,k}(\phi_2)_k\right)^2 \\
&\leq (1 + I + J)L^2 \left\|\phi_1 - \phi_2\right\|_{L^p(\mathbb{R}^K; \mathbb{P}_+)}^2. \quad (38)
\end{aligned}
$$

Here we used the $L$-Lipschitz continuity of the components of $F_\phi(\phi)$ that was proved in the first part. Taking the square root on both sides proves that $F_\phi$ is $(\sqrt{1+I+J})L$-Lipschitz continuous with respect to the domain $L^p(\mathbb{R}^K; \mathbb{P}_+)$. $\qquad\square$

### B.3.2 Well definedness of $(\mathrm{P}_\phi)$

Continuity of the loss functions almost solves the problem of well-definedness of $(\mathrm{P}_\phi)$.

---

**Lemma B.7.** *Let the loss functions $\ell, g_i, h_j$ be $L$-Lipschitz continuous and $\Phi \subseteq L^p(\mathbb{R}^K; \mathbb{P}_+)$. If $\Phi \cap \operatorname{dom} F_\phi \neq \emptyset$, then $\Phi \subseteq \operatorname{dom} F_\phi$.*

---

*Proof.* Let $\phi_0 \in \Phi \cap \operatorname{dom} F_\phi$ and $\phi \in \Phi$. By the triangle inequality,

$$
\left\|F_\phi(\phi)\right\|_2 \leq \left\|F_\phi(\phi) - F_\phi(\phi_0)\right\|_2 + \left\|F_\phi(\phi_0)\right\|_2.
$$

Since $F_\phi : L^p(\mathbb{R}^K; \mathbb{P}_+) \to \mathbb{R}^K$ is $(\sqrt{1+I+J})L$-Lipschitz continuous (Lemma B.6), therefore,

$$
\left\|F_\phi(\phi)\right\|_2 \leq (\sqrt{1+I+J})L \left\|\phi_0 - \phi\right\|_{L^p(\mathbb{R}^K; \mathbb{P}_+)} + \left\|F_\phi(\phi_0)\right\|_2.
$$

Since $\phi_0 \in \operatorname{dom} F_\phi$, therefore $\left\|F_\phi(\phi_0)\right\|_2 < \infty$. Since $\phi, \phi_0 \in L^p(\mathbb{R}^K; \mathbb{P}_+)$, therefore the first term is also bounded. Therefore $\left\|F_\phi(\phi)\right\|_2 < \infty$ and $\Phi \subseteq \operatorname{dom} F_\phi$. $\qquad\square$

### B.3.3 Lipschitz continuity of parametrization

We will now show that pointwise Lipschitz continuity of the map $f_\theta(\mathbf{x})$ at each $\mathbf{x}$ implies the Lipschitz continuity of the map $\theta \mapsto f_\theta \in L^p(\mathbb{R}^K; \mathbb{P}_+)$.

**Lemma B.8.** *Let $f_\theta(\mathbf{x})$ be $L_\theta$-Lipschitz continuous for every $\mathbf{x} \in \mathcal{X}$. Then, the map $f^\Theta : \Theta \ni \theta \mapsto f_\theta \in L^p(\mathbb{R}^K; \mathbb{P}_+)$ is also $L_\theta$-Lipschitz continuous (for $p \in [1, \infty)$).*

*Proof.* Consider $\theta_1, \theta_2 \in \Theta$. Then,

$$\|f_{\theta_1} - f_{\theta_2}\|^p_{L^p(\mathbb{R}^K; \mathbb{P}_+)} = \int |f_{\theta_1}(\mathbf{x}) - f_{\theta_2}(\mathbf{x})|^p \ d\mathbb{P}_+$$

$$\leq \int L_\theta^p \|\theta_1 - \theta_2\|_2^p \ d\mathbb{P}_+ \tag{39}$$

$$\Rightarrow \|f_{\theta_1} - f_{\theta_2}\|^p_{L^p(\mathbb{R}^K; \mathbb{P}_+)} \leq L_\theta^p \|\theta_1 - \theta_2\|_2^p.$$

Equation (39) is the application of the pointwise Lipschitz continuity. Taking the $p^{th}$ root on both sides of the last inequality completes the proof. $\qquad\square$

### B.3.4  Duality gap of $(\mathrm{P}_{\theta\nu})$

Lemma B.4 provides an upper bound for $D_\phi^\star - D_{\phi\nu}^\star$. Next we show that $D_{\theta\nu}^\star \geq D_{\phi\nu}^\star$, and thereafter bound $P_{\theta\nu}^\star - P_\phi^\star$.

**Lemma B.9.** *If $\mathcal{H} \subseteq \Phi$ then $D_{\theta\nu}^\star \geq D_{\phi\nu}^\star$.*

*Proof.* Let $\gamma \in \mathbb{R}_+^{I+2J}$ denote the dual variable for $(\mathrm{P}_{\theta\nu})$ or $(\mathrm{P}_{\phi\nu})$. Clearly $L_{\theta\nu}(f_\theta, \gamma) = L_{\phi\nu}(f_\theta, \gamma)$. Therefore,

$$\inf_{\theta \in \Theta} L_{\theta\nu}(f_\theta, \gamma) = \inf_{f_\theta \in \mathcal{H}} L_{\phi\nu}(f_\theta, \gamma) \geq \inf_{\phi \in \Phi} L_{\phi\nu}(\phi, \gamma)$$

The supremum preserves the inequality, therefore,

$$D_{\theta\nu}^\star = \sup_{\gamma \in \mathbb{R}_+^{I+2J}} \inf_{\theta \in \Theta} L_{\theta\nu}(f_\theta, \gamma) \geq \sup_{\gamma \in \mathbb{R}_+^{I+2J}} \inf_{\phi \in \Phi} L_{\phi\nu}(\phi, \gamma) = D_{\phi\nu}^\star$$

which was to be proved. $\qquad\square$

**Lemma B.10.** *Under Assumption 2, $\mathrm{Feas}(\mathrm{P}_{\theta\nu})$ is non-empty and $P_{\theta\nu}^\star - P_\phi^\star \leq L\nu$.*

*Proof.* Let $\phi^\star \in \mathrm{Opt}(\mathrm{P}_\phi) \neq \emptyset$ be the solution defined by Assumption 2.2. Since $\phi^\star$ is feasible for $(\mathrm{P}_\phi)$, therefore it follows that,

$$\mathbb{E}_{\mathbb{P}_i} [g_i(\phi^\star(\mathbf{x}), y)] \leq 0 \quad \text{and} \quad \mathbb{E}_{\mathbb{Q}_j} [h_j(\phi^\star(\mathbf{x}), y)] = 0.$$

By Assumption 2.2, there exists $f_\theta \in \mathcal{H}$ such that $\|f_\theta - \phi^\star\|_{L^p(\mathbb{R}^K; \mathbb{P}_+)} \leq \nu$. Since the range function $F_\phi(\phi)$ is component-wise $L$-Lipschitz continuous (Lemma B.6), therefore we can upper bound the inequality constraint functions as follows,

$$\mathbb{E}_{\mathbb{P}_i} [g_i(f_\theta(\mathbf{x}), y)] \leq \mathbb{E}_{\mathbb{P}_i} [g_i(\phi^\star(\mathbf{x}), y)] + L \|f_\theta - \phi^\star\|_{L^p(\mathbb{R}^K; \mathbb{P}_+)} \leq 0 + L\nu. \tag{40}$$

Similarly the equality constraint functions can be bounded symmetrically as,

$$\left| \mathbb{E}_{\mathbb{Q}_j} [h_i(f_\theta(\mathbf{x}), y)] - \mathbb{E}_{\mathbb{Q}_j} [h_i(\phi^\star(\mathbf{x}), y)] \right| \leq L \|f_\theta - \phi^\star\|_{L^p(\mathbb{R}^K; \mathbb{P}_+)}$$

$$\Rightarrow \left| \mathbb{E}_{\mathbb{Q}_j} [h_i(f_\theta(\mathbf{x}), y)] \right| \leq L\nu. \tag{41}$$

Equations (40) and (41) imply that $\theta$ is feasible for $(\mathrm{P}_{\theta\nu})$. Therefore, $\mathrm{Feas}(\mathrm{P}_{\theta\nu}) \neq \emptyset$ and,

$$P_{\theta\nu}^\star \leq \mathbb{E}_{\mathbb{D}} [\ell(f_\theta(\mathbf{x}), y)]. \tag{42}$$

But the $L$-Lipschitzness of the objective (Assumption 3, Lemma B.6) implies that,

$$\mathbb{E}_{\mathbb{D}}\left[\ell(f_\theta(\mathbf{x}), y)\right] \leq \mathbb{E}_{\mathbb{D}}\left[\ell(\phi^\star(\mathbf{x}), y)\right] + L \left\| f_\theta - \phi^\star \right\|_{L^p(\mathbb{R}^K; \mathbb{P}_+)}$$
$$\leq P_\phi^\star + L\nu. \tag{43}$$

Therefore (42) and (43) implies,

$$P_{\theta\nu}^\star \leq P_\phi^\star + L\nu,$$

which was to be proved. $\qquad\square$

We will now summarize our arguments and bound the duality gap of ($\mathrm{P}_{\theta\nu}$) in the next lemma.

> **Lemma B.11.** *Under Assumptions 1,2 and 3, if $P_\phi^\star > -\infty$, $Opt(D_\phi)$ is non-empty and for any $\gamma_\phi^\star \in Opt(D_\phi)$, $P_{\theta\nu}^\star - D_{\theta\nu}^\star \leq (1 + \left\| \gamma_\phi^\star \right\|_1)L\nu$.*

*Proof.* We can decompose the duality gap as follows,

$$P_{\theta\nu}^\star - D_{\theta\nu}^\star = (P_{\theta\nu}^\star - P_\phi^\star) + (P_\phi^\star - D_\phi^\star) + (D_\phi^\star - D_{\phi\nu}^\star) + (D_{\phi\nu}^\star - D_{\theta\nu}^\star).$$

From Lemma B.10 (which requires Assumption 2), $P_{\theta\nu}^\star - P_\phi^\star \leq L\nu$.

Next, if $0 \in \mathrm{int}\,(\mathcal{C})$ (Assumption 1) then $0 \in \mathrm{int}\,(\mathcal{C}_\phi)$ since $\mathcal{C} \subseteq \mathcal{C}_\phi$. Then, by the construction of $\mathcal{C}_\phi$, $\mathrm{Feas}(\mathrm{P}_\phi) \neq \emptyset$. Clearly $\mathrm{Feas}(\mathrm{P}_\phi) \subseteq \mathrm{dom}\,F_\phi$, therefore $\Phi \cap \mathrm{dom}\,F_\phi \neq \emptyset$. Since we also have Assumption 3 (Lipschitz losses), we can apply Lemma B.7 to conclude $\Phi \subseteq \mathrm{dom}\,F_\phi$. The premise stipulates that $P_\phi^\star > -\infty$. Assumption 2 gives us atomlessness of the measures and decomposability of $\Phi$, which provides the remaining requirements to invoke Proposition B.1, giving us that $P_\phi^\star - D_\phi^\star = 0$ and $\mathrm{Opt}(\mathrm{D}_\phi) \neq \emptyset$.

Since $\mathrm{Opt}(\mathrm{D}_\phi) \neq \emptyset$, we can apply Lemma B.4 to conclude that $D_\phi^\star - D_{\phi\nu}^\star \leq L\nu \left\| \gamma_\phi^\star \right\|$ for some $\gamma_\phi^\star \in \mathrm{Opt}(\mathrm{D}_\phi)$. Lemma B.9, which only requires that $\mathcal{H} \subseteq \Phi$, proves that $D_{\phi\nu}^\star - D_{\theta\nu}^\star \leq 0$. Putting these bounds together gives us the required bound on the duality gap of ($\mathrm{P}_{\theta\nu}$). $\qquad\square$

**Remark B.4.** *This result is the same upper bound on the parametrized duality gap presented in [23]. This proof technique, specifically Lemma B.10, however, does not work with equality constrained problems such as (P). This is because replacing $\phi^\star$ with $\theta$ leads to a perturbation that localises the constraint function values upto an interval of size $2L\nu$. Therefore, no functional problem (or solution) generates a $\theta$ that satisfies an equality constraint, e.g. $\mathbb{E}_{\mathbb{Q}_j}\left[h_j(f_\theta(\mathbf{x}), y)\right] = 0$, since the constraint function value has to be exactly 0.*

*The limitation of this proof technique also applies to inequality constrained problems. Consider the following family of problems,*

$$P_I^\star = \inf_{\theta \in \Theta} \quad \mathbb{E}_{\mathbb{D}}\left[\ell(f_\theta(\mathbf{x}), y)\right]$$
$$\text{subject to} \quad \mathbb{E}_{\mathbb{P}_i}\left[g_i(f_\theta(\mathbf{x}), y)\right] \leq c_i, \quad \text{for } i = 1, \dots, I. \tag{$\mathrm{P}_I$}$$

*[23] used a tighter functional problem to upper bound $P_I^\star$, namely,*

$$P_{\phi I}^\star = \inf_{\phi \in \Phi} \quad \mathbb{E}_{\mathbb{D}}\left[\ell(\phi(\mathbf{x}), y)\right]$$
$$\text{subject to} \quad \mathbb{E}_{\mathbb{P}_i}\left[g_i(\phi(\mathbf{x}), y)\right] \leq c_i - L\nu, \quad \text{for } i = 1, \dots, I. \tag{$\mathrm{P}_{\phi I}$}$$

*Using Lemma B.10 with $J = 0$, we can prove that $P_I^\star - P_{\phi I}^\star \leq L\nu$ only if ($\mathrm{P}_{\phi I}$) is feasible. However, this may not be the case, if $c_i \leq \inf_{\phi \in \Phi} \mathbb{E}_{\mathbb{P}_i}\left[g_i(\phi(\mathbf{x}), y)\right] + L\nu$, i.e. $c_i$ is in the neighbourhood of the minimal achievable value. In this case the analysis in the next section still applies.*

### B.3.5 Duality gap of (P)

In this section we will bound the duality gap of (P). Since $D^\star \geq D^\star_{\theta\nu}$ (see Lemma B.4), therefore,

$$P^\star - D^\star = (P^\star - P^\star_{\theta\nu}) + (P^\star_{\theta\nu} - D^\star_{\theta\nu}) + (D^\star_{\theta\nu} - D^\star) \leq (P^\star - P^\star_{\theta\nu}) + (P^\star_{\theta\nu} - D^\star_{\theta\nu}). \quad (44)$$

$P^\star_{\theta\nu} - D^\star_{\theta\nu}$ is upper bounded by Lemma B.11. We now upper bound $P^\star - P^\star_{\theta\nu}$.

---

**Lemma B.12.** *Let $Feas(P) \neq \emptyset$. Under Assumption 3, it follows that $P^\star - P^\star_{\theta\nu} \leq LL_\theta R(\nu)$.*

---

*Proof.* Let $\epsilon > 0$ be chosen arbitrarily and let $\theta^\star_\nu$ be a *feasible* solution for $(P_{\theta\nu})$ that is $\epsilon$-optimal, i.e.,

$$P^\star_{\theta\nu} \leq \mathbb{E}_\mathbb{D} \left[ \ell(f_{\theta^\star_\nu}(\mathbf{x}), y) \right] \leq P^\star_{\theta\nu} + \epsilon. \quad (45)$$

Such a solution always exists if $(P_{\theta\nu})$ is feasible, since the optimal value is the infimum of the objective $\mathbb{E}_\mathbb{D} \left[ \ell(f_{\theta^\star_\nu}(\mathbf{x}), y) \right]$ over the feasible set. Note that since (P) is assumed feasible, so is $(P_{\theta\nu})$.

Next for arbitrary $\delta > 0$, let $\theta_0 \in \mathrm{Feas}(P)$ be such that $\|\theta^\star_\nu - \theta_0\| = \inf_{\theta \in \mathrm{Feas}(P)} \|\theta^\star_\nu - \theta\| + \delta$, i.e. $\theta_0$ is $\delta$-close to the projection of $\theta^\star_\nu$ onto the feasible set of (P) (which may not be closed). Note that $\Theta_\nu$ is nothing but $\mathrm{Feas}(P_{\theta\nu})$. Therefore, by the definition of $R(\nu)$,

$$\|\theta^\star_\nu - \theta_0\| \leq R(\nu) + \delta. \quad (46)$$

Next we apply the $L$-Lipschitzness of the map $\phi \mapsto \mathbb{E}_\mathbb{D} \left[ \ell(f_\theta(\mathbf{x}), y) \right]$ (Lemma B.6), and the $L_\theta$-Lipschitzness of the map $\theta \mapsto f_\theta$ (Lemma B.8) to conclude that,

$$\mathbb{E}_\mathbb{D} \left[ \ell(f_{\theta_0}(\mathbf{x}), y) \right] - \mathbb{E}_\mathbb{D} \left[ \ell(f_{\theta^\star_\nu}(\mathbf{x}), y) \right] \leq L \left\| f_{\theta_0} - f_{\theta^\star_\nu} \right\|_{L^P(\mathbb{R}^K; \mathbb{P}_+)} \leq LL_\theta \left\| \theta_0 - \theta^\star_\nu \right\|. \quad (47)$$

Since $\theta_0$ is feasible with respect to (P), therefore $\mathbb{E}_\mathbb{D} \left[ \ell(f_{\theta_0}(\mathbf{x}), y) \right] \geq P^\star$. This and (45) implies that,

$$P^\star - P^\star_{\theta\nu} \leq \mathbb{E}_\mathbb{D} \left[ \ell(f_{\theta_0}(\mathbf{x}), y) \right] - \mathbb{E}_\mathbb{D} \left[ \ell(f_{\theta^\star_\nu}(\mathbf{x}), y) \right] + \epsilon.$$

Thereafter we can apply (46) and (47) to conclude that,

$$P^\star - P^\star_{\theta\nu} \leq LL_\theta R(\nu) + LL_\theta \delta + \epsilon.$$

Since the bound holds for every $\epsilon > 0$ and $\delta > 0$, therefore it also holds for $\epsilon = 0$ and $\delta = 0$ (simply take the infimum on both sides). This completes the proof. $\square$

We will now state the bound on the duality gap for (P).

---

**Proposition B.2.** *Let Assumptions 1, 2 and 3 hold and let $P^\star_\phi > -\infty$. Then $Opt(D_\phi) \neq \emptyset$ and for any $\gamma^\star_\phi \in Opt(D_\phi)$, $P^\star - D^\star \leq (1 + \left\| \gamma^\star_\phi \right\|_1) L\nu + LL_\theta R(\nu)$.*

---

*Proof.* We can invoke Lemma B.11 with the premise to obtain that $Opt(D_\phi) \neq \emptyset$ and $P^\star_{\theta\nu} - D^\star_{\theta\nu} \leq (1 + \left\| \gamma^\star_\phi \right\|_1) L\nu$ for any $\gamma^\star_\phi \in Opt(D_\phi)$. Note that Assumption 1 implies that (P) is feasible due to the construction of $\mathcal{C}$, and this, along with Assumption 3, allows us to invoke Lemma B.12 which states that $P^\star - P^\star_{\theta\nu} \leq LL_\theta R(\nu)$. Applying these bounds to (44) completes the proof. $\square$

### B.4 Dual estimation error

In this section we will upper bound the absolute difference between the optimal values of the dual problem (D) and the empirical dual problem $(\hat{D})$, i.e. $|D^\star - \hat{D}^\star|$, beginning with a technical lemma that helps take the conjunction of probabilistic statements.

**Lemma B.13.** *Let $(\mathcal{X}, \Sigma_\mathcal{X}, \mathbb{P})$ be a measure space and let $\{a_1, \ldots, a_K\}$ be random statements. If $a_k$ holds with probability at least $1 - \delta$, for $k = 1, \ldots, K$, then $\bigwedge_{k=1}^K a_k$, that is the conjuction of $a_k$, holds with probability at least $1 - K\delta$.*

*Proof.* Let $\neg a_k$ denote the negation of $a_k$. Then,

$$\mathbb{P}(\bigwedge_{k=1}^{K} a_k) = 1 - \mathbb{P}(\bigvee_{k=1}^{K} \neg a_k) \geq 1 - \sum_{k=1}^{K} \mathbb{P}(\neg a_k). \tag{48}$$

The last inequality is the union bound ([56] Lemma 2.2). Since $\mathbb{P}(a_k) \geq 1 - \delta$, therefore $\mathbb{P}(\neg a_k) \leq \delta$. Applying this to (48) gives us the bound,

$$\mathbb{P}(\bigwedge_{k=1}^{K} a_k) \geq 1 - K\delta,$$

which completes the proof. $\qquad\square$

---

**Proposition B.3.** *Let $N_{min} = \min\{M_0, M_1 \ldots, M_I, N_1, \ldots, N_J\}$. If Assumptions 1 and 4 hold, then $Opt(D), Opt(\hat{D}) \neq \emptyset$ and, for all $\delta \in (0, 1)$,*

$$|D^\star - \hat{D}^\star| \leq \left(1 + \max\left\{\|\gamma^\star\|_1, \|\hat{\gamma}^\star\|_1\right\}\right) \zeta^{UC}(N_{min}, \delta),$$

*holds with probability at least $1 - (1 + I + J)\delta$ for any $\gamma^\star \in Opt(D), \hat{\gamma}^\star \in Opt(\hat{D})$.*

---

*Proof.* Assumption 1 states that $B(0, \xi) \subseteq \text{int } \mathcal{C} \cap \text{int } \hat{\mathcal{C}}$ and therefore $0 \in \text{int } \mathcal{C}$ and $0 \in \text{int } \hat{\mathcal{C}}$. Therefore Theorem B.1 implies that $\text{Opt}(D) \neq \emptyset$ and $\text{Opt}(\hat{D}) \neq \emptyset$.

The empirical dual function $\hat{q}(\lambda, \mu) = \inf_{\theta \in \Theta} \hat{L}(f_\theta, \lambda, \mu)$ was defined with ($\hat{D}$), similarly define the dual function of (P) as $q(\lambda, \mu) = \inf_{\theta \in \Theta} L(f_\theta, \lambda, \mu)$. Let $\gamma^\star = (\lambda^\star, \mu^\star)$ and $\hat{\gamma}^\star = (\hat{\lambda}^\star, \hat{\mu}^\star)$ be members of $\text{Opt}(D)$ and $\text{Opt}(\hat{D})$ respectively. Therefore,

$$\begin{aligned} D^\star - \hat{D}^\star &= q(\lambda^\star, \mu^\star) - \hat{q}(\hat{\lambda}^\star, \hat{\mu}^\star) \\ &\leq q(\lambda^\star, \mu^\star) - \hat{q}(\lambda^\star, \mu^\star), \end{aligned} \tag{49}$$

where the inequality follows from the suboptimality of $(\lambda^\star, \mu^\star)$ with respect to ($\hat{D}$), i.e. because $\hat{q}(\lambda^\star, \mu^\star) \leq \hat{q}(\hat{\lambda}^\star, \hat{\mu}^\star) = \hat{D}^\star$. Let $\epsilon > 0$ be arbitrary and let $\theta$ be $\epsilon$-optimal with respect to $\hat{q}(\lambda^\star, \mu^\star)$, i.e,

$$\hat{q}(\lambda^\star, \mu^\star) \leq \hat{L}(f_\theta, \lambda^\star, \mu^\star) \leq \hat{q}(\lambda^\star, \mu^\star) + \epsilon. \tag{50}$$

Clearly, $q(\lambda^\star, \mu^\star) \leq L(f_\theta, \lambda^\star, \mu^\star)$ since $q(\lambda^\star, \mu^\star) = \inf_\theta L(f_\theta, \lambda^\star, \mu^\star)$. This fact and (50), applied to (49), implies that,

$$D^\star - \hat{D}^\star \leq q(\lambda^\star, \mu^\star) - \hat{q}(\lambda^\star, \mu^\star) \leq L(f_\theta, \lambda^\star, \mu^\star) - \hat{L}(f_\theta, \lambda^\star, \mu^\star) + \epsilon. \tag{51}$$

Now let us expand the difference of Lagrangians in the RHS of (51),

$$\begin{aligned} L(f_\theta, \lambda^\star, \mu^\star) - \hat{L}(f_\theta, \lambda^\star, \mu^\star) &= \left(\mathbb{E}_{\mathbb{D}}\left[\ell(f_\theta(\mathbf{x}), y)\right] - \frac{1}{M_0}\sum_{m_0=1}^{M_0} \ell\left(f_\theta(\mathbf{x}_{m_0}), y_{m_0}\right)\right) \\ &+ \sum_{i=1}^{I} \lambda_i^\star \left(\mathbb{E}_{\mathbb{P}_i}\left[g_i(f_\theta(\mathbf{x}), y)\right] - \frac{1}{M_i}\sum_{m_i=1}^{M_i} g_i\left(f_\theta(\mathbf{x}_{m_i}), y_{m_i}\right)\right) \\ &+ \sum_{j=1}^{J} \mu_j^\star \left(\mathbb{E}_{\mathbb{Q}_j}\left[h_j(f_\theta(\mathbf{x}), y)\right] - \frac{1}{N_j}\sum_{n_j=1}^{N_j} h_j\left(f_\theta(\mathbf{x}_{n_j}), y_{n_j}\right)\right). \end{aligned} \tag{52}$$

Now, since Assumption 4 holds, therefore the following holds,

$$\left|\mathbb{E}_{\mathbb{D}}\left[\ell(f_\theta(\mathbf{x}), y)\right] - \frac{1}{M_0}\sum_{m_0=1}^{M_0} \ell\left(f_\theta(\mathbf{x}_{m_0}), y_{m_0}\right)\right| \leq \zeta^{\text{UC}}(M_0, \delta), \tag{53}$$

for *every* $\theta \in \Theta$ with probability at least $1 - \delta$, in particular the $\theta$ chosen. We apply the uniform convergence bound on the remaining samples corresponding to $\mathbb{P}_i, \mathbb{Q}_j$ to obtain inequalities analogous to (53). All of the bounds hold simultaneously with probability $1 - (1 + I + J)\delta$ (see Lemma B.13). Applying all the bounds to (52), we obtain,

$$L(f_\theta, \lambda^\star, \mu^\star) - \hat{L}(f_\theta, \lambda^\star, \mu^\star) \leq \zeta^{\text{UC}}(M_0, \delta) + \sum_{i=1}^{I} \lambda_i^\star \zeta^{\text{UC}}(M_i, \delta) + \sum_{j=1}^{J} \mu_j^\star \zeta^{\text{UC}}(N_j, \delta).$$

Since $\zeta^{\text{UC}}(N, \delta)$ is decreasing with respect to $N$, therefore,

$$L(f_\theta, \lambda^\star, \mu^\star) - \hat{L}(f_\theta, \lambda^\star, \mu^\star) \leq (1 + \|\lambda^\star\|_1 + \|\mu^\star\|_1)\zeta^{\text{UC}}(N_{\min}, \delta). \tag{54}$$

Chaining (51) and (54) and noticing that $\|\gamma^\star\|_1 = \|\lambda^\star\|_1 + \|\mu^\star\|_1$, we obtain that $D^\star - \hat{D}^\star \leq (1 + \|\gamma^\star\|_1)\zeta^{\text{UC}}(N_{\min}, \delta) + \epsilon$. Notice that we can eliminate $\epsilon$ since none of the other terms depend on $\epsilon$, and so we can take the infimum on both sides. Therefore,

$$D^\star - \hat{D}^\star \leq (1 + \|\gamma^\star\|_1)\zeta^{\text{UC}}(N, \delta), \tag{55}$$

and this holds with probability greater than $1 - (1 + I + J)\delta$, since it is a consequence of (53) and its analogues for $\mathbb{P}_i, \mathbb{Q}_j$.

We can make a symmetric argument, switching the roles of $\gamma^\star$ and $\hat{\gamma}^\star$, to prove that

$$\hat{D}^\star - D^\star \leq (1 + \|\hat{\gamma}^\star\|_1)\zeta^{\text{UC}}(N_{\min}, \delta). \tag{56}$$

This time we pick some $\theta$ that is $\epsilon$-optimal with respect to $q(\hat{\lambda}^\star, \hat{\mu}^\star)$ (instead of $\hat{q}(\lambda^\star, \mu^\star)$). Since (53) holds for any $\theta$ (with probability greater than $1 - (1 + I + J)\delta$) therefore Equations (55) and (56) hold simultaneously with probability greater than the same bound, $1 - (1 + I + J)\delta$. Therefore, they together imply that $|D^\star - \hat{D}^\star| \leq (1 + \max\{\|\gamma^\star\|_1, \|\hat{\gamma}^\star\|_1\})\zeta^{\text{UC}}(N_{\min}, \delta)$, which completes the proof. $\quad\square$

## B.5 Proof of Theorem 3.1

**Theorem 3.1.** *Let $N_{min} = \min\{M_0, M_1 \ldots, M_I, N_1, \ldots, N_J\}$ and assume that $R(\nu) < \infty$ and $P_\phi^\star > -\infty$. Under Assumptions 1-3 there exist $\gamma_\phi^\star \in Opt(D_\phi), \gamma^\star \in Opt(D)$, and $\hat{\gamma}^\star \in Opt(\hat{D})$. Moreover, for any $\delta \in (0, 1)$, it holds with probability at least $1 - (1 + I + J)\delta$, that*

$$|P^\star - \hat{D}^\star| \leq \left(1 + \left\|\gamma_\phi^\star\right\|_1\right) L\nu + LL_\theta R(\nu) + \left(1 + \max\{\|\gamma^\star\|_1, \|\hat{\gamma}^\star\|_1\}\right) \zeta^{UC}(N_{min}, \delta). \tag{8}$$

*Proof.* Note that the requirements for Proposition B.2 and B.3 are included in the premise. The triangle inequality implies that,

$$|P^\star - \hat{D}^\star| \leq |P^\star - D^\star| + |D^\star - \hat{D}^\star|.$$

Proposition B.2 asserts that $\text{Opt}(D_\phi) \neq \emptyset$, and provides an upper bound for $|P^\star - D^\star| = P^\star - D^\star$, which always holds. Proposition B.3 asserts that $\text{Opt}(D) \neq \emptyset, \text{Opt}(\hat{D}) \neq \emptyset$, and provides an upper bound for $|D^\star - \hat{D}^\star|$ that holds with probability greater than $1 - (1 + I + J)\delta$.

Adding both bounds gives us the necessary bound on $|P^\star - \hat{D}^\star|$, which holds in probability greater than $1 - (1 + I + J)\delta$. $\quad\square$

We next discuss technical distinctions between Theorem 3.1 and [23, Theorem 1] (which pertains to problems where $J = 0$) that were omitted in the main text (see also Remark 3.1).

**Remark B.5.** *Firstly, we lift the assumption that $\Theta$ is compact. Secondly, we prove the case for regression problems ($\mathcal{Y}$ continuous) without additional assumptions (see [23, Assumption 6]). This is due to a (small) modification of the argument for Lemma B.5, which yields both results together. Note that [57] also provided a unified proof for functional strong duality, however, their proof takes a slightly different approach using the weak Lyapunov theorem [54, Chapter IX Theorem 10].*

*Finally, note that while both bounds feature the norm of the Lagrange multiplier of a functional problem, the Lagrange multiplier in [23] depends on $\nu$, while ours does not. This is because we have chosen to treat equalities and inequalities symmetrically (see Remark B.4 for a justification). Therefore, both $\|\gamma_\phi^\star\|_1$ and $R(\nu)$ depend on both inequalities and equalities. However, the proof can be repeated with a slightly different definition of $R(\nu)$ to obtain two separate "sensitivity" terms for inequalities and equalities, with the former term depending on a Lagrange multiplier like [23], and the latter term depending on the modified $R(\nu)$.*

In the next remark we expand upon the sensitivity interpretation of $\|\gamma^\star\|_1$ when (P) is non-convex and not strongly dual, as is generally the case in our setting.

**Remark B.6.** *If (P) is strongly dual, then the perturbation function $P^\star(\mathbf{c}, \mathbf{d})$ is convex (see [58, Section 5.6.1]). In fact, its epigraph is nothing but the set we called the "cost-constraint epigraph", corresponding to (P), i.e.,*

$$
\mathcal{M} = \left\{ (f, \mathbf{u}, \mathbf{v}) \in \mathbb{R}^{1+I+J} \;\middle|\; \begin{array}{llll} \exists \theta \in \Theta & s.t. & \mathbb{E}_{\mathbb{D}}\left[\ell(f_\theta(\mathbf{x}), y)\right] = f, \\ & & \mathbb{E}_{\mathbb{P}_i}\left[g_i(f_\theta(\mathbf{x}), y)\right] \leq u_i, & for\ i = 1, \ldots, I, \\ & and & \mathbb{E}_{\mathbb{Q}_j}\left[h_j(f_\theta(\mathbf{x}), y)\right] = v_j & for\ j = 1, \ldots, J \end{array} \right\}.
$$

*If the perturbation function were additionally differentiable, then (9) holds. More generally, without differentiability, $-(\lambda^\star, \mu^\star)$ is a subgradient of $P^\star(\mathbf{c}, \mathbf{d})$ at $(0^I, 0^J)$ (see [15, Example 5.4.2]), i.e,*

$$
P^\star(\mathbf{c}, \mathbf{d}) - P^\star \geq -\lambda^{\star\top}(\mathbf{c} - 0^I) - \mu^{\star\top}(\mathbf{d} - 0^J). \tag{57}
$$

*We now derive an approximate version of (57) from Proposition B.2 as follows :*

$$
P^\star(\mathbf{c}, \mathbf{d}) - P^\star \geq D^\star(\mathbf{c}, \mathbf{d}) - P^\star \tag{58}
$$

$$
\geq D^\star(\mathbf{c}, \mathbf{d}) - D^\star - \left[(1 + \|\gamma_\phi^\star\|_1)L\nu + LL_\theta R(\nu)\right] \tag{59}
$$

*Here we have used the weak duality relation for $P^\star(\mathbf{c}, \mathbf{d})$ in (58) and Proposition (B.2) in (59). We can bound the gap $D^\star(\mathbf{c}, \mathbf{d}) - D^\star$ in an identical manner as Lemma (B.4) to obtain,*

$$
D^\star(\mathbf{c}, \mathbf{d}) - D^\star \geq -\lambda^{\star\top}(\mathbf{c} - 0^I) - \mu^{\star\top}(\mathbf{d} - 0^J). \tag{60}
$$

*Combining (59) and (60), we obtain an approximate subgradient relation similar to (57). Explicitly,*

$$
P^\star(\mathbf{c}, \mathbf{d}) - P^\star \geq -\lambda^{\star\top}\mathbf{c} - \mu^{\star\top}\mathbf{d} + G(\nu), \tag{61}
$$

*where $G(\nu) = -\left[(1 + \|\gamma_\phi^\star\|_1)L\nu + LL_\theta R(\nu)\right]$ is the remainder term that disappears as $\nu \to 0$. Equation 61 thus enables us to interpret $\lambda^\star$ as constraint sensitivity even in the non-convex case, albeit approximately.*

## C   Proof of Theorem 4.1

In this section we will provide guarantees for Algorithm 1, which approximately solves $(\hat{D})$. Recall the quantities related to the empirical dual problem, i.e. $\hat{L}(f_\theta, \lambda, \mu), \hat{q}(\lambda, \mu)$ and $\hat{D}^\star$ (defined in Section 2). Also define the empirical constraint vector,

$$
\hat{C}(\theta) = \left[ \frac{1}{M_1} \sum_{m_1=1}^{M_1} g_1\big(f_\theta(\mathbf{x}_{m_1}), y_{m_1}\big), \ldots, \frac{1}{M_I} \sum_{m_I=1}^{M_I} g_I\big(f_\theta(\mathbf{x}_{m_I}), y_{m_I}\big), \right.
$$
$$
\left. \frac{1}{N_1} \sum_{n_1=1}^{N_1} h_1\big(f_\theta(\mathbf{x}_{n_1}), y_{n_1}\big), \ldots, \frac{1}{N_J} \sum_{n_J=1}^{N_J} h_J\big(f_\theta(\mathbf{x}_{n_J}), y_{n_J}\big) \right]^\top .
$$

Note that $\hat{C}$ has $I + J$ entries. Let $\gamma$ denote the tuple $(\lambda, \mu)$ in the sequel, and similarly for $\hat{\gamma}^\star, \gamma^{(t-1)}$ etc. We will denote $\hat{q}(\lambda, \mu)$ as $\hat{q}(\gamma)$ and $\hat{L}(f_\theta, \lambda, \mu)$ as $\hat{L}(f_\theta, \gamma)$ for brevity.

We can prove that the set of supergradients for $\hat{q}(\gamma)$ at $\gamma$ is equal to the convex hull $\mathrm{conv}\left( \left\{ \hat{C}(\theta) \mid \theta \in \Theta, \hat{L}(f_\theta, \gamma) = \hat{q}(\gamma) \right\} \right)$ (e.g. with [59, Theorem 2.87]). We will now show that *approximate minimizers* produce *approximate supergradients*.

**Lemma C.1.** *Let $\theta^{(t)}$ satisfy $\hat{q}(\gamma^{(t)}) \leq \hat{L}(f_\theta, \gamma^{(t)}) \leq \hat{q}(\gamma^{(t)}) + \rho$. Then, it follows that, for all $\gamma' \in \mathbb{R}_+^I \times \mathbb{R}^J$*

$$
\hat{q}(\gamma^{(t)}) \geq \hat{q}(\gamma') + \sum_{k=1}^{I+J} (\gamma_k^{(t)} - \gamma_k') \hat{C}_k(\theta^{(t)}) - \rho.
$$

*Proof.* From the premise of the lemma we have that,
$$
\hat{q}(\gamma^{(t)}) \geq \hat{L}(f_{\theta^{(t)}}, \gamma^{(t)}) - \rho. \tag{62}
$$
By the definition of the dual function, we have for any $\gamma' \in \mathbb{R}_+^I \times \mathbb{R}^J$,
$$
\hat{q}(\gamma') \leq \hat{L}(f_{\theta^{(t)}}, \gamma'), \tag{63}
$$
since $\hat{q}(\gamma') = \inf_{\theta \in \Theta} \hat{L}(f_\theta, \gamma')$. Then,
$$
(62) - (63) \Rightarrow \hat{q}(\gamma^{(t)}) - \hat{q}(\gamma') \geq \hat{L}(f_{\theta^{(t)}}, \gamma^{(t)}) - \hat{L}(f_{\theta^{(t)}}, \gamma') - \rho.
$$
If we expand $\hat{L}(f_{\theta^{(t)}}, \gamma^{(t)})$ and $\hat{L}(f_{\theta^{(t)}}, \gamma')$, then the terms in the Lagrangians correspond to the objective cancel out. We are left with the product of the constraint functions $(\hat{C}_1(\theta^{(t)}), \ldots, \hat{C}_{I+J}(\theta^{(t)}))$ and the difference in dual variables $\gamma^{(t)}, \gamma'$, i.e. $\sum_{k=1}^{I+J} (\gamma_k^{(t)} - \gamma_k') \hat{C}_k(\theta^{(t)})$. Bringing $\hat{q}(\gamma')$ to the RHS completes the proof. $\qquad \square$

Subgradient ascent algorithms that use *any* subderivative are not guaranteed to descend monotonically in the objective, in this case the dual function $\hat{q}$. This is also true of approximate subgradient descent. Using Lemma C.1 we can prove a recurrence which helps prove descent with respect to the *mean squared error* in the dual variable, which we define as ,
$$
U_t = \inf_{\hat{\gamma}^\star \in \mathrm{Opt}(\hat{D})} \left\| \gamma^{(t)} - \hat{\gamma}^\star \right\|_2^2, \tag{64}
$$
where $\mathrm{Opt}(\hat{D})$ is the set of optimal dual variables for $(\hat{D})$. We will also need a technical lemma first.

**Lemma C.2.** *Define $[\cdot]_+ : \mathbb{R}^m \to \mathbb{R}_+^m$ such that $[\mathbf{x}]_+ = \max(\mathbf{x}, \mathbf{0})$, where the maximum is taken componentwise. Let $\mathbf{x}, \mathbf{y} \in \mathbb{R}^d$, then it follows that,*
$$
\|[\mathbf{x}]_+ - [\mathbf{y}]_+\|_2 \leq \|\mathbf{x} - \mathbf{y}\|_2 .
$$

*Proof.* Since $\|[\mathbf{x}]_+ - [\mathbf{y}]_+\|_2^2 = \sum_{i=1}^m (\max(x_i, 0) - \max(y_i, 0))^2$, therefore we only need to prove,
$$
|\max(x_i, 0) - \max(y_i, 0)| \leq |x_i - y_i| . \tag{65}
$$
If $x_i > 0, y_i > 0$, both sides are equal. If $x_i < 0, y_i < 0$, the LHS evaluates to 0 so (65) holds true again. If $x_i < 0, y_i > 0$, the LHS is equal to $|y_i|$ and the RHS is equal to $|x_i| + |y_i|$ (trivial) so (65) holds true again. The last case follows by symmetry. This proves (65), which proves the lemma. $\quad \square$

**Lemma C.3.** *Let* $\theta^{(t)}, \gamma^{(t)} = (\lambda^{(t)}, \mu^{(t)})$ *be the* $t^{th}$ *iterate of Algorithm 1. If Assumptions 1 and 5 hold, and the loss functions* $\ell, g_i, h_j$ *are B-bounded, it follows that,*

$$U_t \leq U_{t-1} + 2\eta \left\{ \hat{q}(\gamma^{(t-1)}) - \hat{D}^\star + \rho \right\} + \eta^2 (I + J) B^2.$$

*Proof.* Let $\hat{C}_I(\theta) = \left[ \hat{C}_1(\theta), \ldots, \hat{C}_I(\theta) \right]^\top$ and $\hat{C}_J(\theta) = \left[ \hat{C}_{I+1}(\theta), \ldots, \hat{C}_{I+J}(\theta) \right]^\top$ denote the concatenated empirical constraint functions for the inequalities and equalities respectively. Clearly the constraint vector $\hat{C}(\theta)$ is the concatenation of both. Since Assumption 1 holds and the loss functions are bounded, therefore $\mathrm{Opt}(D_\phi) \neq \emptyset$ (Theorem B.1).

Decompose $\gamma^{(t)} = (\lambda^{(t)}, \mu^{(t)})$, then it holds that,

$$\left\| \gamma^{(t)} - \hat{\gamma}^\star \right\|_2^2 = \left\| \lambda^{(t)} - \hat{\lambda}^\star \right\|_2^2 + \left\| \mu^{(t)} - \hat{\mu}^\star \right\|_2^2. \tag{66}$$

Using the update rules for the dual variables, we obtain,

$$(66) \Rightarrow \left\| \gamma^{(t)} - \hat{\gamma}^\star \right\|_2^2 = \left\| \left[ \lambda^{(t-1)} + \eta \hat{C}_I(\theta^{(t-1)}) \right]_+ - \hat{\lambda}^\star \right\|_2^2 + \left\| \mu^{(t-1)} + \eta \hat{C}_J(\theta^{(t-1)}) - \hat{\mu}^\star \right\|_2^2. \tag{67}$$

where $[\mathbf{x}]_+$ applies $\max(\cdot, 0)$ to each component of $\mathbf{x}$. We can write $\hat{\lambda}^\star = \left[ \hat{\lambda}^\star \right]_+$ since $\hat{\lambda}^\star$ is non-negative by definition. By Lemma C.2 we know that,

$$\left\| \left[ \lambda^{(t-1)} + \eta \hat{C}_I(\theta^{(t-1)}) \right]_+ - \left[ \hat{\lambda}^\star \right]_+ \right\|_2 \leq \left\| \lambda^{(t-1)} + \eta \hat{C}_I(\theta^{(t-1)}) - \hat{\lambda}^\star \right\|_2. \tag{68}$$

Applying (68) to (67), we obtain,

$$\left\| \gamma^{(t)} - \hat{\gamma}^\star \right\|_2^2 \leq \left\| \lambda^{(t-1)} + \eta \hat{C}_I(\theta^{(t-1)}) - \hat{\lambda}^\star \right\|_2^2 + \left\| \mu^{(t-1)} + \eta \hat{C}_J(\theta^{(t-1)}) - \hat{\mu}^\star \right\|_2^2$$
$$= \left\| \gamma^{(t-1)} + \eta \hat{C}(\theta^{(t-1)}) - \hat{\gamma}^\star \right\|_2^2, \tag{69}$$

where in the last step we have recombined inequalities and equalities, since the rest of the proof is not affected by the difference. Expanding the squared norm on the RHS of (69), we obtain,

$$\left\| \gamma^{(t)} - \hat{\gamma}^\star \right\|_2^2 \leq \left\| \gamma^{(t-1)} - \hat{\gamma}^\star \right\|_2^2 + 2\eta (\gamma^{(t-1)} - \hat{\gamma}^\star)^\top \hat{C}(\theta^{(t-1)}) + \eta^2 \left\| \hat{C}(\theta^{(t-1)}) \right\|_2^2. \tag{70}$$

Since Assumption 5 holds, therefore we can apply Lemma C.1, which implies that

$$(\gamma^{(t-1)} - \hat{\gamma}^\star)^\top \hat{C}(\theta^{(t-1)}) \leq \hat{q}(\gamma^{(t-1)}) - \hat{q}(\hat{\gamma}^\star) + \rho = \hat{q}(\gamma^{(t-1)}) - \hat{D}^\star + \rho. \tag{71}$$

And from the boundedness of the loss functions we have that,

$$\left\| \hat{C}(\theta^{(t-1)}) \right\|_2^2 \leq (I + J) B^2. \tag{72}$$

Applying (71) and (72) to (70) we obtain,

$$\left\| \gamma^{(t)} - \hat{\gamma}^\star \right\|_2^2 \leq \left\| \gamma^{(t-1)} - \hat{\gamma}^\star \right\|_2^2 + 2\eta (\hat{q}(\gamma^{(t-1)}) - \hat{D}^\star + \rho) + \eta^2 (I + J) B^2.$$

Applying the operation $\inf_{\hat{\gamma}^\star \in \mathrm{Opt}(\hat{D})}$ to both sides preserves the inequality. Notice that only the LHS and the *first term* in the RHS depends on $\hat{\gamma}^\star$. Taking the other terms out of the infimum and applying the definition of $U_t$ gives us the required bound. $\qquad\square$

Although we cannot directly show that the objective $\hat{q}(\gamma^{(t)})$ monotonically increases, we can show that by picking the learning rate appropriately, the initial phase of the algorithm monotonically reduces the mean squared error ($U_t$) of the dual iterates, upto the point where $\hat{q}(\gamma^{(t)})$ is some $\alpha\rho$ close to $\hat{D}^\star$, for some $\alpha > 1$.

**Lemma C.4.** *Let $\theta^{(t)}, \gamma^{(t)} = (\lambda^{(t)}, \mu^{(t)})$ be the $t^{th}$ iterate of Algorithm 1. Let Assumptions 1 and 5 hold, and let the loss functions $\ell, g_i, h_j$ be B-bounded. Define $T_0 = \min \left\{ t \in \mathbb{N} \mid \hat{D}^{\star} - \hat{q}(\gamma^{(t)}) < \alpha\rho \right\} > 0$ for some $\alpha > 1$ and let $\eta \leq \frac{\alpha\rho}{(I+J)B^2}$. Then it holds for all $t \leq T_0$ that*

$$U_t < U_{t-1}.$$

*Moreover, $T_0 \leq T_{max} = \frac{U_0}{2\eta\rho}$.*

*Proof.* Lemma C.3 proves the following for arbitrary $t$,

$$U_t \leq U_{t-1} + 2\eta \left\{ \hat{q}(\gamma^{(t-1)}) - \hat{D}^{\star} + \rho \right\} + \eta^2 (I + J)B^2. \tag{73}$$

Assume $t \leq T_0$, then the premise states that $\hat{D}^{\star} - \hat{q}(\gamma^{(t-1)}) > \alpha\rho$, i.e. $\hat{q}(\gamma^{(t-1)}) - \hat{D}^{\star} + \rho < (1-\alpha)\rho$. Moreover $\eta \leq \frac{\alpha\rho}{(I+J)B^2} \Rightarrow \eta(I+J)B^2 \leq \alpha\rho$. Applying these bounds to (73) gives us,

$$U_t < U_{t-1} - 2(\alpha - 1)\eta\rho + 2\eta\alpha\rho$$
$$\Rightarrow U_t < U_{t-1} - 2\eta\rho. \tag{74}$$

Equation (74) proves that $U_t < U_{t-1}$ since $-2\eta\rho < 0$.

Now, to achieve an upper bound for $T_0$, apply (74) recursively to obtain,

$$U_{T_0} < U_0 - 2\eta\rho T_0. \tag{75}$$

Since $U_{T_0} \geq 0$, therefore (75) implies that,

$$U_0 - 2\eta\beta T_0 > 0 \Rightarrow T_0 < \frac{U_0}{2\eta\rho}.$$

$\square$

Note that the proof requires that $\alpha > 1$ to show descent. Once $\hat{q}(\gamma^{(t-1)}) - \hat{D}^{\star} \leq \rho$, (73) cannot prove descent because both terms are non-negative. Note the dependence of $T_{\max}$ on $\eta$ and $\rho$. While interpreting $T_0$ and $T_{\max}$, it is important to note that we cannot detect $T_0$ since we have neither $\hat{D}^{\star}$ nor $\hat{q}$ (since our oracle is approximate). However, we can compute $T_{\max}$, which is simply an upper bound on $T_0$. Proposition C.4 does not tell us what happens between $T_0$ and $T_{\max}$, in particular it does not tell us if our iterates become worse, either with respect to the mean squared error or dual function. While giving a guarantee for the *last iterate* (at $T = T_{\max}$) is challenging, we can give a guarantee for the average of the Lagrangian iterates. The following can also be written as a randomized result, such as in [23].

**Theorem 4.1.** *Suppose Assumptions 1 and 5 hold and let the loss functions $\ell, g_i, h_j$ be B-bounded. Let $U_0 = \inf_{\gamma^{\star} \in Opt(\hat{D})} \left\| \gamma^{(0)} - \gamma^{\star} \right\|$. Then, for any $T \in \mathbb{N}$, it holds that,*

$$\left| \hat{D}^{\star} - \frac{1}{T} \sum_{t=0}^{T-1} \left( \hat{L}(f_{\theta^{(t)}}, \gamma^{(t)}) \right) \right| \leq \rho + \frac{U_0}{2\eta T} + \frac{1}{2}(I + J)\eta B^2. \tag{10}$$

*If $\eta \leq \frac{\rho}{(I+J)B^2}$ and $T \geq \frac{U_0}{\eta\rho}$, then the bound is equal to $2\rho$.*

*Proof.* We know, by the definition of the oracle and the order of updates that, for any $t$,

$$\hat{q}(\gamma^{(t)}) \leq \hat{L}(f_{\theta^{(t)}}, \gamma^{(t)}) \leq \hat{q}(\gamma^{(t)}) + \rho. \tag{76}$$

Since $\hat{q}(\gamma^{(t)}) \leq \hat{D}^{\star}$, therefore (76) implies that $\hat{L}(f_{\theta^{(t)}}, \gamma^{(t)}) - \hat{D}^{\star} \leq \rho$ for all $t$, therefore

$$\frac{1}{T} \sum_{t=0}^{T-1} \hat{L}(f_{\theta^{(t)}}, \gamma^{(t)}) - \hat{D}^{\star} \leq \rho, \tag{77}$$

which proves one part of the bound, since $\left(\frac{U_0}{2\eta T} + \frac{1}{2}(I+J)\eta B^2\right)$ is non-negative. On the other hand, since $\hat{q}(\gamma^{(t)}) \leq \hat{L}(f_{\theta^{(t)}}, \gamma^{(t)})$, therefore,

$$\frac{1}{T}\sum_{t=0}^{T-1}\left(\hat{L}(f_{\theta^{(t)}}, \gamma^{(t)}) - \hat{D}^\star\right) \geq \frac{1}{T}\sum_{t=0}^{T-1}\left(\hat{q}(\gamma^{(t)}) - \hat{D}^\star\right). \tag{78}$$

We will now derive an upper bound for $\sum_{t=0}^{T-1}\left(\hat{q}(\gamma^{(t)}) - \hat{D}^\star\right)$ by applying Lemma C.3 recursively $T$ times to obtain,

$$U_T \leq U_0 + 2\eta\sum_{t=0}^{T-1}\left(\hat{q}(\gamma^{(t)}) - \hat{D}^\star\right) + T(2\eta\rho + \eta^2(I+J)B^2). \tag{79}$$

Now, since $U_T \geq 0$, dividing by $2\eta T$ and rearranging the terms, we get the following inequality,

$$\frac{1}{T}\sum_{t=0}^{T-1}\left(\hat{q}(\gamma^{(t)}) - \hat{D}^\star\right) \geq -\frac{U_0}{2\eta T} - \rho - \frac{1}{2}(I+J)\eta B^2.$$

Multiply the above equation by $-1$ and applying (78) we obtain the required upper bound,

$$\hat{D}^\star - \frac{1}{T}\sum_{t=0}^{T-1}\hat{L}(f_{\theta^{(t)}}, \gamma^{(t)}) \leq \rho + \frac{U_0}{2\eta T} + \frac{1}{2}(I+J)\eta B^2. \tag{80}$$

Now since $\eta \leq \frac{\rho}{(I+J)B^2} \Rightarrow \frac{1}{2}\eta(I+J)B^2 \leq \frac{1}{2}\rho$, and $T \geq \frac{U_0}{\eta\rho} \Rightarrow \frac{U_0}{2\eta T} \leq \frac{\rho}{2}$, therefore, (80) reduces to,

$$\hat{D}^\star - \frac{1}{T}\sum_{t=0}^{T-1}\hat{L}(f_{\theta^{(t)}}, \gamma^{(t)}) \leq 2\rho,$$

which proves the remainder. $\qquad\square$

# D  Additional theory

## D.1  Example : minimum norm interpolation

Let $X \in \mathbb{R}^{n \times n}$ and $w, y \in \mathbb{R}^n$. Consider the following problem, that finds the minimum norm solution to a system of linear equalities,

$$P_e^\star = \inf_{w \in \mathbb{R}^n} \quad \frac{1}{2} \|w\|_2^2 \qquad \qquad \text{(P}_e\text{)}$$
$$\text{subject to} \quad Xw = y.$$

The problem (P$_e$) can be transformed to a problem with a single inequality constraint, by constraining the aggregated violation of the equalities, namely,

$$P_i^\star = \inf_{w \in \mathbb{R}^n} \quad \frac{1}{2} \|w\|_2^2 \qquad \qquad \text{(P}_i\text{)}$$
$$\text{subject to} \quad \frac{1}{2} \|Xw - y\|_2^2 \leq \epsilon.$$

The feasibility and optimality sets of (P$_i$) coincide with (P$_e$) when $\epsilon = 0$. However, the dual problems of both problems are different. The dual problem for (P$_e$) does not depend on $\epsilon$, and is well posed. However, the dual solution of (P$_i$) diverges as $\epsilon \to 0$ and the dual problem of (P$_i$) becomes ill-posed. Note that this also implies that recovering a good approximation of the solution to (P$_e$) using an unconstrained objective $\|w\|_2^2 + \lambda \|Xw - y\|_2^2$ would need a very large weight $\lambda$.

We will now state and prove these facts. We need the following standard identities concerning the derivatives of matrix forms (which can be found e.g. in [60] Equations 69 and 81).

**Lemma D.1.** *Let $A \in \mathbb{R}^{n \times n}$ be a symmetric matrix and $x, b \in \mathbb{R}^n$. Then, the following identities hold true.*

  1. $\nabla_x (x^\top A x) = 2Ax$.

  2. $\nabla_x (x^\top b) = \nabla_x (b^\top x) = b$.

We make the following simplifying assumptions.

**Assumption 6.** *Assume that $XX^\top$ is a full rank matrix and there exists $w_0 \in \mathbb{R}^n$ such that $Xw_0 = y$ and $w_0 \neq 0$.*

We will now characterize the dual solution of (P$_i$).

> **Proposition D.1.** *Let Assumption 6 hold. Then there always exists a dual solution of* (P$_i$)*, say $\lambda^\star \geq 0$. Moreover, as $\epsilon \to 0$, $\lambda^\star \to \infty$.*

*Proof.* Consider the constraint function,

$$g(w) = \frac{1}{2} \|Xw - y\|_2^2$$
$$= \frac{1}{2} [Xw - y]^\top [Xw - y]$$
$$= \frac{1}{2} \left( w^\top X^\top X w - 2w^\top X^\top y + \|y\|_2^2 \right).$$

**Existence of dual solution and strong duality.**   It is clear that $g(w)$ is quadratic in $w$. Since its second derivative is the gram matrix $XX^\top$ (Lemma D.1) which is positive semi-definite [61, Theorem 4.6.6.], therefore $g(w)$ is convex [55, Theorem 4.5]. The existence of $w_0$ such that $Xw_0 = y$ further implies that a strictly feasible point ($w_0$) exists. Since the objective is also convex, therefore the set of dual solutions is non-empty and strong duality holds [15, Proposition 5.3.1].

**Existence of primal solution .** Since Feas($P_i$) is non-empty, and the objective is bounded from below by 0, therefore $-\infty < P_i^\star < \infty$. Therefore the following modification of ($P_i$) is well defined, for any $\gamma > 0$,

$$P_{i2}^\star = \inf_{w \in \mathbb{R}^n} \quad \frac{1}{2} \|w\|_2^2$$

$$\text{subject to} \quad \frac{1}{2} \|Xw - y\|_2^2 \le \epsilon \qquad\qquad (\text{P}_{i2})$$

$$\frac{1}{2} \|w\|_2^2 \le P_i^\star + \gamma.$$

Clearly Feas($P_{i2}$) $\subseteq$ Feas($P_i$), therefore $P_i^\star \le P_{i2}^\star$. Note that since $P_i^\star = \inf_{w \in \text{Feas}(P_i)} \frac{1}{2}\|w\|_2^2$, therefore for *any* $\delta > 0$, there exists $w \in$ Feas($P_i$) such that,

$$\frac{1}{2} \|w\|_2^2 \le P_i^\star + \delta. \qquad\qquad (81)$$

Let $\{\delta_j\}_{j=1}^\infty \subseteq \mathbb{R}$ be the sequence $\delta_j = \frac{\gamma}{j}$. Using (81) by setting $\delta = \delta_j$, we can construct a sequence $\{w_j\}$ that satisfies,

$$w_j \in \text{Feas}(P_i) \text{ and } \frac{1}{2}\|w_j\|_2^2 \le P_i^\star + \delta_j.$$

Clearly $\{w_j\} \subseteq$ Feas($P_{i2}$) since $\delta_1 = \gamma$ and $\delta_j$ is strictly decreasing. Since $\delta_j \to 0$, therefore $\frac{1}{2}\|w_j\|_2^2 \to P_i^\star$. Therefore $P_{i2}^\star = P_i^\star$.

Now, since $g(w)$ is convex, and clearly continuous, therefore its sub-level set $g(w) \le \epsilon$ is closed [55, Theorem 7.1]. Clearly, the set $\left\{ w \mid \frac{1}{2}\|w\|_2^2 \le P_i^\star + \delta \right\}$ is nothing but the norm ball $B(0, \sqrt{2(P_i^\star + \delta)})$, and is closed and compact. Therefore Feas($P_{i2}$) is compact. Setting $\delta = \gamma$ in (81) proves that Feas($P_{i2}$) is non-empty as well. Since the objective, $\frac{1}{2}\|w\|_2^2$ is continuous, the compact non-emptiness of the feasibility set implies that Opt($P_{i2}$) $\ne \emptyset$ (see [52, Theorem 4.1.6]).

Clearly, since Opt($P_{i2}$) $\subseteq$ Feas($P_i$) and $P_{i2}^\star = P_i^\star$, therefore Opt($P_{i2}$) $\subseteq$ Opt($P_i$). Therefore Opt($P_i$) $\ne \emptyset$.

**Diverging dual solution.** Now that we have established that the primal and dual solutions exist for all $\epsilon > 0$, let $\lambda_e^\star, w_e^\star$ be a primal dual solution pair. Since strong duality holds, therefore they must satisfy the KKT conditions (see [58][Section 5.5.3]).

The Lagrangian function associated with ($P_i$) is as follows,

$$L(w^\star, \lambda) = \frac{1}{2}\|w^\star\|_2^2 + \lambda^\star \left( \frac{1}{2}\|Xw^\star - y\|_2^2 - \epsilon \right)$$

$$= \frac{1}{2}w^{\star\top}w^\star + \frac{\lambda^\star}{2}\left( w^{\star\top}X^\top X w^\star - 2w^{\star\top}X^\top y + \|y\|_2^2 - 2\epsilon \right)$$

$$= \frac{1}{2}w^{\star\top}\left[ I + \lambda^\star X^\top X \right]w^\star - \lambda^\star w^{\star\top}X^\top y + \frac{\lambda^\star}{2}\left( \|y\|_2^2 - 2\epsilon \right).$$

The KKT condition for the stationarity of the Lagrangian implies that,

$$\nabla_{w^\star} L(w^\star, \lambda^\star) = 0$$

$$\Rightarrow \nabla_{w^\star}\left( \frac{1}{2}w^{\star\top}\left[ I + \lambda^\star X^\top X \right]w^\star - \lambda^\star w^{\star\top}X^\top y \right) = 0$$

$$\Rightarrow \left[ I + \lambda^\star X^\top X \right]w^\star - \lambda^\star X^\top y = 0$$

$$\Rightarrow w^\star = \lambda^\star \left[ I + \lambda^\star X^\top X \right]^{-1} X^\top y$$

$$= \left[ cI + X^\top X \right]^{-1} X^\top y$$

where we have defined $c = \frac{1}{\lambda^\star}$.

Since $X^\top X$ is symmetric, we can assume the following eigen value decomposition,

$$X^\top X = V\Lambda V^\top,$$

where $V \in \mathbb{R}^{n\times n}$ is an orthonormal matrix and $\Lambda \in \mathbb{R}^{n\times n}$ is a diagonal matrix with *positive* entries (since $XX^\top$ is positive semi definite and full rank). Now we will rewrite the constraint function evaluated at $w^\star$.

$$
\begin{aligned}
\|Xw^\star - y\|_2^2 &= \left\| X\left[cI + X^\top X\right]^{-1} X^\top y - y \right\|_2^2 \\
&= \left\| X\left[cI + X^\top X\right]^{-1} X^\top X w_0 - Xw_0 \right\|_2^2 \\
&= \left\| X\left[ \left[cI + X^\top X\right]^{-1} X^\top X - I \right] w_0 \right\|_2^2 \\
&= \left\| X\left[ \left[cVV^\top + V\Lambda V^\top\right]^{-1} V\Lambda V^\top - I \right] w_0 \right\|_2^2 \\
&= \left\| X\left[ V^{-\top}\left[cI + \Lambda\right]^{-1} V^{-1} V\Lambda V^\top - I \right] w_0 \right\|_2^2 \\
&= \left\| X\left[ V\left[cI + \Lambda\right]^{-1} \Lambda V^\top - I \right] w_0 \right\|_2^2 \\
&= w_0^\top \left[ V\left[cI + \Lambda\right]^{-1} \Lambda V^\top - I \right]^\top X^\top X \left[ V\left[cI + \Lambda\right]^{-1} \Lambda V^\top - I \right] w_0 \\
&= w_0^\top \left[ V\left[cI + \Lambda\right]^{-1} \Lambda V^\top - VV^\top \right]^\top V\Lambda V^\top \left[ V\left[cI + \Lambda\right]^{-1} \Lambda V^\top - VV^\top \right] w_0 \\
&= w_0^\top V \left[ \left[cI + \Lambda\right]^{-1} \Lambda - I \right]^\top V^\top V \Lambda V^\top V \left[ \left[cI + \Lambda\right]^{-1} \Lambda - I \right] V^\top w_0 \\
&= w_0^\top V \left[ \left[cI + \Lambda\right]^{-1} \Lambda - I \right]^\top \Lambda \left[ \left[cI + \Lambda\right]^{-1} \Lambda - I \right] V^\top w_0 \\
&= q^\top B q,
\end{aligned}
$$

where,

$$B = \left[ \left[cI + \Lambda\right]^{-1} \Lambda - I \right]^\top \Lambda \left[ \left[cI + \Lambda\right]^{-1} \Lambda - I \right],$$

which is a diagonal matrix, and $q = V^\top w_0$. Clearly,

$$q = V^\top w_0 \Rightarrow q \neq 0,$$

since $w_0 \neq 0$, and $V^\top$ has full rank. Now,

$$B_{ii} = \sigma_i \left( \frac{\sigma_i}{c + \sigma_i} - 1 \right)^2.$$

Therefore,

$$
\begin{aligned}
\|Xw^\star - y\|_2^2 = q^\top B q &= \sum_{i=1}^n B_{ii} q_i^2 \\
&= \sum_{i=1}^n \left( \frac{\sigma_i}{c + \sigma_i} - 1 \right)^2 \sigma_i q_i^2.
\end{aligned}
$$

Now, the feasibility of the primal solution implies,

$$\sum_{i=1}^n \left( \frac{\sigma_i}{c + \sigma_i} - 1 \right)^2 \sigma_i q_i^2 \leq \epsilon.$$

Without loss of generality, assume that $q_i \neq 0$. Since the LHS is a sum of non-negative terms, therefore,

$$\left(\frac{\sigma_i}{c + \sigma_i} - 1\right)^2 \sigma_i q_i^2 \leq \epsilon$$

$$\Rightarrow \frac{c^2}{(c + \sigma_i)^2} \sigma_i q_i^2 \leq \epsilon$$

$$\Rightarrow \frac{1}{(1 + \lambda^\star \sigma_i)^2} \sigma_i q_i^2 \leq \epsilon.$$

Clearly, as $\epsilon \to 0$, the LHS must also vanish, which is only possible if $\lambda^\star \to \infty$ since the remaining terms don't depend on $\epsilon$. $\qquad\square$

Next, we will characterize the dual solution of $(\mathrm{P}_e)$.

---

**Proposition D.2.** *Let Assumption 6 hold. Then, $\mu = -(XX^\top)^{-1}y$ is a dual solution of $(\mathrm{P}_e)$.*

---

*Proof.* We will proceed by explicitly computing the dual function $q(\mu)$ for $(\mathrm{P}_e)$. Note that the Lagrangian

$$L(w, \mu) = \frac{1}{2} \|w\|_2^2 + \mu^{\star\top} (Xw - y)$$

is a convex function of $w$. Therefore, the first order condition is necessary and sufficient to find a global minimiser of $L(w, \mu)$ with respect to $w$ ([58][Section 3.1.3.]). Proceeding,

$$\nabla_w L(w, \mu) = 0$$

$$\Rightarrow \nabla_w \left\{ \frac{1}{2} \|w\|_2^2 + \mu^\top (Xw - y) \right\} = 0$$

$$\Rightarrow w + X^\top \mu = 0$$

$$\Rightarrow w = -X^\top \mu.$$

Thus,

$$q(\mu) = \inf_{w \in \mathbb{R}^n} L(w, \mu) = L(-X^\top \mu, \mu)$$

$$= \frac{1}{2} \left\| -X^\top \mu \right\|_2^2 + \mu^\top \left( X(-X^\top \mu) - y \right)$$

$$= \frac{1}{2} \left\| X^\top \mu \right\|_2^2 - \left\| X^\top \mu \right\|_2^2 - \mu^\top y$$

$$= -\frac{1}{2} \left\| X^\top \mu \right\|_2^2 - \mu^\top y.$$

The first order condition for $q(\mu)$ must be satisfied by $\mu^\star$, therefore,

$$\nabla_\mu q(\mu)|_{\mu = \mu^\star} = 0$$

$$\Rightarrow \nabla_\mu \left\{ \frac{-1}{2} \mu^\top XX^\top \mu - \mu^\top y \right\} \bigg|_{\mu = \mu^\star} = 0$$

$$\Rightarrow -XX^\top \mu^\star - y = 0$$

$$\Rightarrow -XX^\top \mu^\star - y = 0$$

$$\Rightarrow \mu^\star = -(XX^\top)^{-1}y,$$

which completes the proof. $\qquad\square$

### D.2 Comparing equalities with double sided approximation

Consider the following equality constrained problem,

$$P_0^\star = \inf_{x \in \mathcal{X}} \quad \ell(x)$$
$$\text{subject to} \quad h_j(x) = 0, \quad \text{for } j = 1, \dots, J, \tag{$P_0$}$$

and its symmetric relaxation,

$$P_\epsilon^\star = \inf_{x \in \mathcal{X}} \quad \ell(x)$$
$$\text{subject to} \quad -\epsilon \le h_j(x) \le \epsilon, \quad \text{for } j = 1, \dots, J. \tag{$P_\epsilon$}$$

We analysed the relationship between a pair of similar problems in Appendix B.1.5 and Lemma B.4. The only difference is that we have now dropped the inequality constraints $g_i(x) \le 0$ from ($P_0$) for the sake of a clearer exposition.

The difference between ($P_0$) and ($P_\epsilon$) is in the constraints–($P_\epsilon$) has twice the number of constraints as ($P_0$), and they are all inequality constraints, as opposed to the equality constraints in ($P_0$). This implies that both problems have different dual variables. Let $\mu \in \mathbb{R}^J$ denote the dual variable for ($P_0$) and let $\mu_+, \mu_- \in \mathbb{R}_+^J$ denote the dual variables for ($P_\epsilon$), with $+$ corresponding to the upper bound and $-$ corresponding to the lower bound.

Consider the Lagrangian for ($P_\epsilon$):

$$L_\epsilon(x, \mu_+, \mu_-) = \ell(x) + \sum_{j=1}^J \mu_{+,j}(h_j(x) - \epsilon) + \sum_{j=1}^J \mu_{-,j}(-h_j(x) - \epsilon)$$

$$= \ell(x) + \sum_{j=1}^J (\mu_{+,j} - \mu_{-,j}) \, h_j(x) - \epsilon(\|\mu_+\|_1 + \|\mu_-\|_1). \tag{82}$$

It is clear from (82) that at $\epsilon = 0$, $L_\epsilon(x, \mu_+, \mu_-) = L_0(x, \mu_+ - \mu_-)$. Moreover, Lemma B.4 proves that when $\epsilon = 0$, $D_0^\star = D_\epsilon^\star$, i.e. the value of the dual problems are equal. Taking the infimum with respect to $x$ in Equation 82, we can obtain,

$$q_\epsilon(\mu_+, \mu_-) = q_0(\mu_+ - \mu_-) - \epsilon(\|\mu_+\|_1 + \|\mu_-\|_1). \tag{83}$$

Clearly $\text{Opt}(D_\epsilon)$ is not directly equal to $\text{Opt}(D_0)$, however from (83) it is clear that any solution of ($D_\epsilon$) can be transformed to a solution of ($D_0$), when $\epsilon = 0$, by the following mapping,

$$\mu := \mu_+ - \mu_-.$$

Similarly, we can verify (as in the proof of Lemma B.4), that any solution of ($D_0$) can be transformed to a solution of ($D_\epsilon$) (when $\epsilon = 0$) by the following mapping,

$$\mu_{+,j} := \max(\mu_j, 0) \text{ and } \mu_{-,j} := \max(-\mu_j, 0).$$

It is easy to verify that if the constraint functions are $B$-bounded, and the dual variables for ($P_\epsilon$) are initialised with some large positive value,

$$\mu_{+,j}, \mu_{-,j} := B' \gg \eta BT,$$

then the trajectories for the dual variables for both problems under Algorithm 1 are virtually the same. More precisely, if $\mu$ is the dual variable for ($P_0$), and $\mu_e = \mu_+ - \mu_-$ is the *effective* dual variable of ($P_\epsilon$), then their trajectories are identical. Theoretically, if $\mu_+, \mu_-$ are not initialised to a large enough values, the projection by $\max(\cdot, 0)$ on line 6 may cause differences in the trajectories. Figure 7 compares the dual trajectories for (P-DP) and its double sided relaxation for a small value of $\epsilon$, although all the dual variables were initialised to 0. See also Figure 1(b), where we see that "effective" dual variables of the relaxed problem converge to the dual variables of the equality constrained problem as $\epsilon \to 0$.

Note that if the "effective" dual trajectories are identical, then (82) implies that the primal trajectories are identical too, meaning that both methods are virtually the same. Finally, observe that when $\epsilon > 0$, (83) suggests that relaxing the constraints by $\epsilon$ is equivalent to regularizing the dual solution.

# E  Experiment details

Our implementation was made with `pytorch` and other standard ML libraries, and our codebase can be found at `https://github.com/abarthakur/equality-constrained-learning`. All experiments were run on an internal computing resources.

## E.1  Fairness

**Dataset.**   All our experiments on fairness applications were conducted on the COMPAS dataset downloaded from `https://github.com/propublica/compas-analysis`. Our starting point is the `compas-scores-two-years.csv` file, which was processed by the following pipeline following [22].

1. Remove rows where the attribute `days_b_screening_arrest` is not in the range $[-30, 30]$.

2. Remove all columns except `sex, age, race, juv_misd_count, juv_other_count, priors_count, c_charge_degree` and `two_year_recid`.

3. Recode the `race` attribute by clubbing values `Asian, Native-American, Other` to `Other`.

4. Split the data into train ($70\%$) and test ($30\%$) sets.

5. Encode the categorical attribute `race` with one-hot encoding.

6. Encode the binary attributes `sex, c_charge_degree` as (single) binary columns.

7. Quantize the numerical attributes `juv_misd_count, juv_other_count, priors_count` using the following bins, and then use one-hot encoding for the resultant categorical variable.

    (a) `priors_count` $- [(0, 0.99), (0.99, 1), (1, 2), (2, 3), (3, 4), (4, 1000)]$
    (b) `juv_misd_count` $- [(0, 0.99), (0.99, 1), (1, 1000)]$
    (c) `juv_other_count` $-[(0, 0.99), (0.99, 1), (1, 1000)]$

8. Bin the numerical attribute `age` by quantiles using 5 bins. Use the quantiles of the training set to quantize both training and test sets. Encode the resultant categorical variable using one-hot encoding.

9. Remove the binary attribute `two_year_recid` from the features and set it as the target variable $y$.

10. Copy the categorical attribute `race` from the features, and set it as the protected attribute defining $\mathcal{G}_j$ in (P-DP) and (P-F).

This process was repeated 10 times to produce 10 different train-test splits. After preprocessing and filtering, the full dataset (train and test together) consists of 23 features and 6,172 samples.

**Sigmoidal relaxation.**   We replace the indicator functions in (P-DP) and (P-F) with a sigmoid function in order to make the problem tractable (and aligned with our previous results). Explicitly, we use

$$\mathbb{E}_{\mathbb{P}}\left[\mathbb{I}\left[f_\theta(\mathbf{x}) > 0.5\right] | \mathbf{x} \in \mathcal{G}_j\right] \approx \mathbb{E}_{\mathbb{P}}\left[\sigma\left(\alpha\left(f_\theta(\mathbf{x}) - 0.5\right)\right) | \mathbf{x} \in \mathcal{G}_j\right],$$

and similarly for the overall rate $\mathbb{E}_{\mathbb{P}}\left[\mathbb{I}\left[f_\theta(\mathbf{x}) > 0.5\right]\right]$, where $\sigma$ denotes the sigmoid function. A similar approach was taken, e.g., in [24, 23].

**Model and hyperparameters.**   The following are the details of the hyperparameters.

1. The Adam optimizer was used for both primal (step 5 of Alg. 1) and dual updates (step 6–7 of Alg. 1) with learning rates of 0.2 (primal) and 0.001 (dual). The other hyperparameters for Adam were set to their default values ($\epsilon = 10^{-8}, \beta_1 = 0.9, \beta_2 = 0.999$).

2. A logistic regression classifier was used as the model. The cross-entropy loss was used as the objective.

3. The full dataset was used to compute the objective and constraint functions at each step.

4. The temperature parameter $\alpha$ in the sigmoidal approximation of the rate constraints (see Section 5), was set to $8.0$.

5. Training was terminated when the average of the last 100 Lagrangian iterates, changed by less than $10^{-5}$ in a window of 100 steps. In case the termination condition was not met, the algorithm ran till $16,000$ epochs.

**Computing resources.** The fairness experiments were run on CPU-only nodes with 32 threads, with about 10-20 runs in parallel. Each run took at most 20m (when the termination condition was not met).

## E.2 Boundary value problems

Let $\Omega \subseteq \mathbb{R}^d$ be a bounded connected region with boundary $\partial\Omega$, and define the domain of a BVP as $\mathcal{D} = \Omega \times (0, T]$ (where $T \in \mathbb{R}_+$) and let $\mathcal{H} = \{f_\theta \mid \theta \in \Theta\}$ be a set of functions defined on $\mathcal{D}$. A BVP is typically posed as the following problem : We call the three constraints the partial differential equation (PDE), boundary condition (BC) and initial condition (IC) respectively. As discussed in Section 2, the PDE constraint can be transformed to the constraint $\mathbb{E}_{\mathbb{P}_p}[(D[f_\theta](\mathbf{x}, t) - \tau(\mathbf{x}, t))^2] = 0$ for some distribution $\mathbb{P}_p$ that has support over the entire domain $\mathcal{D}$. The difference between this and the original formulation is essentially that the latter is with respect to the $L^2$ norm, while the former is a pointwise constraint, or equivalently over the $L^\infty$ norm with respect to any distribution whose support contains $\mathcal{D}$. The BC and IC constraints can similarly be transformed into their counterparts in (P-BVP) by picking $\mathbb{P}_b$ and $\mathbb{P}_i$ as distributions over $\partial\Omega \times (0, T)$ and $\Omega$ respectively.

**Dataset.** In our experiments we solve the convection BVP, with sinusoidal initial condition and periodic boundary conditions:

$$
\begin{aligned}
\text{find} \quad & f_\theta \in \mathcal{H} \\
\text{subject to} \quad & \frac{\partial f_\theta}{\partial t} + \beta \frac{\partial f_\theta}{\partial x} = 0 \quad && \text{for } x \in [0, 2\pi], t \in (0, 1], \\
& f_\theta(0, t) = f_\theta(2\pi, t) && \text{for } t \in [0, 1], \\
& f_\theta(x, 0) = \sin(x) && \text{for } x \in [0, 2\pi].
\end{aligned}
\quad \text{(BVP-C)}
$$

We express this as the following instance of (P-BVP) :

$$
\begin{aligned}
\underset{\theta \in \Theta}{\text{minimize}} \quad & \frac{\alpha}{2}\|\theta\|_2^2 \\
\text{subject to} \quad & \mathbb{E}_{\mathbb{P}_p}\left[\left(\frac{\partial f_\theta}{\partial t} - \beta \frac{\partial f_\theta}{\partial x}\right)^2\right] = 0, \\
& \mathbb{E}_{\mathbb{P}_b}\left[(f_\theta(0, t) - f_\theta(2\pi, t))^2\right] = 0, \\
& \mathbb{E}_{\mathbb{P}_i}\left[(f_\theta(x, 0) - \sin(x))^2\right] = 0.
\end{aligned}
\quad \text{(P-C)}
$$

Here $\mathbb{P}_p, \mathbb{P}_b$ and $\mathbb{P}_i$ are taken to be uniform distributions over $[0, 2\pi] \times [0, 1]$, $[0, 1]$ and $[0, 2\pi]$ respectively. We followed a training and evaluation setup (including model and hyperparameters) similar to [62]. For the training dataset, 1000 collocation points $(x, t)$ were sampled uniformly, and *dynamically* at each iteration/epoch, from the domain $\mathcal{X} \times \mathcal{T} = [0, 2\pi] \times [0, 1]$ which were used to compute the PDE, IC and BC constraints. For evaluation, a uniform grid over $(x, t)$ with $(512, 251)$ divisions was used as the test set. We used the implementation by [62] to generate the ground truth. Evaluation was done with respect to the relative L2 error on the test set, which can be written as,

$$
\text{Relative } L^2 \text{ Error} = \sqrt{\frac{\sum_{n=1}^{N} (f_\theta(x_n, t_n) - f^\dagger(x_n, t_n))^2}{(f^\dagger(x_n, t_n))^2}},
$$

where $f^\dagger$ is the ground truth solution and $(x_n, t_n)_{n=1}^{N}$ is the test set.

**Model and hyperparameters.** The following are the details of the hyperparameters.

1. All models were trained for 300k iterations or epochs.

2. Both methods (PINN/ (P-BVP)) used a 4 layered MLP with 50 hidden neuron layers and Tanh activation to represent the primal model $f_\theta$.

3. Adam was used to optimize the primal model, with an initial learning rate of 1e-3, and a learning rate scheduler was used that multiplies the learning rate by a factor 0.9 every 5000 steps/epochs (available as the class StepLR from pytorch). The other hyperparameters for Adam were left at their default values ($(\beta_1, \beta_2, \epsilon)$ =(0.9, 0.999, 1e-8)).

4. A batch size of 1000 was used, and each batch was sampled dynamically at each step.

5. For the PINN formulation, the multipliers for the PDE, BC and IC losses were picked to be (1, 100, 100) respectively.

6. No weight decay was applied to either method. In particular, $\alpha$ was set to 0 in (P-C).

7. The dual variables were optimized using Adam with learning rate 1e-4. The other hyperparameters were left at their default values.

**Computing resources.** The BVP experiments were run (one run at a time) on an accelerated node with 16 CPU threads and one GPU. Both PINN and (P-C) took approximately 1h40m for each run (300k epochs), with negligible overhead for the constrained method.

### E.3 Interpolating classifiers

**Dataset.** We used the CIFAR-10 and CIFAR-100 datasets from the torchvision library. We applied dynamic data augmentation, i.e. we transformed samples at train time using the following pipeline :

1. random cropping of the image after padding with 4 pixels,

2. random horizontal flip,

3. random rotation between -15 to 15 degrees,

4. channel-wise normalization by the (pixel) mean and standard deviation from the training set.

**Model and hyperparameters.**

1. The cross entropy loss function was used for $\ell_0$. All runs were for 200 epochs.

2. Both methods (ERM/(P-CI)) were trained with a ResNet18 model, with a standard modification to the first layer using smaller filter sizes to adapt it to smaller images (compared to ImageNet). The reader is referred to our codebase (or the open source repository `https://github.com/kuangliu/pytorch-cifar.git`) for the exact implementation.

3. SGD was used to optimize the primal model for both datasets, with a batch size of 128, and an initial learning rate of 1e-3, and a momentum hyperparameter of 0.9. For CIFAR-10, a cosine annealing learning rate scheduler (CosineAnnealingLR) was used with $T_{\max} = 200$. For CIFAR-100, the initial learning rate was decayed by a factor of 0.2 at the 60th, 120th and 160th epoch (i.e. these were the milestones passed to MultiStepLR).

4. A weight decay value of 5e-4 was used for all runs.

5. Adam was used to optimize the dual variables, with a learning rate of 1e-4 for CIFAR-10, and 1e-5 for CIFAR-100. All other hyperparameters were held to their default values.

**Computing resources.** The interpolation experiments were run (one run at a time) on a workstation with 12 CPU threads and one GPU. On CIFAR-10, ERM took about 16m while (P-CI) took about 18m. On CIFAR-100, the overhead was larger, with ERM still taking about 16m while the constrained method took out 31m. This may be mitigated by a vectorised implementation of the Lagrangian, but is left for future work.

# F   Additional plots

## F.1   Fairness

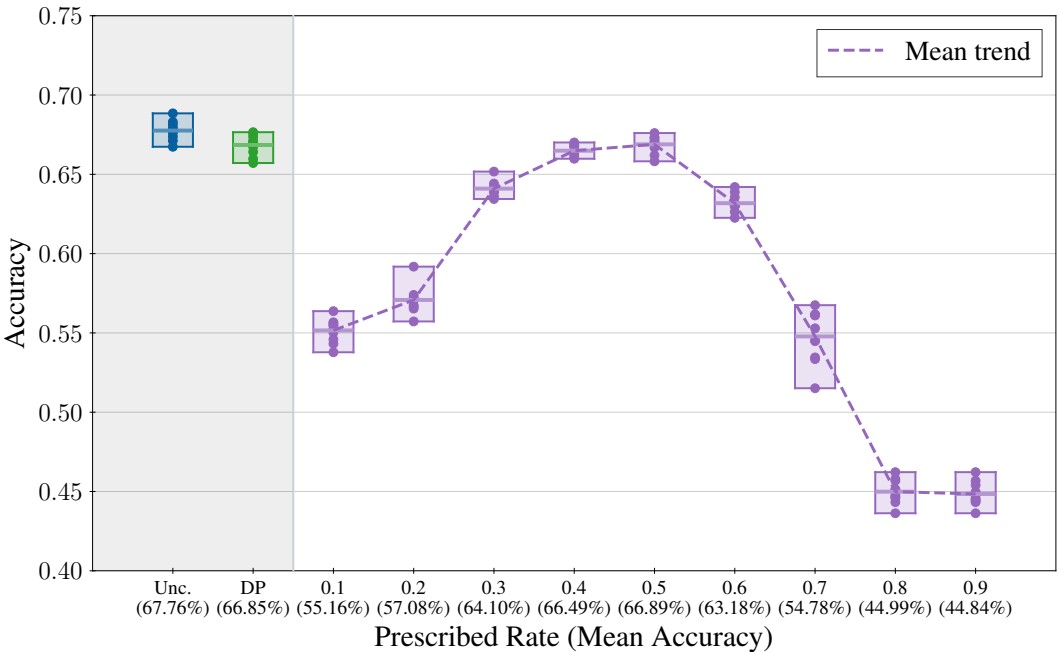

Figure 5: *Prescribed rates.* We see that at $r_j = 0.5$, the mean accuracy of the model is slightly better than the Exact DP solution. At the same time, for $r_j = 0.5$, the model achieves a group disparity that is comparable to the Exact DP solution (see Figure 2(b)) but at a different rate. Thus (P-F) enables new tradeoffs between group disparity and accuracy, that cannot be found by using (P-DP).

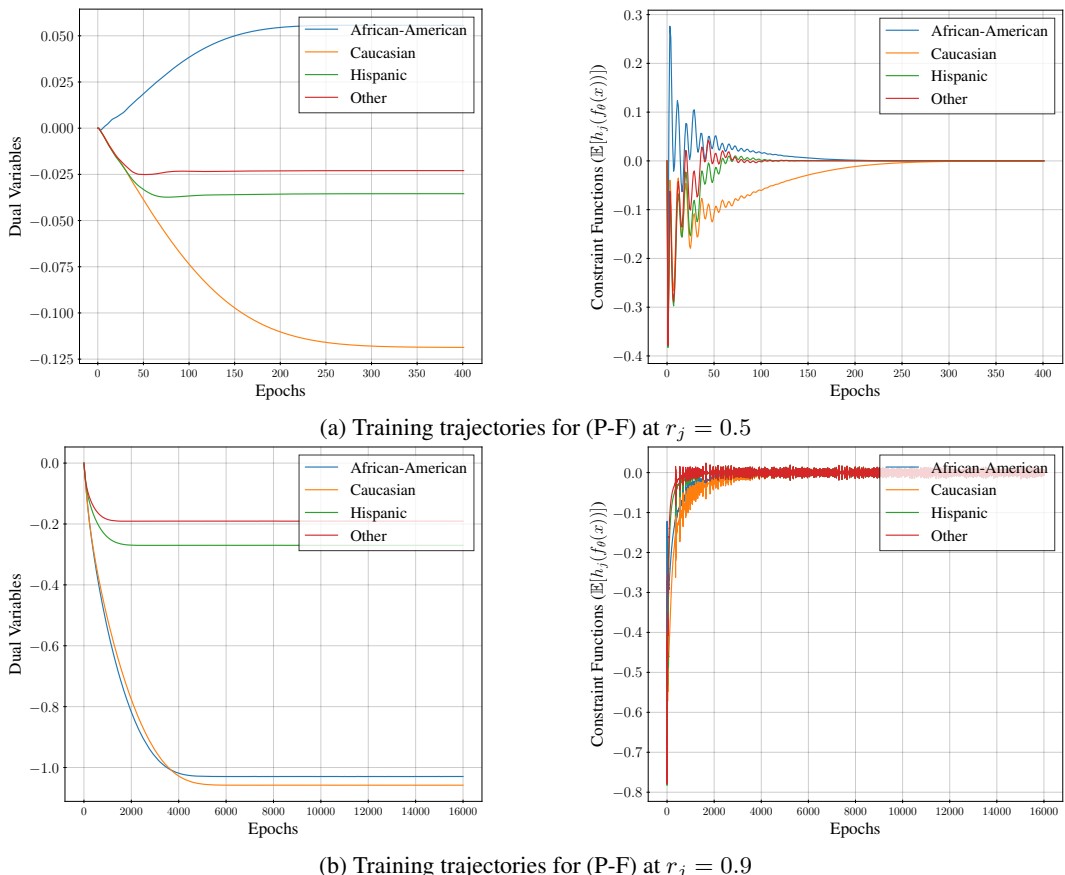

(a) Training trajectories for (P-F) at $r_j = 0.5$

(b) Training trajectories for (P-F) at $r_j = 0.9$

Figure 6: *Prescribed rates.* We see that while training converges quickly for $r_j = 0.5$, at about 400 epochs, it takes more than 10 times as many epochs for $r_j = 0.9$ to converge (and it still does not reach the termination condition). This is understandable, since the *population* prevalence of the binary label, i.e. $P(y = 1)$, is about 0.46. As such, prescribing a very high rate, e.g. $r_j = 0.9$, is a hard constraint to satisfy while maximising the predictive performance. And in fact, it is so restrictive that the model yields the constant classifier after thresholding.

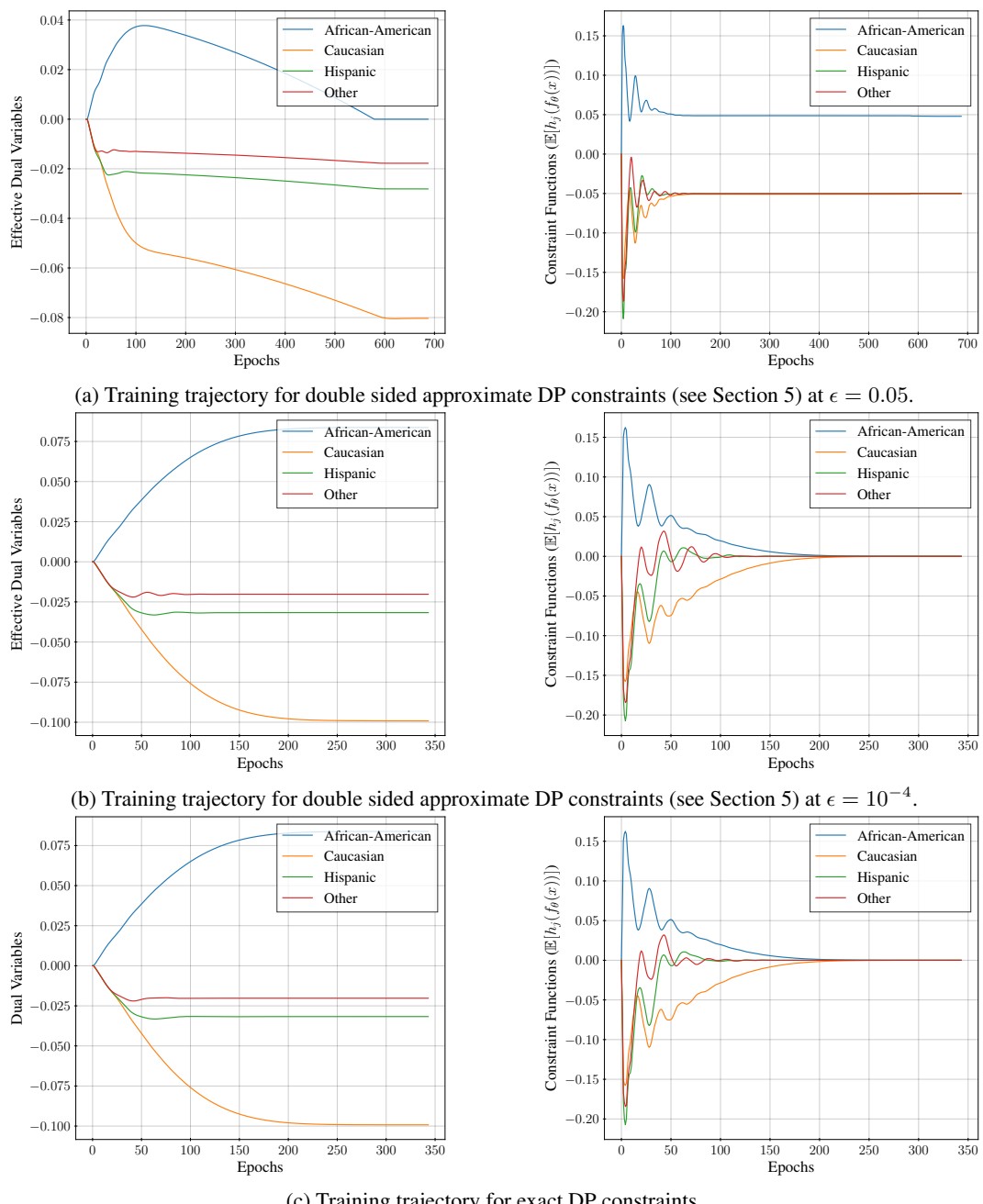

(a) Training trajectory for double sided approximate DP constraints (see Section 5) at $\epsilon = 0.05$.

(b) Training trajectory for double sided approximate DP constraints (see Section 5) at $\epsilon = 10^{-4}$.

(c) Training trajectory for exact DP constraints.

Figure 7: *Exact vs approximate fairness.* The trajectories of the effective dual variables (i.e. the difference of pairs) and constraint functions of the approximately constrained problem and the exact constrained problem are very similar at $\epsilon = 10^{-4}$, as compared to at $\epsilon = 0.05$. Section D.2 discusses theoretical reasons for why this is the case.

## F.2 Boundary value problems

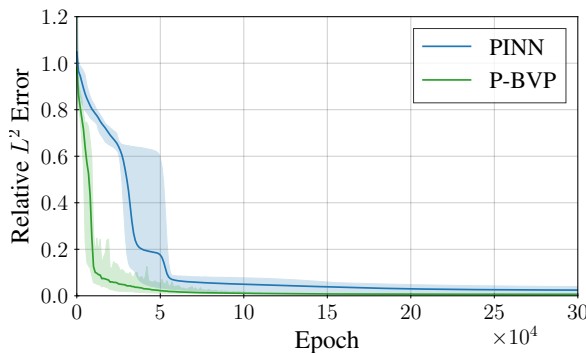

Figure 8: Evolution of relative $L^2$ error for Convection BVP ($\beta = 30$). We see that besides a smaller final error, P-BVP also converges much quicker to a smaller error.

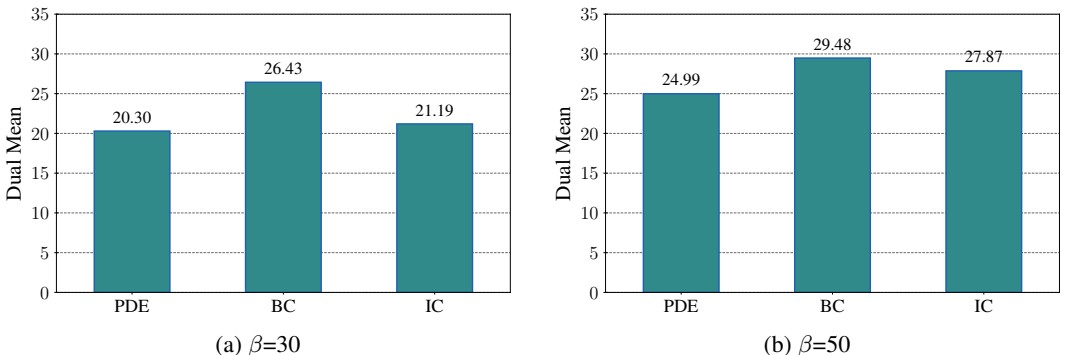

(a) $\beta$=30          (b) $\beta$=50

Figure 9: This figure shows the final dual variable per constraint, averaged across 5 runs. The dual values for the boundary condition are the largest for both values of $\beta$, which can possibly be explained by the simplicity of the solution to the convection equation with a sinusoidal initial condition - which makes the *propagation* of the information through the boundary condition the challenging part of the problem.

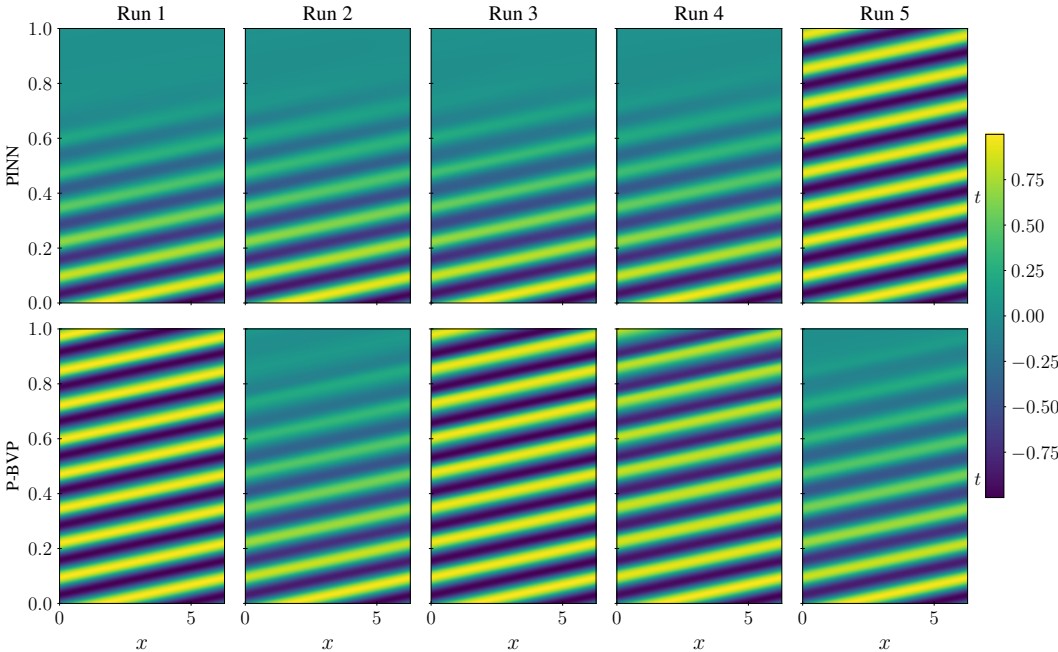

Figure 10: *Predicted solutions for $\beta = 50$.* We can visually see the difference between the solutions of PINN and (P-BVP), where the former struggles to solve the problem for most runs.

## F.3 Interpolating classifiers

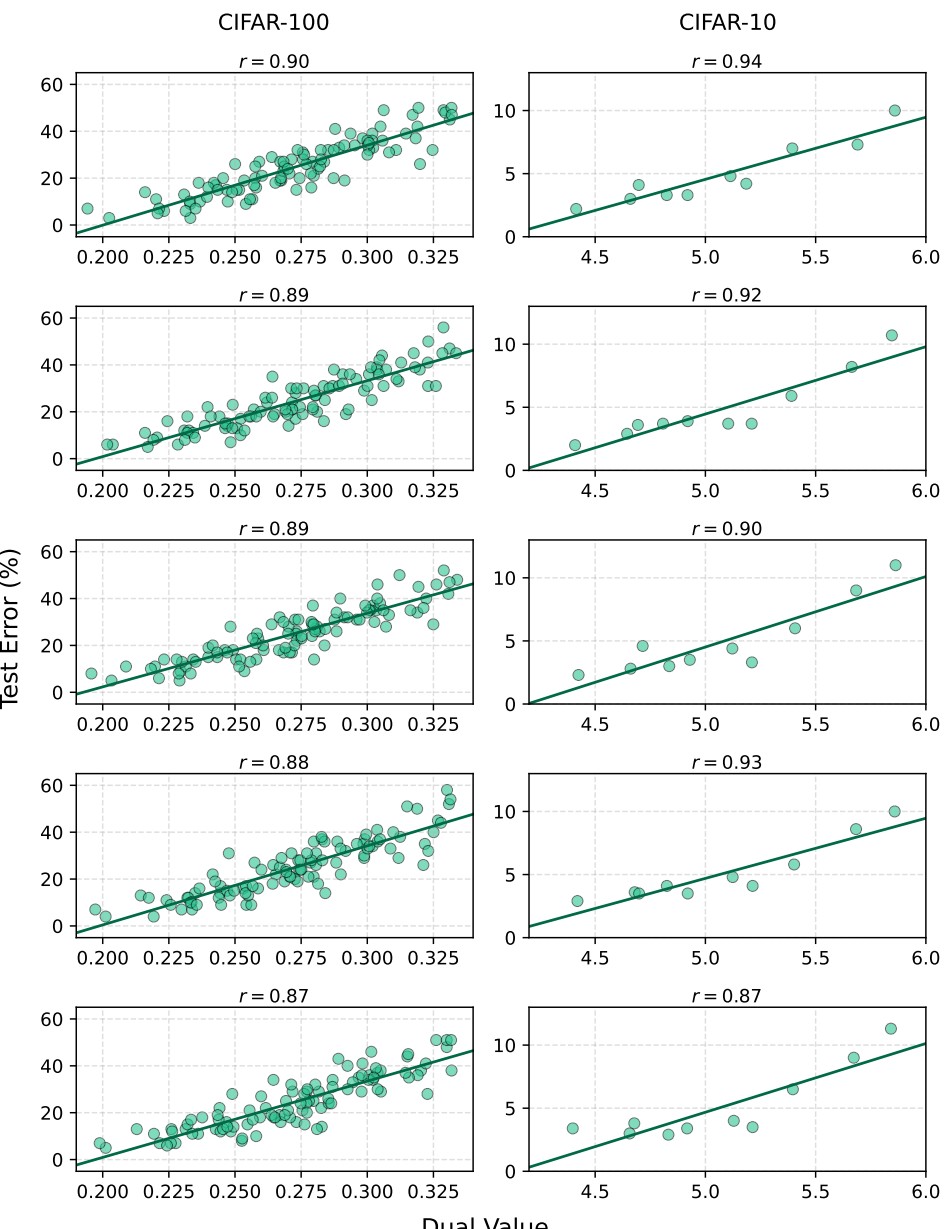

Figure 11: *Dual value vs test error (for all seeds).* We see a strong linear relationship between the dual value and the test error of each class across runs, for both CIFAR-10 and CIFAR-100, with correlation consistently greater than $0.87$.

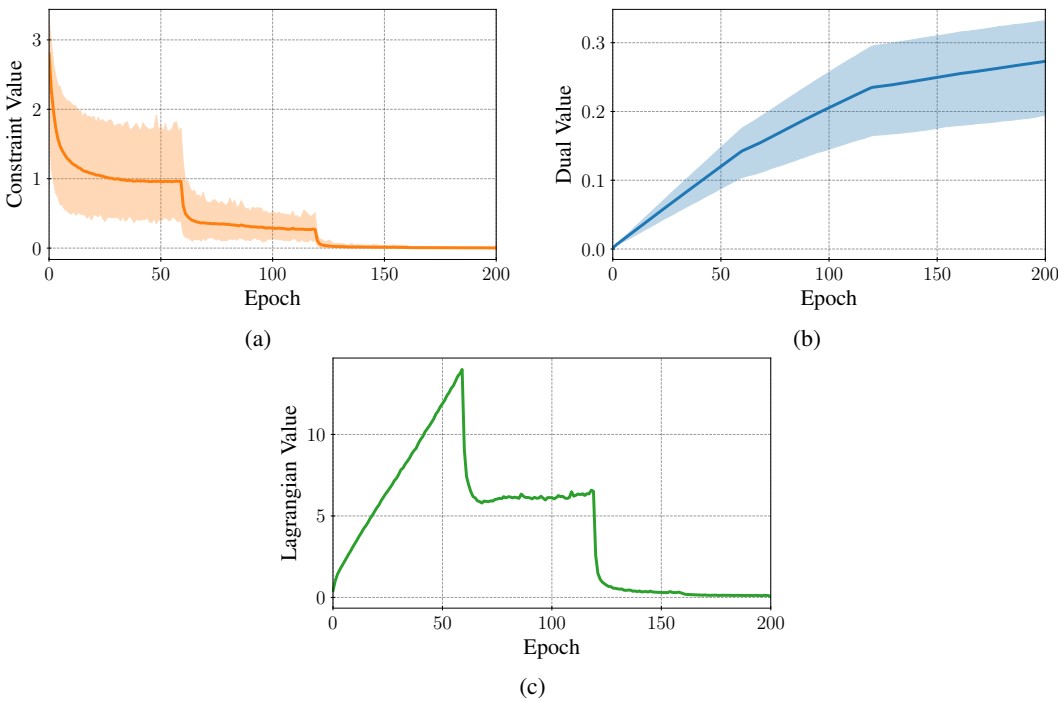

Figure 12: *Training plots for CIFAR-100 for one random seed.* Subfigures (a) and (b) show the constraint and dual values during training, with the mean across classes shown by the solid line, while the maximum and minimum (across classes) is denoted by the extents of the shaded region. Subfigure (c) shows the evolution of the Lagrangian.

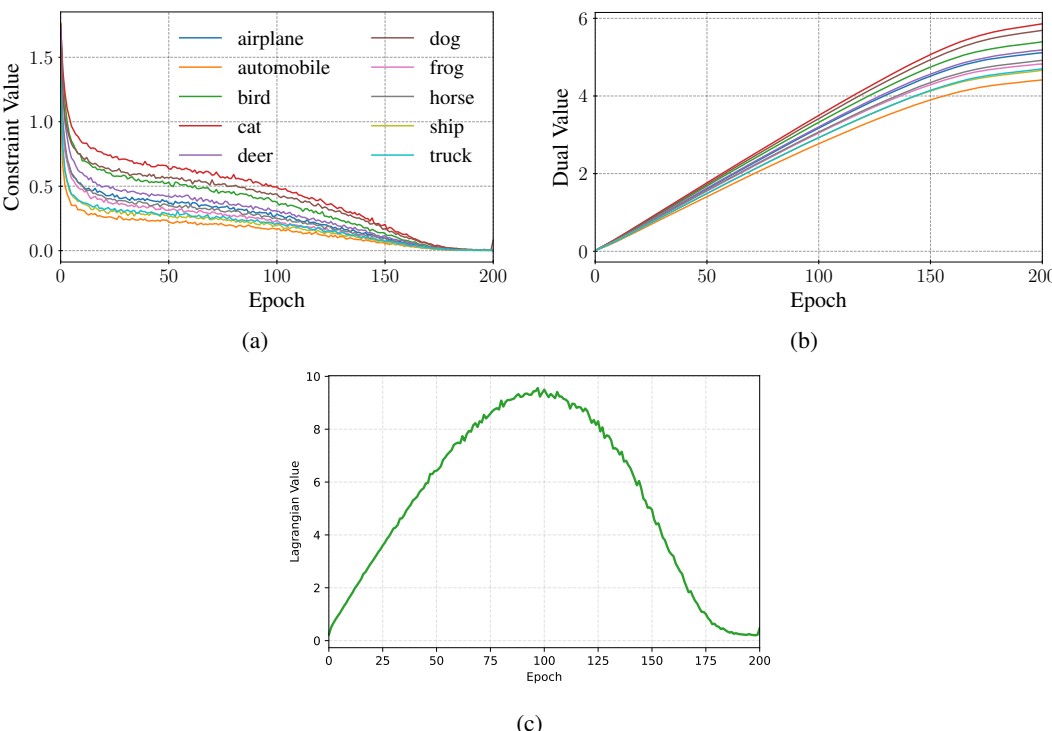

Figure 13: *Training plots for CIFAR-10 for one random seed.* Each class is denoted by a different color in subfigures (a) and (b).

## Appendix references

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
