# OpenReview forum: "Learning with Statistical Equality Constraints"
_NeurIPS.cc/2025/Conference — NeurIPS 2025 poster_

### Official Review · Reviewer_8xzy · 2025-06-13

**Clarity:** 4
**Significance:** 3
**Originality:** 3
**Rating:** 5
**Confidence:** 2

**Summary:**

This article studies the statistical learning problem with statistical equality constraints. It establishes a generalization bound (Thm 3.1) for this problem and discusses clearly and explicitly how this result is complementary to known properties of inequality-constrained statistical optimization problems. It introduces a primal-dual algorithm for the problem that alternates between the solution of unconstrained, empirical learning problems and the optimization of Lagrange parameters via an oracle (Assumption 6). The empirical dual value is related to the average of the Lagrangian iterates (Thm., 4.1), without solving the primal recovery problem, and with respect to the oracle. Toward experiments, the oracle is replaced by single alternating updates. The effectiveness of this practical algorithm is analyzed in an application to practical instances of the problem in which fairness is expressed in the form of statistical equality constraints.

**Questions:**

- While I understand both (P) and (P-DP), it took me a bit to see how (P-DP) can be put in the form (P). Would you like to introduce the $g_i$ that does this instead of using the Iverson bracket in (P-DP)?
- Define the image of $f_\theta$ consistently as $\mathcal{Y}$ (as in Line 52) or $\mathbb{R}^K$ (as in Line 51).

**Ethical Concerns:**

["NO or VERY MINOR ethics concerns only"]

**Final Justification:**

I find all my points addressed and maintain my positive rating.

**Limitations:**

The article communicates explicitly all assumptions made by the authors. This includes, in particular, the important assumption of uniform convergence (Assumption 5). This assumption is satisfied in many relevant cases, including families with finite VC-dimension.

**Paper Formatting Concerns:**

The typesetting is excellent without being too sophisticated, and without deviating much from the template.

**Quality:**

4

**Strengths And Weaknesses:**

Strengths:
- The problem is interesting and important.
- The theoretical results are plausible, the high-level reasoning in the main article is correct and convincing. (I have not checked the proofs in detail).
- Although the article is an immediate continuation of [11,32], the authors are explicit about this fact (e.g. in Line 98) and explain why this continuation is non-trivial (notably in Remark 2.1).
- The empirical method is convincing; the empirical results are interesting and insightful.
- The article is well-written.

Weaknesses:
- I am not sure whether the case of deterministic (hard) equality constraints is merely a special case. The authors should discuss this. If special attention is needed, I suggest to use a more specific title "Learning with Statistical Equality Constraints".

---

> ### Author Rebuttal · Authors · 2025-07-31
>
> > **W1. I am not sure whether the case of deterministic (hard) equality constraints is merely a special case... I suggest to use a more specific title "Learning with Statistical Equality Constraints".**
>
> The reviewer has a point. We will include an explicit remark on this fact following Theorem 3.1 and adopt the title they suggest.
>
>
> > **Q1. While I understand both (P) and (P-DP), it took me a bit to see how (P-DP) can be put in the form (P). Would you like to introduce the $g_i$ that does this instead of using the Iverson bracket in (P-DP)?**
>
> The reviewer has a point that this transformation is not straightforward. Explicitly, (P-DP) can be cast in the form of (P) by taking
>
> $g\_j(f\_{\theta}(x), y) = \mathbb{I}[f\_{\theta}(x) > 0.5] \mathbb{I}[x \in \mathcal{G}\_j] - \mathbb{I}[f\_{\theta}(x) > 0.5]$
>
> and $\mathbb{Q}\_j = \mathbb{P}$, the data distribution. Alternatively, we could consider $g\_j(f\_{\theta}(x), y) = \mathbb{I}[f\_{\theta}(x) > 0.5]$ and take the expectation with respect to $\mathbb{Q}\_j = \mathbb{P}(\cdot | x \in \mathcal{G}\_j) - \mathbb{P}$. While the latter is not a probability measure, the guarantees of Theorem 3.1 hold for general measures.
>
> We will include the first formulation in the camera-ready version of the manuscript.
>
> > **Q2. Define ... consistently as in Line 52...**
>
> We thank the reviewer for pointing out this typo. The correct function signature is $f\_{\theta}: \mathcal{X} \to \mathbb{R}^K$. We will fix this and other typos in the camera-ready version.

---

> > ### Comment · Reviewer_8xzy · 2025-08-05
> > **Acknowledgement**
> >
> > I find all my points addressed and maintain my positive rating.

---

### Official Review · Reviewer_1hsE · 2025-07-01

**Clarity:** 3
**Significance:** 3
**Originality:** 2
**Rating:** 4
**Confidence:** 3

**Summary:**

This paper studies the problem of optimization with equality constraints, where the duality and generalization results obtained for inequality-constrained learning apply to these problems. This is motivated by the practical needs that machine learning systems with goals such as in fairness tends to require equality constraints instead of a bound. This paper developed a generalization theory for equality constrained learning tasks:
- the paper derived additional regularity conditions under which duality and sensitivity results for non-convex equality-constrained optimization problems are obtained
- the paper then propose a practical algorithm based on solving a sequence of unconstrained, empirical learning problems, and characterized the approximation and generalization error of its solutions
- experiments using examples in group fairness show the effectiveness of the proposed solution

**Questions:**

- there is a lack of any discussions on the optimality of the provided error bound: under what scenarios it is tight  / lose?

**Ethical Concerns:**

["NO or VERY MINOR ethics concerns only"]

**Final Justification:**

The author response was helpful in addressing my concern about the oracle optimizer. I will keep my original score for a weak acceptance.

**Limitations:**

yes

**Quality:**

3

**Strengths And Weaknesses:**

Strengths:
- this paper studied a practical but less explored formulation with the equality constraints instead of inequality constraints, the problem is well-motivated from practical applications especially fairness
- the derivation of the error bound is intuitive via the duality formulation, and the bound itself reflects the different drivers of the error, including the sensitivity on constraints / formulation, model capacity, sample size etc
- experimental results show that with small tolerance, inequality formulation achieves similar results as the proposed DP, however the difference becomes non-negligible when the tolerance parameter is high

Weakness:
- The algorithm in section 4 assumes the access to an oracle optimizer. Although the experiments show that single alternating updates with an optimizer eg Adams work relatively well, the results are only based on simple setting and one dataset. The experimental support should be strengthened with more diverse setups and complex data.

---

> ### Author Rebuttal · Authors · 2025-07-31
>
> > **W1. The algorithm in section 4 assumes the access to an oracle optimizer... The experimental support should be strengthened with more diverse setups and complex data.**
>
> Indeed, the convergence guarantees in Theorem 4.1 hold for Algorithm 1 which does require an *unconstrained* learning problem to be (approximately) solved in step 5. This is not, however, *necessary* to obtain good results in practice as our numerical examples illustrate (see, e.g., Figure 4(a) in Appendix F). This fact has been observed in a myriad of constrained learning problems involving inequality constraints of various types (see, e.g., [11,12,19,24,32]). While we believe that explaining this behavior is an interesting research direction, it is beyond the scope of this work.
>
> Nevertheless, we agree with the reviewer that additional examples could help illustrate not only the behavior of the algorithm, but also other use cases of equality constraints. In our response to W1 of `Reviewer Q9iw`, we detail an application to solving a partial differential equation (PDE). But (P) and Algorithm 1 can also be used to train interpolating classifiers. Indeed, if we train a CNN (ResNet18) on CIFAR-10 using the traditional ERM formulation with cross-entropy loss, we obtain a test accuracy of 85.18% after approximately 90 epochs (using ADAM with lr=5e-4 and batch size=256). The same interpolating classifier can also be trained using (P) to disaggregate the classes as in
> $$
> \begin{align*}
> \text{minimize}_{\theta \in \Theta} & \;  \|\theta\|^2 \\\\
> \text{subject to} & \\; \mathbb{E}\_{\mathbb{P}}[\ell(f\_{\theta}(x),y)|y=j] = 0, \quad \text{for } j = 1,\ldots, 10.
> \end{align*}
> $$
> When using Algorithm 1 to solve this problem, the dual variables have the effect of reweighting the loss of each class [see the Lagrangian definition in (1)]. This leads to an improvement in the test accuracy (86.06%) using the same hyperparameters as before. This is due to classes that are harder to fit (e.g., "bird") being "prioritized" during training. Due to NeurIPS restrictions on the use of images in the rebuttal phase, we cannot show the complete training dynamics, but we show the final dual variables values ($\mu_j$) below.
>
> If the reviewer finds it beneficial, we will use the additional content page of the camera-ready to include these use cases. Additionally, if they had another specific application and dataset in mind, we would happily consider using it in our experiments as well.
>
> | airplane | automobile | bird    | cat     | deer    | dog     | frog    | horse  | ship   | truck  |
> | -------- | ---------- | ------- | ------- | ------- | ------- | ------- | ------ | ------ | ------ |
> | 1074.09  | 912.54     | 1124.30 | 1232.01 | 1061.10 | 1180.55 | 1039.51 | 984.14 | 971.45 | 957.56 |
>
>
> > **Q1. there is a lack of any discussions on the optimality of the provided error bound: under what scenarios it is tight / lose?**
>
> In contrast to unconstrained learning, to the best of our knowledge, *no lower bounds* have been proposed for the constrained learning error $|P^\star - \hat{D}^\star|$. As such, we cannot formally discuss the tightness of our bound. We agree that this is an interesting direction, but it is beyond of the scope of this work. Qualitatively, we believe there are situations in which the upper bound *can* be improved, particularly when (P) is convex and more refined perturbation and statistical results can be used (see, e.g., [Bonnans and Shapiro. "Perturbation Analysis of Optimization Problems"; Shapiro et al. "Lectures on Stochastic Programming: Modeling and Theory"] which are references [16;18] from the appendices).

---

> > ### Comment · Reviewer_1hsE · 2025-08-08
> > **Acknowledgement**
> >
> > The author response addressed all my points and I maintain my positive score.

---

### Official Review · Reviewer_SL24 · 2025-07-03

**Clarity:** 3
**Significance:** 3
**Originality:** 4
**Rating:** 4
**Confidence:** 3

**Summary:**

This paper addresses the problem of learning under equality constraints. It proposes a theoretical framework for non-convex problems and proves strong duality and generalization bounds under a set of relatively strong assumptions. Based on this, the authors introduce a practical primal-dual algorithm that reduces the problem to a sequence of unconstrained optimizations.

**Questions:**

please see weakness section

**Ethical Concerns:**

["NO or VERY MINOR ethics concerns only"]

**Final Justification:**

I am satisfactory with the author's responses and vote for the acceptance

**Limitations:**

yes

**Quality:**

3

**Strengths And Weaknesses:**

**Strengths**

- The paper are clearly written.
- The paper fills a theoretical gap by extending constrained learning theory from inequality to equality constraints.

**Weaknesses**

- Would Assumption 1 still hold if multiple potentially conflicting fairness constraints are imposed simultaneously? Have the authors considered or tested such settings?
- It is unclear how robust the theory is when the assumption only approximately holds or fails in more constrained or imbalanced scenarios.
- The algorithm requires solving a full learning problem at each dual update, which can be costly with large models or many constraints.

---

> ### Author Rebuttal · Authors · 2025-07-31
>
> > **W1. Would Assumption 1 still hold if multiple potentially conflicting fairness constraints are imposed simultaneously? Have the authors considered or tested such settings?**
>
> If (P) includes conflicting fairness criteria (in the sense of impossibility theorems such as [Kleinberg et al - Inherent Trade-Offs in the Fair Determination of Risk Scores]), then it is infeasible (has no solution), which in turn implies that Assumption 1 does not hold. Note that since the population problem (P) has no solution in this setting, which makes its value $P^\star$ undefined (or, as per the typical convention, $\infty$), which invalidates any traditional notion of generalization. In such a case, at least one of the equalities needs to be relaxed. In view of the duality results in this paper, it would make sense to relax the most stringent one (largest dual variables) or use an automatic relaxation technique such as in [Hounie et al., Resilient Constrained Learning, 2023]. If the reviewer believes it would be beneficial, we can include an example of such a procedure in the appendices of the camera-ready.
>
>
> > **W2. It is unclear how robust the theory is when the assumption only approximately holds or fails in more constrained or imbalanced scenarios.**
>
> As we comment throughout the paper, the majority of our assumptions are typical in the literature and mild in practice, including Lipschitz continuity of losses and models (Assumption 3 and 4), sufficiently rich parametrizations (Assumption 3), atomlessness of distributions (Assumption 2), and uniform convergence (Assumption 5). As the reviewer correctly notices, Assumption 1 is potentially more stringent. Yet, what it does is ensure that (P) and neighboring problems have at least one solution (as we explain in line 131). In other words, that (P) is and remains feasible even if when perturbed. As we explain above, this is necessary to even talk about generalization: if (P) has no solution, then its value $P^\star$ is undefined and cannot be approximated. This point will be clarified in the camera-ready version of the manuscript.
>
> > **W3. The algorithm requires solving a full learning problem at each dual update, which can be costly with large models or many constraints.**
>
> Indeed, the convergence guarantees in Theorem 4.1 are for Algorithm 1 which does (approximately) solve an *unconstrained* learning problem (step 5). That is, however, not *necessary* to obtain good results in practice as our numerical examples illustrate. Indeed, Figure 4(a) in Appendix F shows that the results of taking a single gradient step at each dual update does yield results that satisfy the constraints. In fact, there is substantial empirical evidence that a perfect oracle (full solution in step 5) is not necessary for constrained learning problems (see, e.g., [11,12,19,24,32]). While we believe that this is an interesting research direction, it is beyond the scope of this work. Note also that while by the model size does affect the complexity of step 5 (primal gradient), the number of constraints only affects the dimensionality of the dual problem, whose gradient is trivial to compute (steps 6 and 7).

---

### Official Review · Reviewer_Q9iw · 2025-07-04

**Clarity:** 2
**Significance:** 3
**Originality:** 3
**Rating:** 5
**Confidence:** 2

**Summary:**

This paper presents a theory and algorithm for equality constrained learning, addressing the limitations of existing approaches to inequality constraints with an application to fair machine learning. The authors dualized a learning problem with inequality and equality constraints, investigated the generalization bound, and proposed a dual ascent algorithm to solve the problem. Moreoverm the authors demonstrated that the proposed algorithm works in practice, even though its theoretical properties were analyzed under the assumption of access to an oracle optimizer, which is not guaranteed in real applications.

**Questions:**

- For my major concerns, please see the weaknesses above.
- In Remark 2.1, for the claim that 'small values $\epsilon > 0$ lead to numerical issues and worse generalization bounds arising from large Lagrange multipliers,' the authors refer to Theorem 3.1. However, it is not immediately clear to me how Theorem 3.1 supports this claim.
- There are some minor formatting issues. It would be better to use consistent terms for references. For example, consistently using either the full term or the abbreviation (e.g., theorem vs. thm, section vs. sec, assumption vs. ass, lemma vs. lem) to avoid confusion.

**Ethical Concerns:**

["NO or VERY MINOR ethics concerns only"]

**Final Justification:**

As highlighted in the original manuscript and further reinforced by the authors’ responses, learning with equality constraints has many applications, yet the supporting theory and algorithms remain underdeveloped. The manuscript is well-organized, and with the added results and clarifications, I see no reason to undervalue the contribution.

**Limitations:**

Although the examples given in the paper seem decent, I doubt fair machine learning is the best example to motivate equality constrained learning. When developing fair models, the goal is usually to balance overall performance with fairness across sensitive groups, rather than enforcing strict equality constraints at all costs. In many cases, it is hard to justify compromising the performance of a better-performing group (often the majority) to improve fairness. This raises a concern that if equality constraints are applied, we may end up with trivial or counterproductive solutions that lower performance for all groups. I think this is why recent algorithmic fairness work shifted to encourage a better performance-fairness trade-off instead.

**Quality:**

3

**Strengths And Weaknesses:**

strengths
- Provides a theory and algorithm for equality constrained learning which has been underexplored.
- Provides theoretical results (though I am not fully confident in their correctness, as they are outside my area of expertise) and a numerical algorithm that works in practice.

weaknesses
- The major contribution of this paper is to develop a theory and algorithm for equality constrained learning. But the examples, applications, and experiments seem biased solely toward fair machine learning. The authors insisted in the introduction that many important requirements are expressed as equality constraints. Why did the authors focus on fair machine learning? I doubt that fair machine learning is the best beneficiary of this work (please see limitations). It would have been better if the authors presented more diverse examples and empirically demonstrated the advantages of applying equality constraints in those examples.
- Figure 1 shows that the approximate algorithm with a small tolerance for the inequality constraints works as good as the exact algorithm with equality constraints. If we are able to achieve a sufficiently good approximation by choosing an appropriate tolerance for the inequality constraint, why do we need an exact algorithm? This example does not support the claim "small values $\epsilon > 0$ lead to numerical issues and worse generalization bounds (although it is true). An example that the inequality with small tolerance cannot approximate the exact one will be more convincing.
- In summary, I’m not entirely convinced that developing theory and optimization algorithms for equality constrained learning is important, however, I do believe this line of work is valuable and will offer useful contributions.

---

> ### Author Rebuttal · Authors · 2025-07-31
>
> We thank the reviewer for their comments.  We address their concerns point-by-point below.
>
>
> > **W1: Why did the authors focus on fair machine learning? I doubt that fair machine learning is the best beneficiary of this work (please see limitations)**
>
> As the reviewer points out, equality constraints find many applications beyond fairness. The manuscript uses fairness to illustrate *one* potential use of equalities since the focus is on deriving generalization bounds for equality-constrained learning problems (despite non-convexity). For instance, consider the unsupervised learning problem of approximating solutions of a differential equations (DE), namely, finding $u$ such that
>
> $$
> \begin{alignat*}{2}
> \\mathcal{D}\[u\](x,t) &= 0, \\quad &&\\forall (x,t) \\in \\Omega \\times (0,T\]\\\\
> u(x,t) &= h(x,t), \\quad &&\\forall x \\in \\partial \\Omega \\times (0,T\]\\\\
> u(x,0) &= h(x,0), \\quad &&\\forall x \\in \\Omega \\cup \\partial\\Omega
> \end{alignat*}
> $$
>
> Here, $\mathcal{D}$ denotes a differential operator, $\Omega$ is a domain with boundary $\partial \Omega$, and $h$ defines the boundary (and initial) conditions (see, e.g., [Raissi et al "Physics-informed neural networks: A deep learning framework for solving forward and inverse problems involving nonlinear partial differential equations", 2019; Daw et al., "Mitigating Propagation Failures in Physics-informed Neural Networks using Retain-Resample-Release (R3) Sampling", 2023] for more details). We can cast this problem using (P) by parametrizing the solution $u$ using, e.g., a NN, as in
>
> $$
> \begin{align*}
> \text{minimize}\_{\theta \in \Theta} &\;\|\theta\|^2 \\\\
> \text{subject to} & \\; \mathbb{E}\_{\mathbb{P}\_1}[(\mathcal{D}\[u\_{\theta}\](x,t))^2] = 0 \\\\
> & \\; \mathbb{E}\_{\mathbb{P}\_2}\[(u\_{\theta}(x,t) - h(x,t))^2\] = 0 \\\\
> & \\; \mathbb{E}\_{\mathbb{P}\_3}\[(u\_{\theta}(x,0) - h(x,0))^2\] = 0
> \end{align*}
> $$
>
> The distributions $\mathbb{P}_1$, $\mathbb{P}_2$, and $\mathbb{P}_3$ are taken to be uniform over $\Omega \times (0,T]$, $\partial \Omega \times (0,T]$, and $\Omega \cup \partial\Omega$ respectively. Below we show preliminary results for a convection problem ($\beta=30$) with sinusoidal boundary conditions following the experimental setup of [Daw et al.] (the starred column was taken from [Daw et al., Table 1] and represent the mean of 5 runs). Note that these improvements come at essentially no additional computational cost, since the dual variable update (step 7 of Algorithm 1) is almost trivial.
>
> | PINN (5 runs)* | Constrained (5 runs) |
> | -------------- | -------------------- |
> | 2.81           | 2.06                 |
>
> A supervised learning example based on the CIFAR-10 dataset can be found in our response to `Reviewer 1hsE`. If the reviewer finds it beneficial, we can include this example and results in the camera-ready.
>
> That being said, we sustain that equality constraints are particularly well-suited for fairness. Indeed, as the reviewer points out, the goal of fair learning is often to obtain the best possible accuracy (performance) while remaining fair. As this manuscript proves, this is exactly what the constrained learning problem (P) can achieve in contrast to penalty approaches. Though certain application may accommodate relaxations of, e.g., demographic parity, that may not always be the case. It is more appropriate in these cases use an equality constraint as in (P-DP) than to try to find a tight relaxation $\epsilon$ that achieves such strict demands (in our example, even $\epsilon=0.05$ substantially changes the solution).
>
> Being accurate may also mean matching prior decisions that are themselves biased. Then, improving fairness requires making predictions that are different from those in the dataset (reducing "accuracy"). The definition of algorithmic fairness in (P-F), which we dub "prescriptive fairness," is more appropriate in these settings. Indeed, while (P-DP) can "overcome" biases that affect the prediction within the groups compared to the population (e.g., too many positive predictions for one group compared to another), (P-F) can *simultaneously* overcome predictive bias affecting the population (e.g., too many positive predictions in general). This is clear from Figure 2a: unconstrained and DP achieve essentially the same predicted recidivism rate. On the other hand, (P-F) enables other desired predictive rates to be achieved uniformly across group (Figure 2b).
>
> We should emphasize at this point that equalities and inequalities are not competing tools to solve fair learning problems, but rather complementary ones. Indeed, (P) accomodates both types of constraints and the most useful type will depend on the application. In fact, fair learning problems may need to involve both, as we detail in our response to `Reviewer SL24`.
>
>
> > **W2: Figure 1 shows that the approximate algorithm with a small tolerance for the inequality constraints works as good as the exact algorithm with equality constraints... This example does not support the claim "small values lead to numerical issues and worse generalization bounds (although it is true)**
>
> As we argue in Remark 2.1 and Appendix D, there is a close relation between (P) with an equality constraint as in $\mathbb{E}[h_j(f_{\theta}(x), y)] =0$ and (P) when that constraint is replaced with two inequality constraints as in $-\epsilon \leq \mathbb{E}[h_j(f_{\theta}(x), y)] \leq \epsilon$. Indeed, it is straightforward that both problems become the same as $\epsilon$ vanishes. Hence, it is natural to expect that their solutions would be similar. There is also a close *numerical* relation between their dual problems which, though not equivalent, are again very similar (see Appendix D.2).
>
> That being said, current generalization guarantees for (inequality) constraints do not hold as $\epsilon$ vanishes (see, e.g., [11,12]). Additionally, it is not straightforward to determine how small $\epsilon$ must be to achieve the performance of the equality-constrained problem. In our example, it has to be as small as $10^{-4}$ (Figure 1), but it could be even smaller for more sensitive problems. As such, there is a need for a new theory and algorithm that can accomodate equality constraints (i.e., vanishing $\epsilon$). We again emphasize that equalities and inequalities are not competing solutions, but complementary tools to tackle constrained learning problems.
>
> Yet, the reviewer has a point that Figure 1 does not support the claim that "small $\epsilon$ lead to numerical issues and worse generalization bounds." That is, in fact, not necessarily the case and depends on how equality constraints are turned into inequalities. Indeed, as we point out in Remark 2.1 and Appendices D.1, replacing an interpolation constraint by the mean-square error can lead to arbitrarily large dual variables as $\epsilon$ vanish. On the other hand, under the conditions of Theorem 3.1, this does not happen when directly relaxing (P) as in Lemma B.5. We will extend Remark 2.1 (line 111) to ensure that this is clear in the camera-ready version of the manuscript.
>
>
> > **W3: In summary, I’m not entirely convinced that developing theory and optimization algorithms for equality constrained learning is important, however, I do believe this line of work is valuable and will offer useful contributions.**
>
> As the reviewer points out in W1, a variety of ML problems are inherently equality constrained learning problems. Yet, as argued in Remark 2.1 and the response to W2, current generalization guarantees do not hold for constrained learning problems containing equality constraints (e.g., [11,12]). Additionally, it is not clear beforehand (i.e., without knowing the equality solution) how tight the inequality relaxation should be to remain close to the equality-constrained problem. Indeed, notice from Figure 1 how much the solution changes even for $\epsilon=0.05$. What is more, certain relaxations may lead to numerical issues (large dual variables, see Appendix D.1). As such, we believe that the theoretical and algorithmic contributions of this manuscript are indeed important. We will make these points clearer in the introduction of the camera-ready version of this paper.
>
>
> > **Q1: In Remark 2.1, ... the authors refer to Theorem 3.1. However, it is not immediately clear to me how Theorem 3.1 supports this claim.**
>
> As we detailed in W2, the quality and generalization properties of the relaxed problem depend on how the equality constraints are relaxed to inequality constraints. Remark 2.1, in particular, refer to the benign, convex learning problem ($\text{P}_e$) presented in Appendix D.1, where we prove that, when replacing interpolation equalities by a bound on the mean squared error as in ($\text{P}_i$), the magnitude of the Lagrange multipliers $||\gamma^\star||$ become arbitrarily large as $\epsilon$ vanishes. Since the guarantees in Theorem 3.1 are proportional to $||\gamma^\star||$, they therefore become vacuous for this problem as $\epsilon$ vanishes. We therefore obtain "worse generalization bounds arising from large Lagrange multipliers (see Thm 3.1)" (Remark 2.1, line 112). We will extend Remark 2.1 to ensure this point (and those from the response of W2) are clear in the camera-ready version.
>
>
> > **Q2: There are some minor formatting issues.**
>
> We will copy-edit the whole manuscript to correct such inconsistencies.

---

> > ### Comment · Reviewer_Q9iw · 2025-08-08
> >
> > I sincerely appreciate the authors’ thoughtful and detailed responses to my questions and concerns. Including another example could further highlight the necessity of developing the new theory and algorithm. Overall, I am pleased with the suggested revisions and have raised my score to 5.

---

### Note · Authors · 2025-08-14

We thank the reviewers for their valuable feedback to improve the clarity and strength of the manuscript. Below, we summarise the main discussion points addressed in detail in our responses.

**Contributions:**
The paper provides generalization guarantees for risk minimization with statistical equality constraints, an underexplored problem (Rev. `Q9iw`, `1hse`, "Strengths"). As noted in our response to Rev. `Q9iw` (W2), while equalities can be approximated by inequalities with tolerance $\epsilon$, existing guarantees for inequalities [11,12] fail as $\epsilon$ vanishes. Algorithmically, it is also unclear how small $\epsilon$ must be to meet the original equality requirements, stressing the importance of these theoretical (Thm 4.1) and algorithmic (Alg. 1) contributions (see also response to Rev. `Q9iw` W3).

**Empirical Evaluation:**
The manuscript already illustrates the use of Alg. 1 in two fairness applications. As discussed in our response to Rev. `Q9iw` (W1), fair ML often seeks to _balance_ fairness with accuracy, which our duality-based method does explicitly unlike penalty methods, making it a key instance of equality-constrained learning. In our response to Rev. `Q9iw` (W1) and `1hse` (W2), we also presented two other examples of equality constraints: (1) approximating PDE solutions and (2) training interpolating classifiers with disaggregated class-wise interpolation constraints. Preliminary results show (1) outperforms a fixed-penalty baseline in relative L2 error (2.81 vs 2.06) for a convection PDE with $\beta=30$ and (2) outperforms ERM in test accuracy (85.18% vs 86.06%) on CIFAR-10. We will include these applications and experiments in the main manuscript.


**Clarifications and typos:**
We thank the reviewers for identifying points needing clarification (Rev. `Q9iw` W2, Q1; Rev. `SL24` W2; Rev. `8xzy` Q1) and typos/formatting issues (Rev. `Q9iw` Q2; Rev. `8xzy` Q2). They will be corrected during the copy-editing of the final version. We will also amend the title to "Learning with Statistical Equality Constraints" as suggested by Rev. `8xzy` (W1).

---

### Decision · Program_Chairs · 2025-09-17

**Decision:**

Accept (poster)

**Comment:**

The reviewers, the authors, and I as Area Chair all agree that the submission is of high enough quality for publication at NeurIPS. Congratulations!